# Towards Sample-efficient Overparameterized Meta-learning

**Yue Sun**
University of Washington
yuesun@uw.edu

**Adhyyan Narang**
University of Washington
adhyyan@uw.edu

**Halil Ibrahim Gulluk**
Bogazici University
hibrahimgulluk@gmail.com

**Samet Oymak**
University of California, Riverside
oymak@ece.ucr.edu

**Maryam Fazel**
University of Washington
mfazel@uw.edu

## Abstract

An overarching goal in machine learning is to build a generalizable model with a small number of samples. To this end, overparameterization has been the subject of immense interest to explain the generalization ability of deep nets even when the size of the dataset is smaller than that of the model. While prior literature focuses on the classical supervised setting, this paper aims to demystify overparameterization for meta-learning. Here we have a sequence of linear-regression tasks and we ask: (1) Given earlier tasks, what is the optimal linear representation of features for a new downstream task? and (2) How many samples do we need to build this representation? This work shows that surprisingly, overparameterization arises as a natural answer to these fundamental meta-learning questions. Specifically, for (1), we first show that learning the optimal representation coincides with the problem of designing a task-aware regularization to promote inductive bias. This inductive bias explains how the downstream task actually benefits from overparameterization, in contrast to prior works on few-shot learning. For (2), we develop a theory to explain how feature covariance can implicitly help reduce the sample complexity well below the degrees of freedom and lead to small estimation error. We then integrate these findings to obtain an overall performance guarantee for our meta-learning algorithm. Numerical experiments on real and synthetic data verify our insights on overparameterized meta-learning.

## 1 Introduction

In a multitude of machine learning (ML) tasks with limited data, it is crucial to build accurate models in a sample-efficient way. Constructing a simple yet informative representation of features is a critical component of learning a model that generalizes well to an unseen test set. The field of meta-learning dates back to [8, 4] and addresses this challenge by transferring insights across distinct but related tasks. Usually, the meta-learner first (1) learns a feature-representation from previously seen tasks and then (2) uses this representation to succeed at an unseen task. The first phase is called representation learning and the second is called few-shot learning. Such information transfer between tasks is the backbone of modern transfer and multitask learning and finds ubiquitous applications in image classification [14], machine translation [6] and reinforcement learning [17].

Recent literature in ML theory has posited that overparameterization can be beneficial to generalization in traditional single-task setups for both regression [28, 38, 3, 32, 29] and classification [31, 30] problems. Empirical literature in deep learning suggests that overparameterization is of interest for both phases of meta-learning as well. Deep networks are stellar representation learners despite containing many more parameters than the sample size. Additionally, overparameterization

is observed to be beneficial in the few-shot phase for transfer-learning in Figure 1(a). A ResNet-50 network pretrained on Imagenet was utilized to obtain a representation of $R$ features for classification on CIFAR-10. All layers except the final (softmax) layer are frozen and are treated as a fixed feature-map. We then train the final layer of the network for the downstream task which yields a linear classifier on pretrained features. The figure plots the effect of increasing $R$ on the test error on CIFAR-10, for different choices of training size $n_2$. For each choice of $n_2$, increasing $R$ beyond $n_2$ is seen to reduce the test-error. These findings are corroborated by [17] (MAML) and [37], who successfully use a transfer learning method that adapts a pre-trained model, with 112980 parameters, to downstream tasks with only 1-5 new training samples.

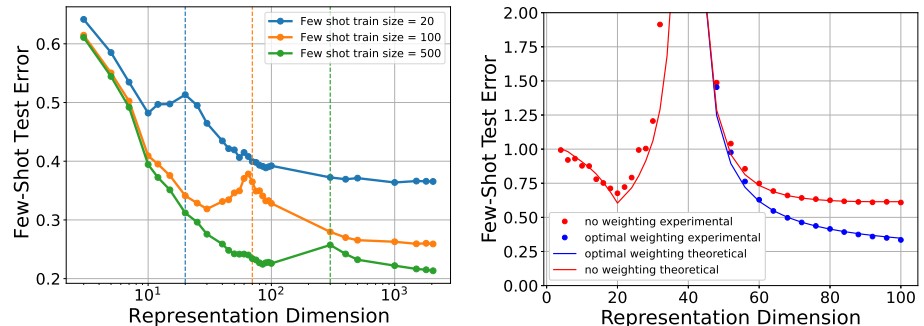

Figure 1: **Illustration of the benefit of overparameterization in the few-shot phase.** (a) Double-descent in transfer learning: dashed lines indicate the location where the number of features $R$ exceed the number of training points; i.e., the transition from under to over-parameterization. The experimental details are contained in the supplement. (b) Illustration of the benefit of using Weighted minL2-interpolation in Definition 3 (blue). See Remark 1 for details and discussion.

In Figure 1(b), we consider a sequence of *linear* regression tasks and plot the few-shot error of our proposed projection and eigen-weighting based meta-learning algorithm for a fixed few-shot training size, but varying dimensionality of features. The resulting curve looks similar to Figure 1(a) and suggests that the observations regarding overparameterization for meta-learning in neural networks can, to a good extent, be captured by linear models, thus motivating their detailed study. This aligns with trends in recent literature: while deep nets are nonlinear, recent advances show that linearized problems such as kernel regression (e.g., via neural tangent kernel [20, 16, 23, 34, 12]) provide a good proxy to understand some of the theoretical properties of practical overparameterized deep nets.

However, existing analysis of subspace-based meta-learning algorithms for both the representation learning and few-shot phases of linear models have typically focused on the classical *underparameterized regime*. These works (see Paragraphs 2-3 of Sec. 1.2) consider the case where representation learning involves projection onto a lower-dimensional subspace. On the other hand, recent works on double descent shows that an *overparameterized* interpolator beats PCA-based method. to build upon these results to develop a theoretical understanding of overparameterized meta-learning[1].

## 1.1 Our contributions

This paper studies meta-learning when each task is a linear regression problem, similar in spirit to [36, 22]. In the representation learning phase, the learner is provided with training data from $T$ distinct tasks, with $n_1$ training samples per task: using this data, it selects a matrix $\boldsymbol{\Lambda} \in \mathbb{R}^{d \times R}$ with arbitrary $R$ to obtain a linear *representation* of features via the map $\boldsymbol{x} \to \boldsymbol{\Lambda}^\top \boldsymbol{x}$. In the few-shot learning phase, the learner faces a new task with $n_2$ training samples and aims to use the representation $\boldsymbol{\Lambda}^\top \boldsymbol{x}$ to aid prediction performance.

We highlight that obtaining the representation consists of two steps: first the learner projects $\boldsymbol{x}$ onto $R$ basis directions, and then performs *eigen-weighting* of each of these directions, as shown in Figure 2(b). The overarching goal of this paper is to propose a scheme to use the knowledge gained from earlier tasks to choose $\boldsymbol{\Lambda}$ that minimizes few-shot risk. This goal enables us to engage with important questions regarding overparameterization:

---

[1] The code for this paper is in `https://github.com/sunyue93/Rep-Learning`.

**Q1:** What should the size $R$ and the representation $\mathbf{\Lambda}$ be to minimize risk at the few-shot phase?

**Q2:** Can we learn the $Rd$ dimensional representation $\mathbf{\Lambda}$ with $N \ll Rd$ samples?

The answers to the questions above will shed light on whether overparameterization is beneficial in few-shot learning and representation learning respectively. Towards this goal, we make several contributions to the finite-sample understanding of *linear* meta-learning, under assumptions discussed in Section 2. Our results are obtained for a general data/task model with *arbitrary task covariance* $\mathbf{\Sigma}_\beta$ *and feature covariance* $\mathbf{\Sigma}_F$ which allows for a rich set of observations.

**Optimal representation for few-shot learning.** As a stepping stone towards the goal of characterizing few-shot risk for different $\mathbf{\Lambda}$, in Section 3 we first consider learning with **known covariances $\mathbf{\Sigma}_T$** and $\mathbf{\Sigma}_F$ respectively (Algorithm 1). Compared to projection-only representations in previous works (see Paragraphs 2-3 of Sec. 1.2), our scheme applies *eigen-weighting* matrix $\mathbf{\Lambda}^*$ to incentivize the optimizer to place higher weight on promising eigen-directions. This eigen-weighting procedure has been shown in the single-task case to be extremely crucial to avail the benefit of overparameterization [5, 29, 32]: it captures an inductive bias that promotes certain features and demotes others. We show that the importance of eigen-weighting extends to the multi-task case as well.

**Canonical task covariance.** Our analysis in Section 3 also reveals that, the optimal subspace and representation matrix are closed-form functions of the *canonical task covariance* $\tilde{\mathbf{\Sigma}}_T = \mathbf{\Sigma}_F^{1/2} \mathbf{\Sigma}_T \mathbf{\Sigma}_F^{1/2}$, which captures the feature saliency by summarizing the feature and task distributions.

**Representation learning.** In practice, task and feature covariances (and hence the canonical covariance) are rarely known apriori. However, we can estimate the principal subspace of the canonical task covariance $\tilde{\mathbf{\Sigma}}_T$ (which has a degree of freedom (DoF) of $\Omega(Rd)$) from data. In Section 4 we first present empirical evidence that feature covariance $\mathbf{\Sigma}_F$ is "positively correlated" with $\tilde{\mathbf{\Sigma}}_T$. Then we propose an efficient algorithm based on Method-of-Moments (MoM), and show that the sample complexity of representation learning is well below $\mathcal{O}(Rd)$ due to the inductive bias. Our sample complexity bound depends on interpretable quantities such as *effective*

| $\mathbf{\Sigma}_F$ | Feature covariance |
|---|---|
| $\mathbf{\Sigma}_T$ | Task covariance |
| $\tilde{\mathbf{\Sigma}}_T$ | Canonical task covariance |
| $n_1$ | Samples per each earlier task |
| $T$ | Number of earlier tasks |
| $N$ | Total sample size $T \times n_1$ |
| $n_2$ | Samples for new task |
| $\mathbf{\Lambda}$ | Eigen-weighting matrix |

Table 1: Main notation

*ranks* $\mathbf{\Sigma}_F, \tilde{\mathbf{\Sigma}}_T$ and improves over prior art (e.g., [22, 36]), even though the prior works were specialized to low-rank $\tilde{\mathbf{\Sigma}}_T$ and identity $\mathbf{\Sigma}_F$ (see Table 2).

**End to end meta-learning guarantee.** In Section 5, we consider the generalization of Section 3, where we have only estimates of the covariances instead of perfect knowledge. This leads to an overall meta-learning guarantee in terms of $\mathbf{\Lambda}^*$, $N$ and $n_2$ and uncovers a bias-variance tradeoff: As $N$ decreases, it becomes more preferable to use a smaller $R$ (more bias, less variance) due to inaccurate estimate of the weak eigen-directions of $\tilde{\mathbf{\Sigma}}_T$. In other words, we find that overparameterization is only beneficial for few-shot learning if the quality of representation learning is sufficiently good. This explains why, in practice, increasing the representation dimension may not help reduce few-shot risk beyond a certain point (see Fig. 5).

## 1.2 Related work

**Overparameterized ML and double-descent** The phenomenon of double-descent was first discovered by [5]. This paper and subsequent works on this topic [3, 32, 31, 29, 10] emphasize the importance of the right prior (sometimes referred to as inductive bias or regularization) to avail the benefits of overparameterization. However, an important question that arises is: where does this prior come from? Our work shows that the prior can come from the insights learned from related previously-seen tasks. Section 3 extends the ideas in [33, 38] to depict how the optimal representation described can be learned from imperfect covariance estimates as well.

**Theory for representation learning** Recent papers [22, 21, 36, 15] propose the theoretical bounds of representation learning when the tasks lie in an exactly $r$ dimensional subspace. [22, 21, 36] discuss method of moment estimators and [36, 15] discuss matrix factorized formulations. [36] shows that the number of samples that enable meaningful representation learning is $\mathcal{O}(dr^2)$. [22, 21, 36]

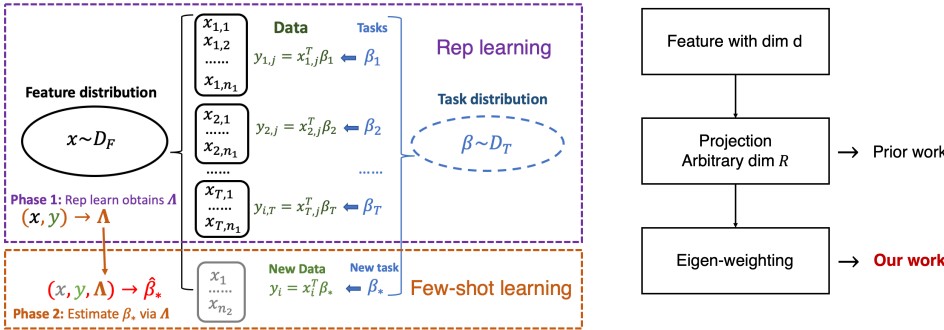

Figure 2: (a) Steps of the meta-learning algorithm. (b) Our representation-learning algorithm has two steps: projection and eigen-weighting. We focus on the use of overparameterization+weighting matrix (Def. 3), and compare this with overparameterization with simple projection (no eigen-weighting), and underparameterization (for which eigen-weighting has no impact and is equivalent to projection). [36, 22, 21, 15] study underparameterized projections only. To distinguish from eigen-weighting, we will refer to simple projections as subspace-based representations.

assume the features follow a standard normal distribution. We define a canonical covariance which handles arbitrary feature and task covariances. We also show that our estimator succeeds with $\mathcal{O}(dr)$ samples when $n_1 \sim r$, and extend the bound to general covariances with effective rank defined.

**Subspace-based meta learning** With tasks being low rank, [22, 21, 36, 18, 15] do few-shot learning in a low dimensional space. [39, 40] study meta-learning for linear bandits. [26] gives information theoretic lower and upper bounds. [7] proposes subspace-based methods for nonlinear problems such as classification. We investigate a representation with arbitrary dimension, specifically interested in overparameterized case and show it yields a smaller error with general task/feature covariances. Related work [15] provides results on overparameterized representation learning, but [15] requires number of samples per pre-training task to obey $n_1 \gtrsim d$, whereas our results apply as soon as $n_1 \gtrsim 1$.

**Mixed Linear Regression (MLR)** In MLR [41, 24, 11], multiple linear regression are executed, similar to representation learning. The difference is that, the tasks are drawn from a finite set, and number of tasks can be larger than $d$ and not necessarily low rank. [25, 9, 27] propose sample complexity bounds of representation learning for mixed linear regression. They can be combined with other structures such as binary task vectors [2] and sparse task vectors [1].

## 2 Problem Setup

The problem we consider consists of two phases:

1. Representation learning: Prior tasks are used to learn a suitable representation to process features.
2. Few-shot learning: A new task is learned with a few samples by using the suitable representation.

This section defines the key notations and describes the data generation procedure for the two phases. In summary, we study linear regression tasks, the features and tasks are generated randomly, i.i.d. from their associated distributions $\mathcal{D}_T$ and $\mathcal{D}_F$, and the two phases share the same feature and task distributions. The setup is summarized in Figure 2(a).

### 2.1 Data generation

**Definition 1 (Task and feature distributions)** *Throughout, $\mathcal{D}_T$ and $\mathcal{D}_F$ denote the distributions of tasks $\boldsymbol{\beta}_i$ and features $\boldsymbol{x}_{ij}$ respectively. These distributions are subGaussian, zero-mean with corresponding covariance matrices $\boldsymbol{\Sigma}_T$ and $\boldsymbol{\Sigma}_F$.*

**Definition 2 (Data distribution for a single task)** *Given a specific realization of task vector $\boldsymbol{\beta} \sim \mathcal{D}_T$, the corresponding label/input distribution $(y, \boldsymbol{x}) \sim \mathcal{D}_{\boldsymbol{\beta}}$ is obtained via $y = \boldsymbol{x}^{\top}\boldsymbol{\beta} + \varepsilon$ where $\boldsymbol{x} \sim \mathcal{D}_F$ and $\varepsilon$ is zero-mean subgaussian noise with variance $\sigma^2$.*

**Data for Representation Learning (Phase 1).** We have $T$ tasks, each with $n_1$ training examples. The task vectors $(\boldsymbol{\beta}_i)_{i=1}^T \subset \mathbb{R}^d$ are drawn i.i.d. from the distribution $\mathcal{D}_T$. The data for $i$th task is given by $(y_{ij}, \boldsymbol{x}_{ij})_{j=1}^{n_1} \overset{\text{i.i.d.}}{\sim} \mathcal{D}_{\boldsymbol{\beta}_i}$. In total, there are $N = T \times n_1$ examples.

**Data for Few-Shot Learning (Phase 2).** Sample task $\boldsymbol{\beta}_\star \sim \mathcal{D}_T$. Few-shot dataset has $n_2$ examples $(y_i, \boldsymbol{x}_i)_{j=1}^{n_2} \overset{\text{i.i.d.}}{\sim} \mathcal{D}_{\boldsymbol{\beta}_\star}$.

We use representation learning data to learn a representation of feature-task distribution, called eigen-weighting matrix $\boldsymbol{\Lambda}$ in Def. 3 below. The matrix $\boldsymbol{\Lambda}$ is passed to few-shot learning stage, helping learn $\boldsymbol{\beta}_\star$ with few data.

## 2.2 Training in Phase 2

We will define a weighted representation, called eigen-weighting matrix, and show how it is applied for few-shot learning. The matrix is learned during representation learning using the data from the $T$ tasks. Denote $\boldsymbol{X} \in \mathbb{R}^{n_2 \times d}$ whose $i^{\text{th}}$ row is $\boldsymbol{x}_i$, and $\boldsymbol{y} = [y_1, ..., y_m]^\top$. We are interested in studying the weighted 2-norm interpolator defined below for overparameterization regime $R \geq n_2$.

**Definition 3 (Eigen-weighting matrix and Weighted $\ell_2$-norm interpolator)** *Let the representation dimension be $R$, where $R$ is any integer between $1$ and $d$. We define an eigen-weighting matrix $\boldsymbol{\Lambda} \in \mathbb{R}^{d \times R}$ and the associated weighted $\ell_2$-norm interpolator*

$$\hat{\boldsymbol{\beta}}_{\boldsymbol{\Lambda}} = \arg\min_{\boldsymbol{\beta}} \|\boldsymbol{\Lambda}^\dagger \boldsymbol{\beta}\|_2 \quad s.t. \quad \boldsymbol{y} = \boldsymbol{X}\boldsymbol{\beta} \quad and \quad \boldsymbol{\beta} \in \text{range\_space}(\boldsymbol{\Lambda}).$$

The solution is equivalent to defining $\hat{\boldsymbol{\alpha}}_{\boldsymbol{\Lambda}} = \boldsymbol{\Lambda}^\dagger \hat{\boldsymbol{\beta}}_{\boldsymbol{\Lambda}}$ and solving an unweighted minimum 2-norm regression with features $\boldsymbol{X}\boldsymbol{\Lambda}$. This corresponds to our few-shot learning problem

$$\hat{\boldsymbol{\alpha}}_{\boldsymbol{\Lambda}} = \arg\min_{\boldsymbol{\alpha}} \|\boldsymbol{\alpha}\|_2 \quad \text{s.t.} \quad \boldsymbol{y} = \boldsymbol{X}\boldsymbol{\Lambda}\boldsymbol{\alpha}$$

from which we obtain $\hat{\boldsymbol{\beta}}_{\boldsymbol{\Lambda}} = \boldsymbol{\Lambda}\hat{\boldsymbol{\alpha}}_{\boldsymbol{\Lambda}}$. When there is no confusion, we can replace $\hat{\boldsymbol{\beta}}_{\boldsymbol{\Lambda}}$ with $\hat{\boldsymbol{\beta}}$. One can easily see that $\hat{\boldsymbol{\beta}} = \boldsymbol{\Lambda}(\boldsymbol{X}\boldsymbol{\Lambda})^\dagger \boldsymbol{y}$. We note that Definition 3 is a special case of the weighted ridge regression discussed in [38], as stated in Observation 1. An alternative equivalence between min-norm interpolation and ridge regression can be found in [32].

**Observation 1** *Let $\boldsymbol{X} \in \mathbb{R}^{n_2 \times d}$ and $\boldsymbol{y} \in \mathbb{R}^{n_2}$, define*

$$\hat{\boldsymbol{\beta}}_1 = \lim_{t \to 0} \text{argmin}_{\boldsymbol{\beta}} \|\boldsymbol{X}\boldsymbol{\beta} - \boldsymbol{y}\|_2^2 + t\boldsymbol{\beta}^\top (\boldsymbol{\Lambda}\boldsymbol{\Lambda}^\top)^\dagger \boldsymbol{\beta}, \ \boldsymbol{\beta} \in \text{column space of } \boldsymbol{\Lambda}. \qquad (2.1)$$

*We have that $\hat{\boldsymbol{\beta}}_1 = \hat{\boldsymbol{\beta}}$.*

# 3 Canonical Covariance and Optimal Representation

In this section, we ask the simpler question: if the covariances $\boldsymbol{\Sigma}_T$ and $\boldsymbol{\Sigma}_F$ are known, what is the best choice of $\boldsymbol{\Lambda}$ to minimize the risk of the interpolator from Definition 3? In general, the covariances are not known; however, the insights from this section help us study the more general case in Section 5. Define the risk as the expected error of inferring the label on the few-shot dataset,

$$\text{risk}(\boldsymbol{\Lambda}, \boldsymbol{\Sigma}_T, \boldsymbol{\Sigma}_F) = \boldsymbol{E}_{\boldsymbol{x}, y, \boldsymbol{\beta}}(y - \boldsymbol{x}^\top \hat{\boldsymbol{\beta}}_{\boldsymbol{\Lambda}})^2 = \boldsymbol{E}_{\boldsymbol{\beta}}(\hat{\boldsymbol{\beta}}_{\boldsymbol{\Lambda}} - \boldsymbol{\beta})^\top \boldsymbol{\Sigma}_F (\hat{\boldsymbol{\beta}}_{\boldsymbol{\Lambda}} - \boldsymbol{\beta}) + \sigma^2. \qquad (3.1)$$

The natural choice of optimization for choosing $\boldsymbol{\Lambda}$ would be to choose the weighting that minimizes the eventual risk of the learned interpolator.

$$\boldsymbol{\Lambda}^* = \arg\min_{\boldsymbol{\Lambda}' \in \mathbb{R}^{d \times R}} \text{risk}(\boldsymbol{\Lambda}', \boldsymbol{\Sigma}_T, \boldsymbol{\Sigma}_F) \qquad (3.2)$$

Since the label $y$ is bilinear in $x$ and $\beta$, we introduce whitened features $\tilde{\boldsymbol{x}} = \boldsymbol{\Sigma}_F^{-1/2} \boldsymbol{x}$ and associated task vector $\tilde{\boldsymbol{\beta}} = \boldsymbol{\Sigma}_F^{1/2} \boldsymbol{\beta}$. This change of variables ensures $\boldsymbol{x}^T \boldsymbol{\beta} = \tilde{\boldsymbol{x}}^T \tilde{\boldsymbol{\beta}}$; now, the task covariance in the transformed coordinates takes the form

$$\tilde{\boldsymbol{\Sigma}}_T = \boldsymbol{\Sigma}_F^{1/2} \boldsymbol{\Sigma}_T \boldsymbol{\Sigma}_F^{1/2},$$

which we call the **canonical task covariance**; it captures the joint behavior of feature and task covariances $\boldsymbol{\Sigma}_F, \boldsymbol{\Sigma}_T$. Below, we observe that the risk in Equation (3.1) is invariant to the change of co-ordinates that we have described above i.e it does not change when $\boldsymbol{\Sigma}_F^{1/2} \boldsymbol{\Sigma}_T \boldsymbol{\Sigma}_F^{1/2}$ is fixed and we vary $\boldsymbol{\Sigma}_F$ and $\boldsymbol{\Sigma}_T$.

---

**Algorithm 1** Constructing the optimal representation

---

**Require:** Projection dimension $R$, noise level $\sigma$, canonical covariance $\tilde{\Sigma}_T$, task covariance $\Sigma_F$.

1: **function** COMPUTEOPTIMALREP($R, \Sigma_F, \tilde{\Sigma}_T, \sigma, n_2$)
2:   $U_1, \Sigma_F^R, \tilde{\Sigma}_T^R, \sigma_R$ = COMPUTEREDUCTION($R, \Sigma_F, \tilde{\Sigma}_T, \sigma$)
3:   Optimization: Get $\theta^*$ from (OPT-REP).
4:   Map to eigenvalues: Set diagonal $\Lambda_R^* \in \mathbb{R}^{R \times R}$ with entries $\Lambda_{R,i}^* = (1/\theta_i^* - 1)^{-2}$.
5:   Lifting and feature whitening: $\Lambda^* \leftarrow U_1 (\Sigma_F^R)^{-1/2} \Lambda_R^*$.
6:   **return** $\Lambda^*$

7: **function** COMPUTEREDUCTION($R, \Sigma_F, \tilde{\Sigma}_T, \sigma$)
8:   Get eigen-decomposition $\tilde{\Sigma}_T = U \Sigma U^\top$.
9:   Principal eigenspace $U_1 \in \mathbb{R}^{d \times R}$ = the first $R$ columns of $U$.
10:   Top eigenvalues: Set $\tilde{\Sigma}_T^R = U_1^\top \tilde{\Sigma}_T U_1, \Sigma_F^R = U_1^\top \Sigma_F U_1$
11:   Equivalent noise level: $\sigma_R^2 \leftarrow \sigma^2 + \text{tr}(\tilde{\Sigma}_T) - \text{tr}(\tilde{\Sigma}_T^R)$.
12:   **return** $U_1, \Sigma_F^R, \tilde{\Sigma}_T^R, \sigma_R$

---

**Observation 2 (Equivalence to problem with whitened features)** *Let data be generated as in Phase 1. Denote $\tilde{\Sigma}_T = \Sigma_F^{1/2} \Sigma_T \Sigma_F^{1/2}$. Then risk$(\Sigma_F^{-1/2} \Lambda, \Sigma_T, \Sigma_F) = $ risk$(\Lambda, \tilde{\Sigma}_T, I)$.*

This observation can be easily verified by substituting the change-of-coordinates into Equation (3.1) and evaluating the risk.

The risk in (3.1) quantifies the quality of representation $\Lambda$; however it is not a manageable function of $\Lambda$ that can be straightforwardly optimized. In this subsection, we show that it is asymptotically equivalent to a different optimization problem, which can be easily solved by analyzing KKT optimality conditions. Theorem 1 characterizes this equivalence; the COMPUTEREDUCTION subroutine of Algorithm 1 calculates key quantities that are used in specifying the reduction, and the COMPUTEOPTIMALREP subroutine of Algorithm 1 uses the solution of the simpler problem to obtain a solution for the original.

**Assumption 1 (Bounded feature covariance)** *There exist positive constants $\Sigma_{\min}, \Sigma_{\max}$ such that $\Sigma_F$ is lower/upper bounded as follows: $0 \prec \Sigma_{\min} I \preceq \Sigma_F \preceq \Sigma_{\max} I$.*

**Assumption 2 (Joint diagonalizability)** *$\Sigma_F$ and $\Sigma_T$ are diagonal matrices.[2]*

**Assumption 3 (Double asymptotic regime)** *We let the dimensions and the sample size grow as $d, R, n_2 \to \infty$ at fixed ratios $\bar{\kappa} := d/n_2$ and $\kappa := R/n_2$.*

**Assumption 4** *The joint empirical distribution of the eigenvalues of $\Lambda_R$ and $\tilde{\Sigma}_T^R$ is given by the average of Dirac $\delta$'s: $\frac{1}{R} \sum_{i=1}^{R} \delta_{\Lambda_{R,i}, \sqrt{R} \tilde{\Sigma}_{T,i}^R}$. It converges to a fixed distribution as $d \to \infty$.*

With these assumptions, we can derive an analytical expression to quantify the risk of a representation $\Lambda$. We will then optimize this analytic expression to obtain a formula for the optimal representation.

**Theorem 1 (Asymptotic risk equivalence)** *Suppose Assumptions 1, 2, 3, 4 hold. Let $\xi > 0$ be the unique number obeying $n_2 = \sum_{i=1}^{R} \left(1 + (\xi \Lambda_i^2)^{-1}\right)^{-1}$. Define $\theta \in \mathbb{R}^R$ with entries $\theta_i = \frac{\xi \Lambda_i^2}{1 + \xi \Lambda_i^2}$ and calculate $\tilde{\Sigma}_T^R, \sigma_R$ using the COMPUTEREDUCTION procedure of Algorithm 1. Then, define the analytic risk formula*

$$f(\theta, \tilde{\Sigma}_T^R, n_2) = \frac{1}{n_2 - \|\theta\|_2^2} \left( n_2 \sum_{i=1}^{R} (1 - \theta_i)^2 \tilde{\Sigma}_{T,i}^R + (\|\theta\|_2^2 + 1)\sigma_R^2 \right). \tag{3.3}$$

*We have that*

$$\lim_{n_2 \to \infty} f(\theta, \tilde{\Sigma}_T^R, n_2) = \lim_{n_2 \to \infty} risk(\Sigma_F^{-1/2} \Lambda, \Sigma_T, \Sigma_F) \tag{3.4}$$

The proof of Theorem 1 applies the convex Gaussian Min-max Theorem (CGMT) in [35] and can be found in the Appendix B.2. We show that as dimension grows, the distribution of the estimator $\hat{\beta}$ converges to a Gaussian distribution and we can calculate the expectation of risk.

---

[2]This is equivalent to the more general scenario where $\Sigma_F$ and $\Sigma_T$ are jointly diagonalizable.

Theorem 1 provides us with a closed-form risk for any linear representation. Now, one can solve for the optimal representation by computing (OPT-REP) below. In order to do this, we propose an algorithm for the optimization problem in Appendix B.5 via a study of the KKT conditions for the problem [3].

$$\boldsymbol{\theta}^* = \arg\min_{\boldsymbol{\theta}} \ f(\boldsymbol{\theta}, \boldsymbol{\Sigma}_T, \boldsymbol{\Sigma}_F), \text{ s.t. } 0 \le \boldsymbol{\theta} < 1, \sum_{i=1}^{R} \boldsymbol{\theta}_i = n_2 \qquad \text{(OPT-REP)}$$

The optimal representation is[4] $\boldsymbol{\Lambda}_{R,i}^* = ((1/\boldsymbol{\theta}_i^* - 1)\xi)^{-2}$. The subroutine COMPUTEOPTIMALREP in Algorithm 1 summarizes this procedure.

**Remark 1** *Thm. 1 states that risk$(\boldsymbol{\Sigma}_F^{-1/2}\boldsymbol{\Lambda}, \boldsymbol{\Sigma}_T, \boldsymbol{\Sigma}_F)$ can be arbitrarily well-approximated by $f(\boldsymbol{\theta}, \tilde{\boldsymbol{\Sigma}}_T^R, n_2)$ if $n_2$ is sufficiently large. In Fig. 1(b), we set $\boldsymbol{\Sigma}_F = \boldsymbol{I}_{100}$, $\boldsymbol{\Sigma}_T = diag(\boldsymbol{I}_{20}, 0.1\boldsymbol{I}_{80})$, $n_2 = 40$. The curves in Fig1(b) are the finite dimensional approximation of $f$ (LHS of (3.4)); the dots are empirical approximations of the risk (RHS of (3.4)). We tested two cases when $\boldsymbol{\Lambda}$ is the optimal eigen-weighting or projection matrix with no weighting. Our theorem is corroborated by the observation that the dots and curves are visibly very close. The approximation is already accurate for the finite dimensional problem with just $n_2 = 40$.*

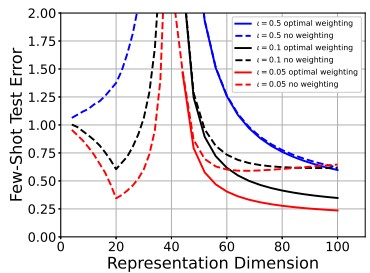

Figure 3: Theoretical risk of optimal representation. $\boldsymbol{\Sigma}_F = \boldsymbol{I}_{100}$, $\boldsymbol{\Sigma}_T = diag(\boldsymbol{I}_{20}, \iota\boldsymbol{I}_{80})$, $n_2 = 40$.

**The benefit of overparameterization.** Theorem 1 leads to an optimal eigen-weighting strategy via asymptotic analysis. In Figure 3, we plot the effect on the risk of increasing $R$ for different shapes of task covariance; the parameter $\iota$ controls how spiked $\boldsymbol{\Sigma}_T$ is, with a smaller value for $\iota$ indicating increased spikedness. For the underparameterized problem, the weighting does not have any impact on the risk. In the overparameterized regime, the eigen-weighted learner achieves lower few-shot error than its unweighted ($\boldsymbol{\Lambda} = \boldsymbol{I}$) counterpart, showing that eigen-weighting becomes critical.

The eigen-weighting procedure can introduce inductive bias during few-shot learning, and helps explain how optimal representation minimizing the few-shot risk can be overparameterized with $R \gg n_2$. We note that, an $R$ dimensional representation can be recovered by a $d$ dimensional representation matrix of rank $R$, thus the underparameterized case can never beat $d$ dimensional case in theory. The error with optimal eigen-weighting in overparameterized regime is smaller than the respective underparameterized counterpart. The error is lower with smaller $\iota$. It implies that, while $\tilde{\boldsymbol{\Sigma}}_T$ gets closer to low-rank, the excess error caused by choosing small dimension $R$ (equal to the gap $\sigma_R^2 - \sigma^2$ in Algo 1) is not as significant.

Low dimensional representations zero out features and cause bias. By contrast, when $\tilde{\boldsymbol{\Sigma}}_T \in \mathbb{R}^{d \times d}$ is not low rank, every feature contributes to learning with the importance of the features reflected by the weights. This viewpoint is in similar spirit to that of [19] where the authors devise a misspecified linear regression to demonstrate the benefits of overparameterization. Our algorithm allows arbitrary representation dimension $R$ and eigen-weighting.

## 4   Representation Learning

In this section, we will show how to estimate the useful distribution in representation learning phase that enables us to calculate eigen-weighting matrix $\boldsymbol{\Lambda}^*$. Note that $\boldsymbol{\Lambda}^*$ depends on the canonical covariance $\tilde{\boldsymbol{\Sigma}}_T = \boldsymbol{\Sigma}_F^{1/2}\boldsymbol{\Sigma}_T\boldsymbol{\Sigma}_F^{1/2}$. Learning the $R$-dimensional principal subspace of $\tilde{\boldsymbol{\Sigma}}_T$ enables us[5] to calculate $\boldsymbol{\Lambda}^*$. Denote this subspace by $\tilde{\boldsymbol{S}}_T$.

---

[3]In Sec. 5 the constraint is $\underline{\theta} \le \boldsymbol{\theta} \le 1 - \frac{d-n_2}{n_2}\underline{\theta}$ for robustness concerns.

[4]In the algorithm, $\xi = 1$ and $\boldsymbol{\Lambda}_{R,i} = (1/\boldsymbol{\theta}_i^* - 1)^{-2}$, because $c\boldsymbol{\Lambda}^*$ for any constant $c$ gives the same $\hat{\boldsymbol{\beta}}$.

[5]We also need to estimate $\boldsymbol{\Sigma}_F$ for whitening. Estimating $\boldsymbol{\Sigma}_F$ is rather easy and incurs smaller error compared to $\tilde{\boldsymbol{\Sigma}}_T$. The analysis is provided in the first part of Appendix B.

**Subspace estimation vs. inductive bias.** The subspace-based representation $\tilde{\boldsymbol{S}}_T$ has degrees of freedom$= Rd$. When $\tilde{\boldsymbol{\Sigma}}_T$ is exactly rank $R$ and features are whitened, [36] provides a sample-complexity lower bound of $\Omega(Rd)$ examples and gives an algorithm achieving $\mathcal{O}(R^2 d)$ samples. However, in practice, deep nets learn good representations despite overparameterization. In this section, recalling our **Q2**, we argue that the inductive bias of the feature distribution can implicitly accelerate learning the canonical covariance. This differentiates our results from most prior works such as [22, 21, 36] in two aspects:

1. Rather than focusing on a *low dimensional* subspace and assuming $N \gtrsim Rd$, we can estimate $\tilde{\boldsymbol{\Sigma}}_T$ or $\tilde{\boldsymbol{S}}_T$ in the overparameterized regime $N \lesssim Rd$.
2. Rather than assuming whitened features $\boldsymbol{\Sigma}_F = \boldsymbol{I}$ and achieving a sample complexity of $R^2 d$, our learning guarantee holds for arbitrary covariance matrices $\boldsymbol{\Sigma}_F, \boldsymbol{\Sigma}_T$. The sample complexity depends on *effective rank* and can be arbitrarily smaller than DoF. We showcase our bounds via a spiked covariance setting in Example 1 below.

For learning $\tilde{\boldsymbol{\Sigma}}_T$ or its subspace $\tilde{\boldsymbol{S}}_T$, we investigate the method-of-moments (MoM) estimator.

**Definition 4 (MoM Estimator)** *For* $1 \leq i \leq T$, *define* $\hat{\boldsymbol{b}}_{i,1} = 2n_1^{-1} \sum_{j=1}^{n_1/2} y_{ij} \boldsymbol{x}_{ij}$, $\hat{\boldsymbol{b}}_{i,2} = 2n_1^{-1} \sum_{j=n_1/2+1}^{n_1} y_{ij} \boldsymbol{x}_{ij}$. *Set*

$$\hat{\boldsymbol{M}} = n_1^{-1} \sum_{i=1}^{T} (\boldsymbol{b}_{i,1} \boldsymbol{b}_{i,2}^\top + \boldsymbol{b}_{i,2} \boldsymbol{b}_{i,1}^\top),$$

*The expectation of* $\hat{\boldsymbol{M}}$ *is equal to* $\boldsymbol{M} = \boldsymbol{\Sigma}_F \boldsymbol{\Sigma}_T \boldsymbol{\Sigma}_F$.

**Inductive bias in representation learning:** Recall that canonical covariance $\tilde{\boldsymbol{\Sigma}}_T = \boldsymbol{\Sigma}_F^{1/2} \boldsymbol{\Sigma}_T \boldsymbol{\Sigma}_F^{1/2}$ is the attribute of interest. However, feature covariance $\boldsymbol{\Sigma}_F^{1/2}$ term implicitly modulates the estimation procedure because the population MoM is not $\tilde{\boldsymbol{\Sigma}}_T$ but $\boldsymbol{M} = \boldsymbol{\Sigma}_F^{1/2} \tilde{\boldsymbol{\Sigma}}_T \boldsymbol{\Sigma}_F^{1/2}$. For instance, when estimating the principle canonical subspace $\tilde{\boldsymbol{S}}_T$, the degree of alignment between $\boldsymbol{\Sigma}_F$ and $\tilde{\boldsymbol{\Sigma}}_T$ can make or break the estimation procedure: If $\boldsymbol{\Sigma}_F$ and $\tilde{\boldsymbol{\Sigma}}_T$ have *well-aligned* principal subspaces, $\tilde{\boldsymbol{S}}_T$ will be easier to estimate since $\boldsymbol{\Sigma}_F$ will amplify the $\tilde{\boldsymbol{S}}_T$ direction within $\boldsymbol{M}$.

We verify the inductive bias on practical image dataset, reported in Appendix A. We assessed correlation coefficient between covariances $\tilde{\boldsymbol{\Sigma}}_T, \boldsymbol{\Sigma}_F$ via the canonical-feature alignment score defined as the correlation coefficient

$$\rho(\boldsymbol{\Sigma}_F, \tilde{\boldsymbol{\Sigma}}_T) := \frac{\langle \boldsymbol{\Sigma}_F, \tilde{\boldsymbol{\Sigma}}_T \rangle}{\|\boldsymbol{\Sigma}_F\|_F \|\tilde{\boldsymbol{\Sigma}}_T\|_F} = \frac{\text{trace}(\boldsymbol{M})}{\|\boldsymbol{\Sigma}_F\|_F \|\tilde{\boldsymbol{\Sigma}}_T\|_F}.$$

Observe that, the MoM estimator $\boldsymbol{M}$ naturally shows up in the alignment definition because the inner product of $\tilde{\boldsymbol{\Sigma}}_T, \boldsymbol{\Sigma}_F$ is equal to trace$(\boldsymbol{M})$. This further supports our inductive bias intuition. As reference, we compared it to canonical-identity alignment defined as $\frac{\text{trace}(\tilde{\boldsymbol{\Sigma}}_T)}{\sqrt{d} \|\tilde{\boldsymbol{\Sigma}}_T\|_F}$ (replacing $\boldsymbol{\Sigma}_F$ with $\boldsymbol{I}$). The canonical-feature alignment score is higher than the canonical-identity alignment score. This significant score difference exemplifies how $\boldsymbol{\Sigma}_F$ and $\tilde{\boldsymbol{\Sigma}}_T$ can synergistically align with each other (inductive bias). This alignment helps our MoM estimator defined below, illustrated by Example 1 (spiked covariance).

In the following subsections, let $N = n_1 T$ refer to the total tasks in representation-learning phase. Let $r_F = \mathbf{tr}(\boldsymbol{\Sigma}_F)$, $r_T = \mathbf{tr}(\boldsymbol{\Sigma}_T)$, and $\tilde{r}_T = \mathbf{tr}(\tilde{\boldsymbol{\Sigma}}_T)$. Define the approximate low-rankness measure of feature covariance by[6]

$$s_F = \min \ s'_F, \ \text{s.t.} \ s'_F \in \{1, ..., d\}, \ s'_F/d \geq \lambda_{s'_F+1}(\boldsymbol{\Sigma}_F)$$

We have two results for this estimator.

1. Generally, we can estimate $\boldsymbol{M}$ with $\mathcal{O}(r_F \tilde{r}_T^2)$ samples.
2. Let $n_1 \geq s_T$, we can estimate $\boldsymbol{M}$ with $\mathcal{O}(s_F \tilde{r}_T)$ samples.

Paper [36] has sample complexity $\mathcal{O}(dr^2)$ ($r$ is exact rank). Our sample complexity is $\mathcal{O}(r_F \tilde{r}_T^2)$. $r_F, \tilde{r}_T$ can be seen as effective ranks and our bounds are always smaller than [36]. We will discuss later in Example 1. Our second result says when $n_1 \geq s_T$, our sample complexity achieves the $\mathcal{O}(dr)$ which is proven a lower bound in [36].

---

[6]The $(s_F + 1)$-th eigenvalue is smaller than $s_F/d$. Note the top eigenvalue is 1.

| feature cov | $\mathbf{\Sigma}_F = \boldsymbol{I}, \mathbf{\Sigma}_T = \mathrm{diag}(\boldsymbol{I}_{s_T}, \boldsymbol{0})$ | | | $\mathbf{\Sigma}_F = \mathrm{diag}(\boldsymbol{I}_{s_F}, \iota_F \boldsymbol{I}_{d-s_F}),$ $\mathbf{\Sigma}_T = \mathrm{diag}(\boldsymbol{I}_{s_T}, \iota_T \boldsymbol{I}_{d-s_T})$ | | |
|---|---|---|---|---|---|---|
| estimator | sample $N$ | sample $n_1$ | error | sample $N$ | sample $n_1$ | error |
| MoM | $ds_T^2$ | 1 | $(ds_T^2/N)^{1/2}$ | $r_F r_T^2$ | 1 | $(r_F r_T^2/N)^{1/2}$ |
| MoM | $ds_T$ | $s_T$ | $(s_T/n_1)^{1/2}$ | $r_F r_T$ | $r_T$ | $(r_T/n_1)^{1/2}$ |

Table 2: **Right side:** Sample complexity and error of MoM estimators. $s_F$ ($s_T$) is the dimension of the principal eigenspace of the feature (task) covariance. $r_F = s_F + \iota_F(d - s_F)$, $r_T = s_T + \iota_T(d - s_T)$ are the effective ranks. **Left side:** This is the well-studied setting of identity feature covariance and low-rank task covariance. Our bound in the second row is the first result to achieve optimal sample complexity of $\mathcal{O}(ds_T)$ (cf. [36, 22]).

**Theorem 2** *Let data be generated as in Phase 1. Assume* $\|\mathbf{\Sigma}_F\|, \|\mathbf{\Sigma}_T\| = 1$ *for normalization[7].*

*1. Let $n_1$ be a even number. Then with probability at least $1 - N^{-100}$,*

$$\|\hat{\boldsymbol{M}} - \boldsymbol{M}\| \lesssim (\tilde{r}_T + \sigma^2)\sqrt{\frac{r_F}{N}} + \sqrt{\frac{r_T}{T}}.$$

*2. Assume $T \geq s_F$. If $n_1 \gtrsim \tilde{r}_T + \sigma^2$, then with probability at least $1 - CT^{-100}$*

$$\|\hat{\boldsymbol{M}} - \boldsymbol{M}\| \lesssim \left((\tilde{r}_T + \sigma^2)/n_1\right)^{1/2}.$$

*Denote the top-$R$ principal subspaces of $\boldsymbol{M}, \hat{\boldsymbol{M}}$ by $\boldsymbol{M}_{top}, \hat{\boldsymbol{M}}_{top}$ and assume the eigen-gap condition $\lambda_R(\boldsymbol{M}) - \lambda_{R+1}(\boldsymbol{M}) > 2\|\hat{\boldsymbol{M}} - \boldsymbol{M}\|$. Then a direct application of Davis-Kahan Theorem [13] bounds the subspace angle as follows*

$$angle(\boldsymbol{M}_{top}, \hat{\boldsymbol{M}}_{top}) \lesssim \|\hat{\boldsymbol{M}} - \boldsymbol{M}\|/(\lambda_R(\boldsymbol{M}) - \lambda_{R+1}(\boldsymbol{M})).$$

*Estimating eigenspace of canonical covariance.* Note that if $\mathbf{\Sigma}_F$ and $\mathbf{\Sigma}_T$ are aligned, (e.g. Example 1 below with $s_F = s_T = R$), then $\boldsymbol{M}_{\text{top}} = \tilde{\boldsymbol{S}}_T$ is exactly the principal subspace of $\tilde{\mathbf{\Sigma}}_T$. Theorem 2 indeed gives estimation error for the principal subspace of $\tilde{\mathbf{\Sigma}}_T$. Note that, such alignment is and more general requirement compared to related works which require whitened features [36, 22].

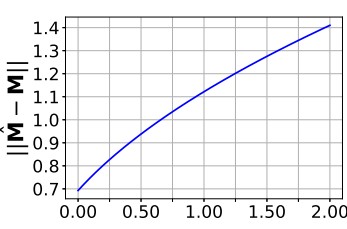

Figure 4: Error of MoM estimator

**Example 1 (Spiked $\tilde{\mathbf{\Sigma}}_T$, Aligned principal subspaces)**
*Suppose the spectra of $\mathbf{\Sigma}_F$ and $\tilde{\mathbf{\Sigma}}_T$ are bimodal as follows $\mathbf{\Sigma}_F = diag(\boldsymbol{I}_{s_F}, \iota_F \boldsymbol{I}_{d-s_F})$, $\mathbf{\Sigma}_T = diag(\boldsymbol{I}_{s_T}, \iota_T \boldsymbol{I}_{d-s_T})$. Set statistical error $Err_{T,N} := \sqrt{r_T^2 r_F/N} + \sqrt{r_T/T}$. When $\iota_T, \iota_F < 1$, $s_F \geq s_T$, the recovery error of $\tilde{\mathbf{\Sigma}}_T$ and its principal subspace $\tilde{\boldsymbol{S}}_T$ are bounded as*

$$angle(\hat{\boldsymbol{M}}_{top}, \tilde{\boldsymbol{S}}_T) \lesssim Err_{T,N} + \iota_F^2 \iota_T \quad and \quad \|\hat{\boldsymbol{M}} - \tilde{\mathbf{\Sigma}}_T\| \lesssim Err_{T,N} + \iota_F \iota_T.$$

The estimation errors for $\tilde{\mathbf{\Sigma}}_T, \tilde{\boldsymbol{S}}_T$ are controlled in terms of the effective ranks and the spectrum tails $\iota_F, \iota_T$. Typically $s_F s_T \gtrsim n_1$ so $\sqrt{r_T^2 r_F/N}$ term dominates the statistical error in practice. In Fig. 4 we plot the error of estimating $\boldsymbol{M}$ (whose principal subspace coincides with $\tilde{\mathbf{\Sigma}}_T$). $\mathbf{\Sigma}_F = \mathrm{diag}(\boldsymbol{I}_{30}, \iota \boldsymbol{I}_{70})$, $\mathbf{\Sigma}_T = \mathrm{diag}(\boldsymbol{I}_{30}, \boldsymbol{0}_{70})$. $T = N = 100$. We can see that the error increase with $\iota$ .

## 5  Robustness of Optimal Representation and Overall Meta-Learning Bound

In Section 3, we described the algorithm for computing the optimal representation with *known* distributions of features and tasks. In Section 4, we proposed the MoM estimator in representation learning phase to estimate the unknown covariance matrices. In this section, we study the algorithm's behaviors when we calculate $\mathbf{\Lambda}$ using the *estimated* canonical covariance, rather than the full-information setting of Section 3.

---

[7]This is simply equivalent to scaling $y_{ij}$, which does not affect the normalized error $\|\hat{\boldsymbol{M}} - \boldsymbol{M}\|/\|\boldsymbol{M}\|$. In the appendix we define $\mathcal{S} = \max\{\|\mathbf{\Sigma}_F\|, \|\mathbf{\Sigma}_T\|\}$ and prove the theorem for general $\mathcal{S}$.

Armed with the provably reliable estimators of Section 4, we can replace $\tilde{\boldsymbol{\Sigma}}_T$ and $\boldsymbol{\Sigma}_F$ in Algorithm 1 with our estimators. In this section, we inquire: how does the estimation error in covariance-estimation in representation learning stage affect the downstream few-shot learning risk? That says, we are interested in[8] $\text{risk}(\boldsymbol{\Lambda}, \boldsymbol{\Sigma}_T, \boldsymbol{\Sigma}_F) - \text{risk}(\boldsymbol{\Lambda}^*, \boldsymbol{\Sigma}_T, \boldsymbol{\Sigma}_F)$.

Let us replace the constraint in (OPT-REP) by $\underline{\theta} \le \boldsymbol{\theta} \le 1 - \frac{d-n_2}{n_2}\underline{\theta}$. This changes the "optimization" step in Algorithm 1. Theorem 3 does not require an explicit computation of the optimal representation by enforcing $\underline{\theta}$. Instead, we use the robustness of such a representation (due to its well-conditioned nature) to deduce its stability. That said, for practical computation of optimal representation, we simply use Algorithm 1. We can then evaluate $\underline{\theta}$ after-the-fact as the minimum singular value of this representation to apply Theorem 3 without assuming an explicit $\underline{\theta}$.

Let $\boldsymbol{\Lambda}_{\underline{\theta}}(R) = \text{COMPUTEOPTIMALREP}(R, \boldsymbol{\Sigma}_F, \hat{M}, \sigma, n_2)$ denote the estimated optimal representation and $\boldsymbol{\Lambda}_{\underline{\theta}}^*(R) = \text{COMPUTEOPTIMALREP}(R, \boldsymbol{\Sigma}_F, \tilde{\boldsymbol{\Sigma}}_T, \sigma, n_2)$ denote the true optimal representation, which cannot be accessed in practice. Below we present the bound of the whole meta-learning algorithm. It shows that a bounded error in representation learning leads to a bounded increase on the downstream few-shot learning risk, thus quantifying the robustness of few-shot learning to errors in covariance estimates.

**Theorem 3** *Let $\boldsymbol{\Lambda}_{\underline{\theta}}(R)$, $\boldsymbol{\Lambda}_{\underline{\theta}}^*(R)$ be as defined above, and $r_F = \text{tr}(\boldsymbol{\Sigma}_F)$, $r_T = \text{tr}(\boldsymbol{\Sigma}_T)$, $\tilde{r}_T = \text{tr}(\tilde{\boldsymbol{\Sigma}}_T)$. The risk of meta-learning algorithm satisfies[9]*

$$risk(\boldsymbol{\Lambda}_{\underline{\theta}}(R), \boldsymbol{\Sigma}_T, \boldsymbol{\Sigma}_F) - risk(\boldsymbol{\Lambda}_{\underline{\theta}}^*(R), \boldsymbol{\Sigma}_T, \boldsymbol{\Sigma}_F) \lesssim \frac{n_2^2}{d(R-n_2)(2n_2 - R\underline{\theta})\underline{\theta}} \left[ (\tilde{r}_T + \sigma^2)\sqrt{\frac{r_F}{N}} + \sqrt{\frac{r_T}{T}} \right].$$

Notice that as the number of previous tasks $T$ and total representation-learning samples $N$ observed increases, the risk of the estimated $\boldsymbol{\Lambda}_{\underline{\theta}}(R)$ approaches that of the optimal $\boldsymbol{\Lambda}_{\underline{\theta}}^*(R)$ as we expect. The result only applies to the overparameterized regime of interest $R > n_2$. The expression of risk in the underparameterized case is different, and covered by the second case of Equation(4.4) in [38]. We plot it in Fig 1(b) on the left side of the peak as a comparison.

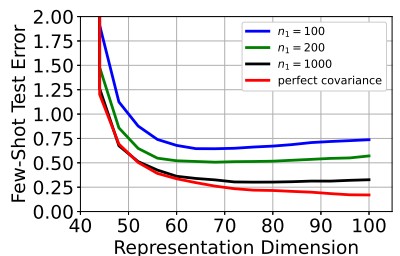

Figure 5: End to end learning guarantees. $d = 100, n_2 = 40, T = 200$, $\boldsymbol{\Sigma}_T = (\boldsymbol{I}_{20}, 0.05 \cdot \boldsymbol{I}_{80})$, $\boldsymbol{\Sigma}_F = \boldsymbol{I}_{100}$.

**Risk with respect to PCA level $R$.** In Fig. 5, we plot the error of the whole meta-learning algorithm. We simulate representation learning and get $\hat{M}$, use it to compute $\boldsymbol{\Lambda}$ and plot the theoretical downstream risk (experiments match, see Fig. 1 (b)). Mainly, we compare the behavior of Theorem 3 with different $R$. When $R$ grows, we search $\boldsymbol{\Lambda}$ in a larger space. The optimal $\boldsymbol{\Lambda}$ in a feasible *sub*set is always no better than searching in a larger space, thus the risk decreases with $R$ increasing. At the same time, representation learning error increases with $R$ since we need to fit a matrix in a larger space. In essence, this result provides a theoretical justification on a sweet-spot for the optimal representation. $d = R$ is optimal when $N = \infty$, i.e., representation learning error is 0. As $N$ decreases, there is a tradeoff between learning error and truncating small eigenvalues. Thus choosing $R$ adaptively with $N$ can strike the right bias-variance tradeoff between the excess risk (variance) and the risk due to suboptimal representation.

## 6 Conclusion

In this paper, we study the sample efficiency of meta-learning with linear representations. We show that the optimal representation is typically overparameterized and outperforms subspace-based representations for general data distributions. We refine the sample complexity analysis for learning arbitrary distributions and show the importance of inductive bias of feature and task. Finally we provide an end-to-end bound for the meta-learning algorithm showing the tradeoff of choosing larger representation dimension v.s. robustness against representation learning error.

---

[8]Note that Sec.6 of [38] gives the exact value of $\text{risk}(\boldsymbol{\Lambda}^*, \boldsymbol{\Sigma}_T, \boldsymbol{\Sigma}_F)$ so we have an end to end error guarantee.

[9]The bracketed expression applies first conclusion of Theorem 3. One can plug in the second as well.

## Acknowledgements

This work is supported in part by the NSF TRIPODS II grant DMS 2023166, NSF TRIPODS CCF 1740551, NSF CCF-2046816, NSF CCF 2007036, Army Research Office grant W911NF-21-1-0312, and the Moorthy Professorship at UW ECE.

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
