# Towards Sample-efficient Overparameterized Meta-learning

**Yue Sun**
University of Washington
yuesun@uw.edu

**Adhyyan Narang**
University of Washington
adhyyan@uw.edu

**Halil Ibrahim Gulluk**
Bogazici University
hibrahimgulluk@gmail.com

**Samet Oymak**
University of California, Riverside
oymak@ece.ucr.edu

**Maryam Fazel**
University of Washington
mfazel@uw.edu

## Abstract

An overarching goal in machine learning is to build a generalizable model with few samples. To this end, overparameterization has been the subject of immense interest to explain the generalization ability of deep nets even when the size of the dataset is smaller than that of the model. While the prior literature focuses on the classical supervised setting, this paper aims to demystify overparameterization for meta-learning. Here we have a sequence of linear-regression tasks and we ask: (1) Given earlier tasks, what is the optimal linear representation of features for a new downstream task? and (2) How many samples do we need to build this representation? This work shows that surprisingly, overparameterization arises as a natural answer to these fundamental meta-learning questions. Specifically, for (1), we first show that learning the optimal representation coincides with the problem of designing a task-aware regularization to promote inductive bias. We leverage this inductive bias to explain how the downstream task actually benefits from overparameterization, in contrast to prior works on few-shot learning. For (2), we develop a theory to explain how feature covariance can implicitly help reduce the sample complexity well below the degrees of freedom and lead to small estimation error. We then integrate these findings to obtain an overall performance guarantee for our meta-learning algorithm. Numerical experiments on real and synthetic data verify our insights on overparameterized meta-learning.

## Organization of the appendix

The appendix consists of the proof of our main results including the following parts:

- We included a short section and Figure 6 containing more experiments on real data. This verifies the positive correlation between the canonical task covariance and feature covariance across distinct datasets which supports the theory developed in Section 4.
- Optimal representation. The proof for optimal overparameterized representation is in Sec. B. We show that we can use an $R$ dimensional representation of feature for few-shot learning, and it can beat typical PCA (low dimensional/underparameterized) representation with optimal weighting matrix $\mathbf{\Lambda}^*$.
  - In Sec. B.1 we first prove Observation 1. In Remark 2 we analyze the projection from $d$ to $R$ dimensional space, where we calculate the PCA truncation noise.
  - In Sec. B.2 and B.3 we provide the asymptotic analysis of optimal weighting. By asymptotic we refer to the regime where $n_2, d \to \infty$ and the eigenvalues of task and feature covariance

35th Conference on Neural Information Processing Systems (NeurIPS 2021), Sydney, Australia.

matrices converge to a fixed distribution. We show that $\hat{\boldsymbol{\beta}}_{\boldsymbol{\Lambda}}$ converges to a Gaussian distribution parameterized by $\boldsymbol{\Lambda}$, and use it to express the risk.
– We extend the asymptotic case (infinite dimensional) to the non-asymptotic (finite dimensional) regime in Sec. B.4. We define the risk function with respect to representation matrix $\boldsymbol{\Lambda}$, and in Sec. B.5 solve for the optimal representation by minimizing risk.
- Representation learning. Sec. C includes the proof for the result about representation learning in Sec. 4, including the sample complexity and error guarantee of MoM estimators.
– We first analyze the estimation of feature covariance matrix $\boldsymbol{\Sigma}_F$ in Sec. C.1 which is the most straightforward.
– We prove the second result of Thm. 2 in Sec. C.2.2. With the assumption that each task has $\Omega(s_T)$ corresponding samples, the sample complexity is **reduced by a factor of** $s_T$ **compared to MoM**, which meets the *information theoretical lower bound* in [37].
– We extend the Bernstein type technique for obtaining the estimation error of $\hat{M}$ in Sec. C.2. The estimator given in [37], slightly different from ours, is also analyzed.
- End to end bound. We prove the robustness of the optimal representation in Sec. D, which leads to the overall error guarantee of the proposed meta-learning algorithm.

## Contents

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

}(\Lambda, \Sigma_T, \Sigma_F) = E_{x,y,\beta}(y - x^\top \hat{\beta}_\Lambda)^2 = E_\beta(\hat{\beta}_\Lambda - \beta)^\top \Sigma_F(\hat{\beta}_\Lambda - \beta) + \sigma^2. \qquad (3.1)$$

The natural choice of optimization for choosing $\Lambda$ would be to choose the weighting that minimizes the eventual risk of the learned interpolator.

$$\Lambda^* = \arg\min_{\Lambda' \in \mathbb{R}^{d \times R}} \text{risk}(\Lambda', \Sigma_T, \Sigma_F) \qquad (3.2)$$

Since the label $y$ is bilinear in $x$ and $\beta$, we introduce whitened features $\tilde{x} = \Sigma_F^{-1/2} x$ and associated task vector $\tilde{\beta} = \Sigma_F^{1/2}\beta$. This change of variables ensures $x^T\beta = \tilde{x}^T\tilde{\beta}$; now, the task covariance in the transformed coordinates takes the form

$$\tilde{\Sigma}_T = \Sigma_F^{1/2} \Sigma_T \Sigma_F^{1/2},$$

which we call the **canonical task covariance**; it captures the joint behavior of feature and task covariances $\Sigma_F, \Sigma_T$. Below, we observe that the risk in Equation (3.1) is invariant to the change of co-ordinates that we have described above i.e it does not change when $\Sigma_F^{1/2} \Sigma_T \Sigma_F^{1/2}$ is fixed and we vary $\Sigma_F$ and $\Sigma_T$.

**Observation 2 (Equivalence to problem with whitened features)** *Let data be generated as in Phase 1. Denote $\tilde{\Sigma}_T = \Sigma_F^{1/2} \Sigma_T \Sigma_F^{1/2}$. Then $\text{risk}(\Sigma_F^{-1/2}\Lambda, \Sigma_T, \Sigma_F) = \text{risk}(\Lambda, \tilde{\Sigma}_T, I)$.*

This observation can be easily verified by substituting the change-of-coordinates into Equation (3.1) and evaluating the risk.

The risk in (3.1) quantifies the quality of representation $\Lambda$; however it is not a manageable function of $\Lambda$ that can be straightforwardly optimized. In this subsection, we show that it is asymptotically equivalent to a different optimization problem, which can be easily solved by analyzing KKT optimality conditions. Theorem 1 characterizes this equivalence; the COMPUTEREDUCTION subroutine of Algorithm 1 calculates key quantities that are used in specifying the reduction, and the COMPUTEOP-TIMALREP subroutine of Algorithm 1 uses the solution of the simpler problem to obtain a solution for the original.

**Assumption 1 (Bounded feature covariance)** *There exist positive constants $\Sigma_{\min}$, $\Sigma_{\max}$ such that $\Sigma_F$ is lower/upper bounded as follows: $0 \prec \Sigma_{\min} I \preceq \Sigma_F \preceq \Sigma_{\max} I$.*

**Assumption 2 (Joint diagonalizability)** $\Sigma_F$ and $\Sigma_T$ are diagonal matrices.[1]

---
[1]This is equivalent to the more general scenario where $\Sigma_F$ and $\Sigma_T$ are jointly diagonalizable.

**Assumption 3 (Double asymptotic regime)** *We let the dimensions and the sample size grow as $d, R, n_2 \to \infty$ at fixed ratios $\bar{\kappa} := d/n_2$ and $\kappa := R/n_2$.*

**Assumption 4** *The joint empirical distribution of the eigenvalues of $\Lambda_R$ and $\tilde{\Sigma}_T^R$ is given by the average of Dirac $\delta$'s: $\frac{1}{R}\sum_{i=1}^{R}\delta_{\Lambda_{R,i},\sqrt{R}\tilde{\Sigma}_{T,i}^R}$. It converges to a fixed distribution as $d \to \infty$.*

With these assumptions, we can derive an analytical expression to quantify the risk of a representation $\Lambda$. We will then optimize this analytic expression to obtain a formula for the optimal representation.

**Theorem 1 (Asymptotic risk equivalence)** *Suppose Assumptions 1, 2, 3, 4 hold. Let $\xi > 0$ be the unique number obeying $n_2 = \sum_{i=1}^{R}\left(1 + (\xi\Lambda_i^2)^{-1}\right)^{-1}$. Define $\boldsymbol{\theta} \in \mathbb{R}^R$ with entries $\boldsymbol{\theta}_i = \frac{\xi\Lambda_i^2}{1+\xi\Lambda_i^2}$ and calculate $\tilde{\Sigma}_T^R, \sigma_R$ using the* COMPUTEREDUCTION *procedure of Algorithm 1. Then, define the analytic risk formula*

$$f(\boldsymbol{\theta}, \tilde{\Sigma}_T^R, n_2) = \frac{1}{n_2 - \|\boldsymbol{\theta}\|_2^2}\left(n_2\sum_{i=1}^{R}(1-\boldsymbol{\theta}_i)^2\tilde{\Sigma}_{T,i}^R + (\|\boldsymbol{\

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

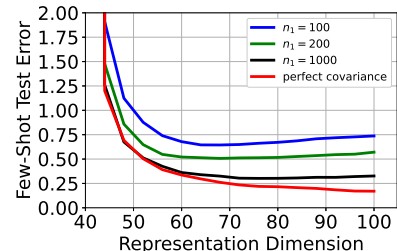

Figure 5: End to end learning guarantes. $d = 100, n_2 = 40, T = 200, \mathbf{\Sigma}_T = (\mathbf{I}_{20}, 0.05 \cdot \mathbf{I}_{80}), \mathbf{\Sigma}_F = \mathbf{I}_{100}$.

$\mathbf{\Lambda}$ in a larger space. The optimal $\mathbf{\Lambda}$ in a feasible *sub*set is always no better than searching in a larger space, thus the risk decreases with $R$ increasing. At the same time, representation learning error increases with $R$ since we need to fit a matrix in a larger space. In essence, this result provides a theoretical justification on a sweet-spot for the optimal representation. $d = R$ is optimal when $N = \infty$, i.e., representation learning error is $0$. As $N$ decreases, there is a tradeoff between learning error and truncating small eigenvalues. Thus choosing $R$ adaptively with $N$ can strike the right bias-variance tradeoff between the excess risk (variance) and the risk due to suboptimal representation.

## 6    Conclusion

In this paper, we study the sample efficiency of meta-learning with linear representations. We show that the optimal representation is typically overparameterized and outperforms subspace-based representations for general data distributions. We refine the sample complexity analysis for learning arbitrary distributions and show the importance of inductive bias of feature and task. Finally we provide an end-to-end bound for the meta-learning algorithm showing the tradeoff of choosing larger representation dimension v.s. robustness against representation learning error.

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

# A  Numerical verification of inductive bias for representation learning

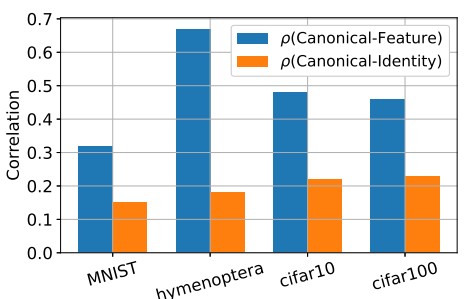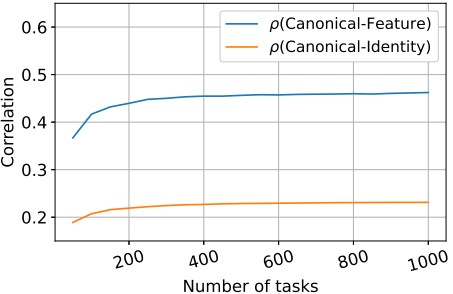

Figure 6: (a) Alignment of feature-task on image classification models. The result of MNIST uses the setting in Sec. 4. We apply the pretrained ResNet classification model on the other three datasets, compute the (last layer) feature/task covariances and get the alignments. The alignment is a measure of correlation which is denoted by $\rho$ here. (b) We use the cifar100 dataset, take the pretrained ResNet18 network and vary the number of tasks (i.e., varying the number of output classes of the neural net, also equivalent to number of rows of the last layer matrix $B$ defined below). The alignments increase with number of tasks.

We add a figure with experiments on a few image datasets. We take the pretrained ResNet18 neural network, and feed the images into it. For every image, we take the last (closest to output) layer output as the feature $\boldsymbol{x}$, which is of dimension $d = 512$. The weights of the last layer are the tasks, which is a $T \times d$ matrix (We call it $B$). $T = 1000$, each row of $B$ is a task vector. Then $Bx \in \mathbb{R}^T$ generates the label, whose each entry corresponds to each class. We calculate the feature and task covariance, as well as the alignments defined in Sec. 4. We can clearly see the inductive bias of every dataset.

# B  Analysis of optimal representation

## B.1  Proof of Observation 1 and equivalent noise

**Observation 1** *Let $\boldsymbol{\Lambda} \in \mathbb{R}^{d \times R}$, $\boldsymbol{X} \in \mathbb{R}^{n_2 \times d}$ and $\boldsymbol{y} \in \mathbb{R}^n_2$, and define*

$$\hat{\boldsymbol{\beta}} = \boldsymbol{\Lambda}(\boldsymbol{X}\boldsymbol{\Lambda})^{\dagger}\boldsymbol{y}, \tag{B.1}$$

$$\hat{\boldsymbol{\beta}}_1 = \lim_{t \to 0} \mathrm{argmin}_{\boldsymbol{\beta}} \|\boldsymbol{X}\boldsymbol{\beta} - \boldsymbol{y}\|^2 + t\boldsymbol{\beta}^{\top}(\boldsymbol{\Lambda}\boldsymbol{\Lambda}^{\top})^{\dagger}\boldsymbol{\beta} \tag{B.2}$$

*Then $\hat{\boldsymbol{\beta}}_1 = \hat{\boldsymbol{\beta}}$.*

**Proof** Denote the SVD $(\boldsymbol{X}\boldsymbol{\Lambda})^{\top} = \boldsymbol{U}\boldsymbol{\Sigma}\boldsymbol{V}^{\top}$, where $\boldsymbol{U} \in \mathbb{R}^{R \times R}, \boldsymbol{\Sigma} \in \mathbb{R}^{R \times n_2}, \boldsymbol{V} \in \mathbb{R}^{n_2 \times n_2}$.

$$
\begin{aligned}
\hat{\boldsymbol{\beta}}_1 &= \lim_{t \to 0} \mathrm{argmin}_{\boldsymbol{\beta}} \|\boldsymbol{X}\boldsymbol{\beta} - \boldsymbol{y}\|^2 + t\boldsymbol{\beta}^{\top}(\boldsymbol{\Lambda}\boldsymbol{\Lambda}^{\top})^{\dagger}\boldsymbol{\beta} \\
&= \lim_{t \to 0} (\boldsymbol{X}^{\top}\boldsymbol{X} + t(\boldsymbol{\Lambda}\boldsymbol{\Lambda}^{\top})^{\dagger})^{-1}\boldsymbol{X}\boldsymbol{y} \\
&= \lim_{s \to \infty} s\boldsymbol{\Lambda}(s\boldsymbol{\Lambda}^{\top}\boldsymbol{X}^{\top}\boldsymbol{X}\boldsymbol{\Lambda} + I)^{-1}\boldsymbol{\Lambda}^{\top}\boldsymbol{X}^{\top}\boldsymbol{y} \\
&= \lim_{s \to \infty} s\boldsymbol{\Lambda}(s\boldsymbol{U}\boldsymbol{\Sigma}\boldsymbol{V}^{\top}\boldsymbol{V}\boldsymbol{\Sigma}^{\top}\boldsymbol{U}^{\top} + I)^{-1}\boldsymbol{U}\boldsymbol{\Sigma}\boldsymbol{V}^{\top}\boldsymbol{y} \\
&= \lim_{s \to \infty} s\boldsymbol{\Lambda}(s\boldsymbol{U}\mathrm{diag}(\boldsymbol{\Sigma}^{\top}\boldsymbol{\Sigma} + I_{n_2}, I_{R-n_2})\boldsymbol{U}^{\top})^{-1}\boldsymbol{U}\boldsymbol{\Sigma}\boldsymbol{V}^{\top}\boldsymbol{y} \\
&= \lim_{s \to \infty} \boldsymbol{\Lambda}\boldsymbol{U}(\mathrm{diag}(\boldsymbol{\Sigma}^{\top}\boldsymbol{\Sigma}, I_{R-n_2}/s))^{-1}\boldsymbol{\Sigma}\boldsymbol{V}^{\top}\boldsymbol{y}. \\
&= \boldsymbol{\Lambda}(\boldsymbol{X}\boldsymbol{\Lambda})^{\dagger}\boldsymbol{y}
\end{aligned}
$$

∎

The risk of $\hat{\boldsymbol{\beta}}$ is given by

$$\mathrm{risk}(\hat{\boldsymbol{\beta}}) = \boldsymbol{E}(y - \boldsymbol{x}^\top \hat{\boldsymbol{\beta}}) = \boldsymbol{E}(\hat{\boldsymbol{\beta}} - \boldsymbol{\beta})^\top \boldsymbol{\Sigma}_F (\hat{\boldsymbol{\beta}} - \boldsymbol{\beta}) + \sigma^2.$$

In Sec. B.2, we study the asymptotic optimal representation. Below, we characterize the properties of the problem for fixed $\boldsymbol{\beta}$ and arbitrary input covariance $\boldsymbol{\Sigma}_F$. We first go over this and then discuss how to obtain the optimal representation $\boldsymbol{\Lambda}^*$ minimizing test risk.

**Remark 2** *Projection onto $R$ dimensional subspace. For the remaining proof after this part, we will mainly analyze the relation between $\boldsymbol{\Lambda}_R$ and $\boldsymbol{\theta}$ in Thm. 1, which lie in an $R$ dimensional subspace. Here we will build the connection from the $d$ dimensional problem to $R$ dimensional, mainly computing the equivalent noise below. The equivalent noise consists of original noise and the extra noise caused by PCA truncation.*

*Let $\boldsymbol{x}_R$ be the projection of $\boldsymbol{x}$ onto the $R$-dimensional subspace spanned by columns of $\boldsymbol{U}_1$, and $\boldsymbol{x}_{R\perp}$ is the projection of $\boldsymbol{x}$ onto the orthogonal complement. Namely, $\boldsymbol{x}_R = \boldsymbol{U}_1^\top \boldsymbol{x} \in \mathbb{R}^R$ and $\boldsymbol{x}_{R\perp} = \boldsymbol{U}_2^\top \boldsymbol{x} \in \mathbb{R}^{(d-R)}$. Similarly we can define $\boldsymbol{\beta}_R$ and $\boldsymbol{\beta}_{R\perp}$. Thus,*

$$y = \boldsymbol{x}^\top \boldsymbol{\beta} + \varepsilon = \boldsymbol{x}_R^\top \boldsymbol{\beta}_R + \boldsymbol{x}_{R\perp}^\top \boldsymbol{\beta}_{R\perp} + \varepsilon \tag{B.3}$$

*We can treat $\varepsilon_R = \boldsymbol{x}_{R\perp}^\top \boldsymbol{\beta}_{R\perp} + \varepsilon$ as the new noise, and try to solve for $\boldsymbol{\beta}_R$. Then define $\boldsymbol{\Sigma}_{T,R\perp}$ as the matrix containing the same eigenvectors as $\boldsymbol{\Sigma}_T$ and the top $R$ eigenvalues are zeroed out, our noise variance becomes $\sigma_R^2 = \sigma^2 + \boldsymbol{E}(\|\boldsymbol{x}_{R\perp}\|^2 \|\boldsymbol{\beta}_{R\perp}\|^2) = \sigma^2 + \mathrm{tr}(\tilde{\boldsymbol{\Sigma}}_T) - \mathrm{tr}(\tilde{\boldsymbol{\Sigma}}_T^R)$ in our algorithm. If we are still in overparameterized regime, namely $R > n_2$, then we define optimal representation on top of it.*

*In summary, the $R$-SVD truncation reduces the search space of $\boldsymbol{\Lambda}$ into $R$ dimensional space, where the covariance of the noise in $\boldsymbol{y}$ increases from $\sigma^2 \boldsymbol{I}$ to $\sigma_R^2 \boldsymbol{I}$.*

## B.2 Distributional characterization of least norm solution

In this part, for simplicity of discussion, we focus on the $R$ dimensional space while omitting the projection step, and the equivalence of a diagonal eigen-weighting matrix $\boldsymbol{\Lambda}_R \in \mathbb{R}^{R \times R}$ and $\boldsymbol{\theta} \in \mathbb{R}^R$ in Thm. 1. Here, we assume a truncated feature matrix $\tilde{\boldsymbol{X}} \in \mathbb{R}^{n \times R}$ where the feature is projected into an $R$ dimensional space.

Define $\tilde{\boldsymbol{X}} \in \mathbb{R}^{n \times R}, \tilde{\boldsymbol{y}} \in \mathbb{R}^n$. We study the following least norm solution of the least squares problem

$$\hat{\boldsymbol{\beta}} = \arg\min_{\boldsymbol{\beta}'} \|\boldsymbol{\beta}'\|, \quad \text{s.t.,} \ \tilde{\boldsymbol{X}} \boldsymbol{\beta}' = \tilde{\boldsymbol{y}} \tag{B.4}$$

**Assumption 5** *Assume the rows of $\tilde{\boldsymbol{X}}$ are independently drawn from $\mathcal{N}(0, \tilde{\boldsymbol{\Sigma}}_{\boldsymbol{X}})$. We focus on a double asymptotic regime where $R, n \to \infty$ at fixed overparameterization ratio $\kappa := R/n > 0$.*

**Assumption 6** *The covariance matrix $\tilde{\boldsymbol{\Sigma}}_{\boldsymbol{X}}$ is diagonal and there exist constants $\Sigma_{\min}, \Sigma_{\max} \in (0, \infty)$ such that: $0 \prec \Sigma_{\min} \boldsymbol{I} \preceq \tilde{\boldsymbol{\Sigma}}_{\boldsymbol{X}} \preceq \Sigma_{\max} \boldsymbol{I}$.*

**Assumption 7** *The joint empirical distribution of $\{(\lambda_i(\tilde{\boldsymbol{\Sigma}}_{\boldsymbol{X}}), \boldsymbol{\beta}_i)\}_{i \in [R]}$ converges in Wasserstein-$k$ distance to a probability distribution $\mu$ on $\mathbb{R}_{>0} \times \mathbb{R}$ for some $T \geq 4$. That is $\frac{1}{R} \sum_{i \in [R]} \delta_{(\lambda_i(\tilde{\boldsymbol{\Sigma}}_{\boldsymbol{X}}), \boldsymbol{\beta}_i)} \overset{W_k}{\Longrightarrow} \mu$.*

**Definition 5 (Asymptotic distribution characterization – Overparameterized regime)** *[36] Let random variables $(\Sigma, B) \sim \mu$ (where $\mu$ is defined in Assumption 7) and fix $\kappa > 1$. Define parameter $\xi$ as the unique positive solution to the following equation*

$$\mathbb{E}_\mu \left[ \left( 1 + (\xi \cdot \Sigma)^{-1} \right)^{-1} \right] = \kappa^{-1}. \tag{B.5}$$

*Define the positive parameter $\gamma$ as follows:*

$$\gamma := \left( \sigma^2 + \mathbb{E}_\mu \left[ \frac{B^2 \Sigma}{(1 + \xi \Sigma)^2} \right] \right) \bigg/ \left( 1 - \kappa \, \mathbb{E}_\mu \left[ \frac{1}{(1 + (\xi \Sigma)^{-1})^2} \right] \right). \tag{B.6}$$

With these and $H \sim \mathcal{N}(0, 1)$, define the random variable

$$X_{\kappa,\sigma^2}(\Sigma, B, H) := \left(1 - \frac{1}{1+\xi\Sigma}\right)B + \sqrt{\kappa}\frac{\sqrt{\gamma}\,\Sigma^{-1/2}}{1+(\xi\Sigma)^{-1}}H, \tag{B.7}$$

and let $\Pi_{\kappa,\sigma^2}$ be its distribution.

**Theorem 4 (Asymptotic distribution characterization – Overparameterized linear Gaussian problem)**
*[36] Fix $\kappa > 1$ and suppose Assumptions 6 and 7 hold. Let*

$$\frac{1}{R}\sum_{i=1}^{R}\delta_{\sqrt{R}\hat{\beta}_i, \sqrt{R}\beta_i, \tilde{\Sigma}_{\boldsymbol{X}_{i,i}}}$$

*be the joint empirical distribution of $(\sqrt{R}\hat{\boldsymbol{\beta}}, \sqrt{R}\boldsymbol{\beta}, \tilde{\Sigma}_{\boldsymbol{X}})$ and it converges to a fixed distribution as dimension grows. Let $f : \mathbb{R}^3 \to \mathbb{R}$ be a function in $\mathrm{PL}(2)$. We have that*

$$\frac{1}{R}\sum_{i=1}^{R}f(\sqrt{R}\hat{\beta}_i, \sqrt{R}\beta_i, \tilde{\Sigma}_{\boldsymbol{X}_{i,i}}) \xrightarrow{P} \mathbb{E}\left[f(X_{\kappa,\sigma^2}, B, \Sigma)\right]. \tag{B.8}$$

*In particular, the risk is given by*

$$risk(\hat{\boldsymbol{\beta}}_n) \xrightarrow{P} \mathbb{E}[\Sigma(B - X_{\kappa,\sigma^2})] + \sigma_R^2 \tag{B.9}$$

$$= \mathbb{E}[\frac{\Sigma}{(1+\xi\Sigma)^2}B^2 + \frac{\kappa\gamma}{(1+(\xi\Sigma)^{-1})^2}] + \sigma_R^2. \tag{B.10}$$

## B.3   Finding Optimal Representation

Now, for simplicity (and actually without losing generality) assume $\tilde{\Sigma}_{\boldsymbol{X}} = \boldsymbol{I}$. This means that empirical measure of $\boldsymbol{\Sigma}_F$ trivially converges to $\Sigma = 1$. With the representation $\boldsymbol{\Lambda}^*$ with asymptotic distribution $\Lambda$, the ML problem has the following mapping

$$\boldsymbol{\beta} \to \boldsymbol{\Lambda}_R^{-1}\boldsymbol{\beta} \quad \text{and} \quad \tilde{\boldsymbol{\Sigma}}_{\boldsymbol{X}} \to \boldsymbol{\Lambda}_R\tilde{\boldsymbol{\Sigma}}_{\boldsymbol{X}}\boldsymbol{\Lambda}_R.$$

This means the empirical measure converges to the following mapped distributions

$$B \to \bar{B} = \Lambda^{-1}B \quad \text{and} \quad \Sigma = 1 \to \bar{\Sigma} = \Lambda^2\Sigma = \Lambda^2.$$

**Our question:** Craft the optimal distribution $\Lambda$ to minimize the representation learning risk. Specifically, for a given $(B, \Lambda)$ pair, we know from the theorem above that

$$\text{risk}^{\boldsymbol{\Lambda}_R}(\hat{\boldsymbol{\beta}}_n) \xrightarrow{P} \mathbb{E}[\frac{\bar{\Sigma}}{(1+\xi\bar{\Sigma})^2}\bar{B}^2 + \frac{\kappa\gamma}{(1+(\xi\bar{\Sigma})^{-1})^2}] + \sigma_R^2 \tag{B.11}$$

$$= \mathbb{E}[\frac{B^2}{(1+\xi\Lambda^2)^2} + \frac{\kappa\gamma}{(1+(\xi\Lambda^2)^{-1})^2}] + \sigma_R^2. \tag{B.12}$$

Thus, the optimal weighting strategy (asymptotically) is given by the distribution

$$\Lambda^* = \arg\min_\Lambda \mathbb{E}[\frac{B^2}{(1+\xi\Lambda^2)^2} + \frac{\kappa\gamma}{(1+(\xi\Lambda^2)^{-1})^2}],$$

where $\gamma, \xi$ are strictly positive scalars that are also functions of $\Lambda$.

## B.4   Non-asymptotic Analysis (for simpler insights)

We apply the discussion iin Sec. B.2 non-asymptotically in few-shot learning. Remember we define $\boldsymbol{X} \in \mathbb{R}^{n_2 \times R}, \boldsymbol{y} \in \mathbb{R}^{n_2}$, each row of $\boldsymbol{X}$ is independently drawn from $\mathcal{N}(0, \boldsymbol{\Sigma}_F)$. We study the following least norm solution of the least squares problem

$$\hat{\boldsymbol{\beta}} = \arg\min_{\boldsymbol{\beta}'} \|\boldsymbol{\beta}'\|, \quad \text{s.t.,} \ \boldsymbol{X}\boldsymbol{\beta}' = \boldsymbol{y}. \tag{B.13}$$

**Definition 6 (Non-asymptotic distribution characterization)** *Set $\kappa = R/n_2 > 1$. Given $\sigma_R > 0$, covariance $\boldsymbol{\Sigma}_F$ and latent vector $\boldsymbol{\beta}$ and define the unique non-negative terms $\xi, \gamma, \boldsymbol{z} \in \mathbb{R}^R$ and $\boldsymbol{\phi} \in \mathbb{R}^R$ as follows:*

$$\xi > 0 \quad \textit{is the solution of} \quad \kappa^{-1} = R^{-1} \sum_{i=1}^{R} \left(1 + (\xi \boldsymbol{\Sigma}_{F,i})^{-1}\right)^{-1},$$

$$\gamma = \frac{\sigma_R^2 + \frac{1}{R} \sum_{i=1}^{R} \frac{\boldsymbol{\Sigma}_{F,i} \boldsymbol{\beta}_i^2}{(1 + \xi \boldsymbol{\Sigma}_F)^2}}{1 - \frac{\kappa}{R} \sum_{i=1}^{R} \left(1 + (\xi \boldsymbol{\Sigma}_{F,i})^{-1}\right)^{-2}}.$$

*Let $\boldsymbol{h} \sim \mathcal{N}(0, \mathrm{I}/R)$. The non-asymptotic distributional prediction is given by the following random vector*

$$\hat{\boldsymbol{\beta}}(\boldsymbol{\Sigma}_F) = \frac{1}{1 + (\xi \boldsymbol{\Sigma}_F)^{-1}} \odot \boldsymbol{\beta} + \frac{\sqrt{\kappa \gamma} \boldsymbol{\Sigma}_F^{-1/2}}{1 + (\xi \boldsymbol{\Sigma}_F)^{-1}} \odot \boldsymbol{h}.$$

Note that, the above formulas can be slightly simplified to have a cleaner look by introducing an additional variable $\boldsymbol{z} = \frac{1}{1 + (\xi \boldsymbol{\Sigma}_F)^{-1}}$.

Also note that, the terms in the non-asymptotic distribution characterization and asymptotic distribution characterization have one to one correspondence. Non-asymptotic distribution characterization is essentially a discretized version of asymptotic DC where instead of expectations (which is integral over pdf) we have summations.

Now, we can use this distribution to predict the test risk by using Def. 6 in the risk expression.

Going back to representation question, without losing generality, assume $\boldsymbol{\Sigma}_F = \boldsymbol{I}$ and let us find optimal $\boldsymbol{\Lambda}_R$. Then

$$\hat{\boldsymbol{\beta}} = \boldsymbol{\Lambda}_R \left[ \frac{1}{1 + (\xi \boldsymbol{\Lambda}_R^2)^{-1}} \odot \boldsymbol{\Lambda}_R^{-1} \boldsymbol{\beta} + \frac{\sqrt{\kappa \gamma} \boldsymbol{\Lambda}_R^{-1}}{1 + (\xi \boldsymbol{\Lambda}_R^2)^{-1}} \odot \boldsymbol{h} \right].$$

The risk is given by (using $\boldsymbol{h} \sim \mathcal{N}(0, \boldsymbol{I}_p)$)

$$\mathrm{risk}^{\boldsymbol{\Lambda}_R}(\hat{\boldsymbol{\beta}}_n) - \sigma_R^2 = \mathbb{E}[(\hat{\boldsymbol{\beta}} - \boldsymbol{\beta})^\top \boldsymbol{\Sigma}_F (\hat{\boldsymbol{\beta}} - \boldsymbol{\beta})] \tag{B.14}$$

$$= \sum_{i=1}^{R} \frac{\boldsymbol{\Sigma}_{T,i}}{(1 + \xi (\boldsymbol{\Lambda}_{R,i})^2)^2} + \sum_{i=1}^{R} \frac{\kappa \gamma}{(1 + (\xi (\boldsymbol{\Lambda}_{R,i})^2)^{-1})^2}. \tag{B.15}$$

Here, note that $\xi$ is function of $\boldsymbol{\Lambda}^*$ and $\gamma$ is function of $\boldsymbol{\beta}, \boldsymbol{\Lambda}^*$. If we don't know $\boldsymbol{\Sigma}_T$, we use the estimation from representation learning $\hat{\boldsymbol{\Sigma}}_T$ instead.

To find the optimal representation, we will solve the following optimization problem that minimizes the risk.

$$\min_{\boldsymbol{\Lambda}^*} \quad \sum_{i=1}^{R} \frac{\boldsymbol{\beta}_i^2}{(1 + \xi (\boldsymbol{\Lambda}_{R,i})^2)^2} + \sum_{i=1}^{R} \frac{\kappa \gamma}{(1 + (\xi (\boldsymbol{\Lambda}_{R,i})^2)^{-1})^2}$$

$$\text{s.t.} \quad \kappa^{-1} = \frac{1}{R} \sum_{i=1}^{R} (1 + (\xi (\boldsymbol{\Lambda}_{R,i})^2)^{-1})^{-1} \tag{B.16}$$

$$\gamma = \frac{\sigma_R^2 + \sum_{i=1}^{R} \frac{\boldsymbol{\beta}_i^2}{(1 + \xi (\boldsymbol{\Lambda}_{R,i})^2)^2}}{1 - \frac{\kappa}{R} \sum_{i=1}^{R} (1 + (\xi (\boldsymbol{\Lambda}_{R,i})^2)^{-1})^{-2}}.$$

So we plug in the expression of $\gamma$ and get

$$\kappa \gamma = \frac{\sigma_R^2 + \frac{1}{R} \sum_{i=1}^{R} \frac{\boldsymbol{\beta}_i^2}{(1 + \xi (\boldsymbol{\Lambda}_{R,i})^2)^2}}{\kappa^{-1} - \frac{1}{R} \sum_{i=1}^{R} (1 + (\xi (\boldsymbol{\Lambda}_{R,i})^2)^{-1})^{-2}} = \frac{R \sigma_R^2 + \sum_{i=1}^{R} \frac{\boldsymbol{\beta}_i^2}{(1 + \xi (\boldsymbol{\Lambda}_{R,i})^2)^2}}{\sum \frac{\xi (\boldsymbol{\Lambda}_{R,i})^2}{(1 + \xi (\boldsymbol{\Lambda}_{R,i})^2)^2}}. \tag{B.17}$$

Let $\boldsymbol{\theta}_i = \frac{\xi(\boldsymbol{\Lambda}_{R,i})^2}{1+\xi(\boldsymbol{\Lambda}_{R,i})^2}$, then the objective function becomes

$$\sum_{i=1}^{R}\boldsymbol{\Sigma}_{T,i}(1-\boldsymbol{\theta}_i)^2+(\sum_{i=1}^{R}\boldsymbol{\theta}_i^2)\frac{R\sigma_R^2 + \sum \boldsymbol{\Sigma}_{T,i}(1-\boldsymbol{\theta}_i)^2}{\sum_{i=1}^{R}\boldsymbol{\theta}_i(1-\boldsymbol{\theta}_i)} = \frac{n_2(\sum_{i=1}^{R}\boldsymbol{\Sigma}_{T,i}(1-\boldsymbol{\theta}_i)^2) + R\sigma_R^2(\sum_{i=1}^{R}\boldsymbol{\theta}_i^2)}{n_2 - \sum_{i=1}^{R}\boldsymbol{\theta}_i^2}$$

such that $0 \leq \boldsymbol{\theta}_i < 1$ and $\sum_{i=1}^{R}\boldsymbol{\theta}_i = \frac{R}{\kappa} = n_2$. This quantity is same as the objective (B.16). We divide this quantity by $d$ to get the risk function, which is same as the definition of $f$ in (3.3).

## B.5 Solving the optimization problem.

Here, we propose the algorithm for minimizing $f(\boldsymbol{\theta})$. We explore the KKT condition for its optimality.

The objective function is

$$f(\boldsymbol{\theta}) = \sum_{i=1}^{R}\boldsymbol{\Sigma}_{T,i}(1 - \boldsymbol{\theta}_i)^2 + (\sum_{i=1}^{R}\boldsymbol{\theta}_i^2)\frac{R\sigma_R^2 + \sum \boldsymbol{\Sigma}_{T,i}(1 - \boldsymbol{\theta}_i)^2}{\sum_{i=1}^{R}\boldsymbol{\theta}_i(1 - \boldsymbol{\theta}_i)}. \tag{B.18}$$

**Lemma 1** *Let $C, S, V \in \mathbb{R}$. Define*

$$\phi(\boldsymbol{\Sigma}_{T,i}; C, V, S) := \frac{Cp(R - n_2 - S)^2}{2n_2(V + R\sigma_R^2 + (R - n_2 - S)\boldsymbol{\Sigma}_{T,i}^2)}$$

*and we find the root of the following equations:*

$$\sum_{i=1}^{R}\phi(\boldsymbol{\Sigma}_{T,i}; C, V, S) = R - n_2,$$

$$\sum_{i=1}^{R}\phi^2(\boldsymbol{\Sigma}_{T,i}; C, V, S) = S - (2n_2 - R),$$

$$\sum_{i=1}^{R}\boldsymbol{\Sigma}_{T,i}\phi^2(\boldsymbol{\Sigma}_{T,i}; C, V, S) = V.$$

*Let $\boldsymbol{\theta}_i = 1 - \phi(\boldsymbol{\Sigma}_{T,i}; C^*, V^*, S^*)$ where $C^*, V^*, S^*$ are the roots, then*

$$\boldsymbol{\theta} = \arg\min_{\boldsymbol{\theta}'} f(\boldsymbol{\theta}'), \quad s.t., \ 0 \leq \boldsymbol{\theta}' < 1, \ \sum_{i=1}^{R}\boldsymbol{\theta}'_i = n_2.$$

**Proof** Define $s = \sum_{i=1}^{R}\boldsymbol{\theta}_i^2$, $\phi_i = 1 - \boldsymbol{\theta}_i$. Define $Q = \frac{1}{R}\sum_{i=1}^{R}\boldsymbol{\Sigma}_{T,i}\phi_i^2$. Then

$$\begin{aligned}
f(\phi) &= \sum_{i=1}^{R}\boldsymbol{\Sigma}_{T,i}\phi_i^2 + \frac{s}{n_2 - s}(R\sigma_R^2 + \sum_{i=1}^{R}\boldsymbol{\Sigma}_{T,i}\phi_i^2) \\
&= R(Q + \frac{s}{n_2 - s}(\sigma_R^2 + Q)) \\
&= \frac{Rn_2}{R - n_2 - \sum_{i=1}^{R}\phi_i^2}(Q + \sigma_R^2).
\end{aligned}$$

The last line uses

$$s = \sum_{i=1}^{R}(1 - \phi^2) = R - 2\sum_{i=1}^{R}\phi_i + \sum_{i=1}^{R}\phi_i^2 = R - 2(R - n_2) + \sum_{i=1}^{R}\phi_i^2 = 2n_2 - R + \sum_{i=1}^{R}\phi_i^2.$$

Now define $\sum_{i=1}^{R}\phi_i^2 = S$, and we compute the gradient of $f$, we have

$$\frac{df}{R\phi_i} = \left(2n_2(\sum_{j=1}^{R}\boldsymbol{\Sigma}_{Tj}\phi_j^2 + (R - n_2 - s)\boldsymbol{\Sigma}_{T,i}) + 2Rn_2\sigma_R^2\right)\phi_i.$$

Suppose $0 < \phi_i < 1$, then we need $\frac{df}{R\phi_i}$ equal to each other for all $i$. Suppose $\frac{df}{R\phi_i} = C$, and denote $\sum \boldsymbol{\Sigma}_{Tj}\phi_j^2 = V$, we can solve for $\phi_i$ from $\frac{df}{R\phi_i} = C$ as

$$\phi_i = \frac{Cd(R - n_2 - S)^2}{2n_2(V + R\sigma_R^2 + (R - n_2 - S)\boldsymbol{\Sigma}_{T,i}{}^2)} := \phi(\boldsymbol{\Sigma}_{T,i}; C, V, S). \tag{B.19}$$

We define the function $\phi(\boldsymbol{\Sigma}_{T,i}; C, V, S)$ as above, and use the fact that

$$\sum_{i=1}^{R} \phi(\boldsymbol{\Sigma}_{T,i}; C, V, S) = R - n_2,$$

$$\sum_{i=1}^{R} \phi^2(\boldsymbol{\Sigma}_{T,i}; C, V, S) = S - (2n_2 - R),$$

$$\sum_{i=1}^{R} \boldsymbol{\Sigma}_{T,i}\phi^2(\boldsymbol{\Sigma}_{T,i}; C, V, S) = V.$$

We can solve[9] $C, V, S$ and retrieve $\phi_i$ by (B.19). $\boldsymbol{\theta}_i = 1 - \phi_i$. ∎

# C  Analysis of MoM estimators

## C.1  Covariance estimator

We will first present the estimation error of the feature covariance $\boldsymbol{\Sigma}_F$, which is not covered in the main paper due to limitation of space. Note that if $\boldsymbol{\Sigma}_F$ is fully aligned with $\boldsymbol{\Sigma}_T$, e.g., $\boldsymbol{\Sigma}_F = \boldsymbol{\Sigma}_T$, then estimating $\boldsymbol{\Sigma}_F$ is enough for getting optimal representation, and we will show it has lower sample complexity and error compared to estimating canonical covariance $\tilde{\boldsymbol{\Sigma}}_T$. That is a naive case, if it does not work, this intermediate result will help in our latter proof.

We will use the following Bernstein type concentration lemma, generalized from [37, Lemma 29]:

**Lemma 2** *Let $\boldsymbol{Z} \in \mathbb{R}^{n_1 \times n_2}$. Choose $T_0, \sigma^2$ such that*

1. *$\boldsymbol{P}(\|\boldsymbol{Z}\| \geq C_0 T_0 + t) \leq \exp(-c\sqrt{t/T_0})$.*
2. *$\|\boldsymbol{E}(\boldsymbol{Z}\boldsymbol{Z}^\top)\|, \|\boldsymbol{E}(\boldsymbol{Z}^\top\boldsymbol{Z})\| \leq \sigma^2$.*

*Then with probability at least $1 - (nT_0)^{-c}$, $c > 10$,*

$$\|\frac{1}{n}\sum_{i=1}^{n} \boldsymbol{Z}_i - \boldsymbol{E}(\boldsymbol{Z}_i)\| \lesssim \log(nT_0)\left(\frac{T_0 \log(nT_0)}{n} + \frac{\sigma}{\sqrt{n}}\right).$$

**Proof** Define $K = \log^2(C_K nT_0)$ for $C_K > 0$, $\boldsymbol{Z}' = \boldsymbol{Z}\mathbf{1}(\|\boldsymbol{Z}\| \leq KT_0)$, then

$$\|\boldsymbol{E}(\boldsymbol{Z} - \boldsymbol{Z}')\| \leq \int_{KT_0}^{\infty} \exp(-c\sqrt{t/T_0})dt \lesssim (1 + \sqrt{K})\exp(-c\sqrt{K})T_0$$
$$\lesssim (1 + \log(C_K nT_0))(nT_0)^{-C}.$$

We can choose $C_K$ large enough so that $C > 10$. We will use [37, Lemma 29]. Set $R = \log^2(C_K nT_0)T_0 + C_0 T_0$, $\Delta = (1 + \log(C_K nT_0))(nT_0)^{-C}$, $t = C_t \log(nT_0)(\frac{T_0 \log(nT_0)}{n} + \frac{\sigma}{\sqrt{n}})$ for some $C_t > 0$, plugging in the last inequality of [37, Lemma 29], the LHS is smaller than $(nT_0)^{-c}$ for some $c$. We can also check $\boldsymbol{P}(\|\boldsymbol{Z}\| \geq R) \leq (nT_0)^{-c}$ for some $c$, thus we prove the lemma. ∎

---

[9]For the root of 3-dim problem, the worst case we can grid the space and search with time complexity $\mathcal{O}(\varepsilon^{-3})$.

**Feature Covariance.** We can directly estimate the covariance of features by

$$\hat{\boldsymbol{\Sigma}}_F = \frac{1}{N} \sum_{j=1}^{n_1} \sum_{i=1}^{T} \boldsymbol{x}_{ij} \boldsymbol{x}_{ij}^\top, \tag{C.1}$$

The mean of this estimator is $\boldsymbol{\Sigma}_F$ and we can estimate the top $r$ eigenvector of $\boldsymbol{\Sigma}_F$ with $\tilde{\mathcal{O}}(r)$ samples.

As we have defined in Phase 1, features $\boldsymbol{x}_{ij}$ are generated from $\mathcal{N}(0, \boldsymbol{\Sigma}_F)$. We aim to estimate the covariance $\boldsymbol{\Sigma}_F$. Although there are different kinds of algorithms, such as maximum likelihood estimator [1], to be consistent with the algorithms in the latter sections, we study the sample covariance matrix defined by (C.1).

**Lemma 3** *Suppose $\boldsymbol{x}_i$, $i = 1, ..., N$ are generated independently from $\mathcal{N}(0, \boldsymbol{\Sigma}_F)$. We estimate* (C.1)*, then when $N \gtrsim r_F$, with probability $1 - \mathcal{O}((N\mathbf{tr}(\boldsymbol{\Sigma}_F))^{-C})$,*

$$\|\hat{\boldsymbol{\Sigma}}_F - \boldsymbol{\Sigma}_F\| \lesssim \sqrt{\frac{\|\boldsymbol{\Sigma}_F\|\mathbf{tr}(\boldsymbol{\Sigma}_F)}{N}}.$$

*Denote the span of top $s_F$ eigenvectors of $\boldsymbol{\Sigma}_F$ as $\boldsymbol{W}$ and the span of top $s_F$ eigenvectors of $\hat{\boldsymbol{\Sigma}}_F$ as $\hat{\boldsymbol{W}}$. Let $\delta_\lambda = \lambda_{s_F}(\boldsymbol{\Sigma}_F) - \lambda_{s_F+1}(\boldsymbol{\Sigma}_F)$. Then if $N \gtrsim \frac{\|\boldsymbol{\Sigma}_F\|\mathbf{tr}(\boldsymbol{\Sigma}_F)}{\delta_\lambda^2}$, we have*

$$\sin(\angle \boldsymbol{W}, \hat{\boldsymbol{W}}) \lesssim \sqrt{\frac{\|\boldsymbol{\Sigma}_F\|\mathbf{tr}(\boldsymbol{\Sigma}_F)}{N\delta_\lambda^2}}$$

**Example 2** *When $\boldsymbol{\Sigma}_F = diag(\boldsymbol{I}_{s_F}, 0)$, we have $\sin(\angle \boldsymbol{W}, \hat{\boldsymbol{W}}) \lesssim \sqrt{\frac{s_F}{N}}$.*

Lemma 3 gives the quality of the estimation of the covariance of features $\boldsymbol{x}$. When the condition number of the matrix $\boldsymbol{\Sigma}_F$ is close to 1, we need $N \gtrsim d$ to get an estimation with error $\mathcal{O}(1)$. However, when the matrix $\boldsymbol{\Sigma}_F$ is close to rank $r_F$, the amount of samples to achieve the same error is smaller, and we can use $N \gtrsim r_F$ samples to get $\mathcal{O}(1)$ estimation error.

We will use Bernstein type concentration results to bound its error, and a similar technique will be used for $\hat{\boldsymbol{M}}$ in the next sections.

**Proof** First we observe that, the features $\boldsymbol{x}_{ij}$ among different tasks are generated i.i.d. from $\mathcal{N}(0, \boldsymbol{\Sigma}_F)$. So we can rewrite (C.1) as

$$\hat{\boldsymbol{\Sigma}}_F = \frac{1}{N} \sum_{i=1}^{N} \boldsymbol{x}_i \boldsymbol{x}_i^\top \tag{C.2}$$

where $\boldsymbol{x}_i \sim \mathcal{N}(0, \boldsymbol{\Sigma}_F)$. The error of $\hat{\boldsymbol{\Sigma}}_F$ depends on $N$ regardless of $T$ and $n_1$ respectively.

First, we know by concentration inequality

$$\boldsymbol{P}(\|\boldsymbol{x}\boldsymbol{x}^\top\| - \mathbf{tr}(\boldsymbol{\Sigma}_F) \geq t) = \boldsymbol{P}(\|\boldsymbol{x}\|^2 - \mathbf{tr}(\boldsymbol{\Sigma}_F) \geq t) \leq \exp(-c\min\{\frac{t^2}{\mathbf{tr}(\boldsymbol{\Sigma}_F^2)}, \frac{t}{\|\boldsymbol{\Sigma}_F\|}\}). \tag{C.3}$$

We will use the fact $\sqrt{\mathbf{tr}(\boldsymbol{\Sigma}_F^2)} \leq \mathbf{tr}(\boldsymbol{\Sigma}_F)$. Define $K = C_0 \log(N\mathbf{tr}(\boldsymbol{\Sigma}_F))\mathbf{tr}(\boldsymbol{\Sigma}_F)$, $\boldsymbol{Z} = \boldsymbol{x}\boldsymbol{x}^\top$, $\boldsymbol{Z}' = \boldsymbol{Z} \cdot \boldsymbol{1}\{\|\boldsymbol{Z}\| \leq K\}$ where $\boldsymbol{1}$ means indicator function ($\boldsymbol{1}(\text{True}) = 1, \boldsymbol{1}(\text{False}) = 0$), for some positive number $C_0$. Then

$$\begin{aligned}
\|\boldsymbol{E}(\boldsymbol{Z} - \boldsymbol{Z}')\| &\leq \int_{t=K}^{\infty} (\exp(-c\frac{t^2}{\mathbf{tr}^2(\boldsymbol{\Sigma}_F)}) + \exp(-c\frac{t}{\|\boldsymbol{\Sigma}_F\|}))dt \\
&\leq \int_{t=K}^{\infty} (\exp(-c\frac{t}{\mathbf{tr}(\boldsymbol{\Sigma}_F)}) + \exp(-c\frac{t}{\|\boldsymbol{\Sigma}_F\|}))dt \\
&\leq 2\frac{\mathbf{tr}(\boldsymbol{\Sigma}_F)}{c} \exp(-c\frac{K}{\mathbf{tr}(\boldsymbol{\Sigma}_F)}) \\
&\leq \frac{\sqrt{K\mathbf{tr}^2(\boldsymbol{\Sigma}_F)}}{c} \exp(-\frac{cK}{\mathbf{tr}(\boldsymbol{\Sigma}_F)}) \\
&\lesssim (N\mathbf{tr}(\boldsymbol{\Sigma}_F))^{-C}
\end{aligned}$$

where $C \geq C_0 - 3/2$. Then we compute $(\boldsymbol{xx}^\top)^2 = \|\boldsymbol{x}\|^2 \boldsymbol{xx}^\top$. Let $\boldsymbol{\Sigma}_F$ be diagonal (the proof is invariant from the basis. In other words, if $\boldsymbol{\Sigma}_F$ is not diagonal, then we can make the eigenvectors of $\boldsymbol{\Sigma}_F$ as basis and the proof applies). Then

$$\boldsymbol{E}(\|\boldsymbol{x}\|^2 \boldsymbol{xx}^\top)_{ij} = \begin{cases} \boldsymbol{\Sigma}_{Fii}(\mathbf{tr}(\boldsymbol{\Sigma}_F) + 2\boldsymbol{\Sigma}_{Fii}), & i = j, \\ 0, & i \neq j. \end{cases} \tag{C.4}$$

So $\|\boldsymbol{E}(\|\boldsymbol{x}\|^2 \boldsymbol{xx}^\top)\| \leq \|\boldsymbol{\Sigma}_F\|(\mathbf{tr}(\boldsymbol{\Sigma}_F) + 2\|\boldsymbol{\Sigma}_F\|) \approx \|\boldsymbol{\Sigma}_F\|\mathbf{tr}(\boldsymbol{\Sigma}_F)$. $\approx$ means $\gtrsim$ and $\lesssim$.

Using Lemma 2, with (C.3) and the inequality above, we get that with probability $1 - \mathcal{O}((N\mathbf{tr}(\boldsymbol{\Sigma}_F))^{-C})$,

$$\|\hat{\boldsymbol{\Sigma}}_F - \boldsymbol{\Sigma}_F\| \lesssim \log(N\mathbf{tr}(\boldsymbol{\Sigma}_F)) \left( \frac{\log(N\mathbf{tr}(\boldsymbol{\Sigma}_F))\mathbf{tr}(\boldsymbol{\Sigma}_F)}{N} + \sqrt{\frac{\|\boldsymbol{\Sigma}_F\|\mathbf{tr}(\boldsymbol{\Sigma}_F)}{N}} \right). \tag{C.5}$$

If the number above is smaller than $\lambda_r - \lambda_{r+1}$, we have that

$$N \gtrsim \frac{\|\boldsymbol{\Sigma}_F\|\mathbf{tr}(\boldsymbol{\Sigma}_F)}{(\lambda_r - \lambda_{r+1})^2} \tag{C.6}$$

which is $\mathcal{O}(r)$ if condition number is 1.

The bound of the angle of top $R$ eigenvector subspace is a direct application of the following lemma.

**Lemma 4** *[14] Let $\boldsymbol{A}$ be a square matrix. Let $\hat{\boldsymbol{W}}$, $\boldsymbol{W}$ denote the span of top $r$ singular vectors of $\hat{A}$ and $\boldsymbol{A}$. Suppose $\|\hat{\boldsymbol{A}} - \boldsymbol{A}\| \leq \Delta$, and $\sigma_r(\boldsymbol{A}) - \sigma_{r+1}(\boldsymbol{A}) \geq \Delta$, then*

$$\sin(\angle \boldsymbol{W}, \hat{\boldsymbol{W}}) \leq \frac{\Delta}{\sigma_r(\boldsymbol{A}) - \sigma_{r+1}(\boldsymbol{A}) - \Delta}.$$

So that the error of principle subspace recovery of feature covariance is upper bounded by $\frac{\|\hat{\boldsymbol{\Sigma}}_F - \boldsymbol{\Sigma}_F\|}{\sigma_r(\boldsymbol{\Sigma}_F) - \sigma_{r+1}(\boldsymbol{\Sigma}_F) - \|\hat{\boldsymbol{\Sigma}}_F - \boldsymbol{\Sigma}_F\|}$, where $\|\hat{\boldsymbol{\Sigma}}_F - \boldsymbol{\Sigma}_F\|$ is calculated in (C.5). ∎

## C.2 Method of moment

This section contains three parts. We first bound the norm of task vectors. Then we analyze the second result of Thm. 2, where $n_1$ is lower bounded by effective rank. Last we prove the first result of Thm. 2 which is a generalization of [37].

### C.2.1 Property of task vectors

We first study the property of the tasks $\boldsymbol{\beta}_1, ..., \boldsymbol{\beta}_T$. We know that, for any $\boldsymbol{\beta} \sim \mathcal{N}(0, \boldsymbol{\Sigma}_T)$,

$$\boldsymbol{P}(\|\boldsymbol{\beta}\|^2 - \mathbf{tr}(\boldsymbol{\Sigma}_T) \geq t) \leq \exp(-c \min\{\frac{t^2}{\mathbf{tr}(\boldsymbol{\Sigma}_T^2)}, \frac{t}{\|\boldsymbol{\Sigma}_T\|}\}).$$

So that with probability at least $1 - \delta$, we have

$$\|\boldsymbol{\beta}_i\|^2 \lesssim \mathbf{tr}(\boldsymbol{\Sigma}_T) + \sqrt{(\log(1/\delta) + \log(T))\mathbf{tr}(\boldsymbol{\Sigma}_T^2)} + (\log(1/\delta) + \log(T))\|\boldsymbol{\Sigma}_T\|$$

$$\lesssim \mathbf{tr}(\boldsymbol{\Sigma}_T) + \log(T/\delta)\sqrt{\mathbf{tr}(\boldsymbol{\Sigma}_T^2)} \lesssim \mathbf{tr}(\boldsymbol{\Sigma}_T)\log(T/\delta), \ \forall i = 1, ..., T. \tag{C.7}$$

With similar technique we know that with probability at least $1 - \delta$,

$$\|\boldsymbol{\Sigma}_F \boldsymbol{\beta}_i\|^2 \lesssim \mathbf{tr}(\boldsymbol{\Sigma}_F \boldsymbol{\Sigma}_T \boldsymbol{\Sigma}_F) + \log(T/\delta)\sqrt{\mathbf{tr}((\boldsymbol{\Sigma}_F \boldsymbol{\Sigma}_T \boldsymbol{\Sigma}_F)^2)}, \ \forall i = 1, ..., T. \tag{C.8}$$

$$\|\boldsymbol{\Sigma}_F^{1/2} \boldsymbol{\beta}_i\|^2 \lesssim \mathbf{tr}(\boldsymbol{\Sigma}_F^{1/2} \boldsymbol{\Sigma}_T \boldsymbol{\Sigma}_F^{1/2}) + \log(T/\delta)\sqrt{\mathbf{tr}((\boldsymbol{\Sigma}_F^{1/2} \boldsymbol{\Sigma}_T \boldsymbol{\Sigma}_F^{1/2})^2)}, \ \forall i = 1, ..., T. \tag{C.9}$$

We will use $\delta = T^{-c}$ for some constant $c$ so that $\log(T/\delta) = (c+1)\log(T) \approx \log(T)$. Later, we will use the norm bounds of above quantities which happen with probability at least $1 - T^{-c}$.

### C.2.2 Estimating with fewer samples when each task contains enough samples

In this part we will prove Theorem 6, which is the second case of Theorem 2. First we will give a description of standard normal features, then prove the general version.

**Theorem 5** (Standard normal feature, noiseless) *Let data be generated as in Phase 1, let* $\mathcal{S} = \max\{\|\mathbf{\Sigma}_F\|, \|\mathbf{\Sigma}_T\|\}$ *in this theorem and the following section*[10], $\tilde{r}_T = \mathbf{tr}(\mathbf{\Sigma}_T\mathbf{\Sigma}_F)$, $r_F = \mathbf{tr}(\mathbf{\Sigma}_F)$, $r_T = \mathbf{tr}(\mathbf{\Sigma}_T)$. *Suppose* $\sigma = 0$, $\mathbf{\Sigma}_F = \mathbf{I}$, *and suppose the rank of* $\mathbf{\Sigma}_T$ *is* $s_T$. *Define* $\hat{\boldsymbol{\beta}}_i = n_1^{-1}\sum_{j=1}^{n_1} y_{ij}\boldsymbol{x}_{ij}$, $\boldsymbol{B} = [\boldsymbol{\beta}_1, ..., \boldsymbol{\beta}_T]$, *and* $\hat{\boldsymbol{B}} = [\hat{\boldsymbol{\beta}}_1, ..., \hat{\boldsymbol{\beta}}_T]$. *Let* $n_1 > c_1 r_T \lambda_{s_T}^{-1}(\mathbf{\Sigma}_T)$, *with probability* $1 - \mathcal{O}(T^{-C})$, *where* $C$ *is constant,*

$$\sigma_{\max}(\hat{\boldsymbol{B}} - \boldsymbol{B}) \lesssim \sqrt{\frac{Tr_T}{n_1}}.$$

*Denote the span of top* $s_T$ *singular column vectors of* $\hat{\boldsymbol{B}}$ *and* $\mathbf{\Sigma}_T$ *as* $\hat{\boldsymbol{W}}, \boldsymbol{W}$, *then*

$$\sin(\angle\hat{\boldsymbol{W}}, \boldsymbol{W}) \lesssim \sqrt{\frac{r_T}{n_1\lambda_{s_T}(\mathbf{\Sigma}_T)}}.$$

*For example, if* $\mathbf{\Sigma}_T = \mathrm{diag}(\boldsymbol{I}_{s_T}, 0)$, *then* $\sin(\angle\hat{\boldsymbol{W}}, \boldsymbol{W}) \lesssim \sqrt{s_T/n_1}$.

**Proof** We first estimate $\boldsymbol{\beta}_i$ with

$$\hat{\boldsymbol{\beta}}_i = \frac{1}{n_1}\sum_{j=1}^{n_1} y_{ij}\boldsymbol{x}_{ij}.$$

Then we fix $\boldsymbol{\beta}_i$ and compute the covariance of $y_{ij}\boldsymbol{x}_{ij}$ (its mean is $\boldsymbol{\beta}_i$).

$$\mathrm{Cov}(y_{ij}\boldsymbol{x}_{ij} - \boldsymbol{\beta}_i) = \boldsymbol{E}(\boldsymbol{x}_{ij}\boldsymbol{x}_{ij}^\top\boldsymbol{\beta}_i\boldsymbol{\beta}_i^\top\boldsymbol{x}_{ij}\boldsymbol{x}_{ij}^\top) - \boldsymbol{\beta}_i\boldsymbol{\beta}_i^\top \precsim \|\boldsymbol{\beta}_i\|^2\boldsymbol{I}.$$

The first term is similar to (C.4), where the bound can is in [37, Lemma 5]. The vector $\hat{\boldsymbol{\beta}}_i$ is the average of $y_{ij}\boldsymbol{x}_{ij}$ over all $j$. With concentration we know that

$$\mathrm{Cov}(\hat{\boldsymbol{\beta}}_i - \boldsymbol{\beta}_i) \precsim \frac{\|\boldsymbol{\beta}_i\|^2}{n_1}\boldsymbol{I}. \tag{C.10}$$

Let $\boldsymbol{B} = [\boldsymbol{\beta}_1, ..., \boldsymbol{\beta}_T]$, and $\hat{\boldsymbol{B}} = [\hat{\boldsymbol{\beta}}_1, ..., \hat{\boldsymbol{\beta}}_T]$. Then we know the covariance of each column of $\hat{\boldsymbol{B}} - \boldsymbol{B}$ is bounded by (C.10). Thus with a constant $c$ and probability $1 - \exp(-cT^2)$,

$$\sigma_{\max}^2(\hat{\boldsymbol{B}} - \boldsymbol{B}) \lesssim \frac{T\|\boldsymbol{\beta}_i\|^2}{n_1}. \tag{C.11}$$

We have proved in (C.7) that $\|\boldsymbol{\beta}_i\|^2 \le \log(T)\mathbf{tr}(\mathbf{\Sigma}_T)$ with probability $1 - T^{-c}$. The columns of $\boldsymbol{B}$ is generated from $\mathcal{N}(0, \mathbf{\Sigma}_T)$, so that

$$\sigma_{\max}(\hat{\boldsymbol{B}} - \boldsymbol{B}) \lesssim \sqrt{\frac{T\log(T)\mathbf{tr}(\mathbf{\Sigma}_T)}{n_1}}.$$

Now we study $\boldsymbol{B}$. We know that $\boldsymbol{E}(\boldsymbol{B}\boldsymbol{B}^\top) = \boldsymbol{E}(\sum_{i=1}^T \boldsymbol{\beta}_i\boldsymbol{\beta}_i^\top) = T\mathbf{\Sigma}_T$. $\boldsymbol{B}$ is a matrix with independent columns. Thus let $n_1 > c_1\mathbf{tr}(\mathbf{\Sigma}_T)\lambda_{s_T}^{-1}(\mathbf{\Sigma}_T)$, $T > \max\{c_2 d, \frac{\|\mathbf{\Sigma}_T\|\mathbf{tr}(\mathbf{\Sigma}_T)}{\lambda_{s_T}^2(\mathbf{\Sigma}_T)}\}$, then with Lemma 3, for Gaussian matrix with independent columns [38], with probability at least $1 - \mathcal{O}(T^{-c_3} + (T\mathbf{tr}(\mathbf{\Sigma}_T))^{-c_4} + \exp(-c_5T^2)) = 1 - \mathcal{O}(T^{-C})$, where $c_i$ are constants,

$$\sigma_{s_T}(\boldsymbol{B}) \ge \sqrt{T\lambda_{s_T}(\mathbf{\Sigma}_T) - \mathcal{O}(\sqrt{T\|\mathbf{\Sigma}_T\|\mathbf{tr}(\mathbf{\Sigma}_T)})}.$$

Denote the span of top $s_T$ singular vectors of $\hat{\boldsymbol{B}}$ and $\mathbf{\Sigma}_T$ as $\hat{\boldsymbol{W}}, \boldsymbol{W}$, with Lemma 4,

$$\sin(\angle\hat{\boldsymbol{W}}, \boldsymbol{W}) \le \sqrt{\frac{\log(T)\mathbf{tr}(\mathbf{\Sigma}_T)}{n_1\lambda_{s_T}(\mathbf{\Sigma}_T)}}.$$

---

[10]in the paper we assume $\mathcal{S} = 1$ for simplicity.

Next, we will propose a theorem with general feature covariance and noisy data, which is a generalization of Theorem 5.

**Theorem 6** *Let data be generated as in Phase 1. Suppose $\hat{\boldsymbol{b}}_i = n_1^{-1} \sum_{j=1}^{n_1} y_{ij} \boldsymbol{x}_{ij}$, $\boldsymbol{B} = \boldsymbol{\Sigma}_F[\boldsymbol{\beta}_1, ..., \boldsymbol{\beta}_T]$, and $\hat{\boldsymbol{B}} = [\hat{\boldsymbol{b}}_1, ..., \hat{\boldsymbol{b}}_T]$. Let $\delta_\lambda = \lambda_{s_T}(\boldsymbol{\Sigma}_F \boldsymbol{\Sigma}_T \boldsymbol{\Sigma}_F) - \lambda_{s_T+1}(\boldsymbol{\Sigma}_F \boldsymbol{\Sigma}_T \boldsymbol{\Sigma}_F))$, suppose $\boldsymbol{\Sigma}_F$ is approximately rank $s_F$,*

$$n_1 \gtrsim (\mathbf{tr}(\boldsymbol{\Sigma}_T \boldsymbol{\Sigma}_F) + \sigma^2) \|\boldsymbol{\Sigma}_F\|,$$

$$T \gtrsim \max\{s_F, \frac{d\lambda_{s_F+1}(\boldsymbol{\Sigma}_F)}{\|\boldsymbol{\Sigma}_F\|}\},$$

*then with probability $1 - \mathcal{O}(T^{-C})$, where $C$ is constant,*

$$\sigma_{\max}(\hat{\boldsymbol{B}} - \boldsymbol{B}) \lesssim \sqrt{\frac{T(\mathbf{tr}(\boldsymbol{\Sigma}_T \boldsymbol{\Sigma}_F) + \sigma^2) \|\boldsymbol{\Sigma}_F\|}{n_1}}.$$

*Denote the span of top $s_T$ singular vectors of $\hat{\boldsymbol{B}}$ and $\boldsymbol{\Sigma}_F \boldsymbol{\Sigma}_T \boldsymbol{\Sigma}_F$ as $\hat{\boldsymbol{W}}, \boldsymbol{W}$, if further we assume $T \gtrsim \frac{\|\boldsymbol{\Sigma}_F \boldsymbol{\Sigma}_T \boldsymbol{\Sigma}_F\| \mathbf{tr}(\boldsymbol{\Sigma}_F \boldsymbol{\Sigma}_T \boldsymbol{\Sigma}_F)}{\delta_\lambda^2}$, then*

$$\sin(\angle \hat{\boldsymbol{W}}, \boldsymbol{W}) \lesssim \sqrt{\frac{(\mathbf{tr}(\boldsymbol{\Sigma}_T \boldsymbol{\Sigma}_F) + \sigma^2) \|\boldsymbol{\Sigma}_F\|}{n_1 \delta_\lambda^2}}.$$

**Example 3** *Suppose $\boldsymbol{\Sigma}_F = diag(\boldsymbol{I}_{s_F}, \iota \boldsymbol{I}_{d-s_F})$, and $\boldsymbol{\Sigma}_T = diag(\boldsymbol{I}_{s_T}, 0)$, $\sigma = 0$. Suppose $\iota d < s_F$. Then with $T \gtrsim s_F$, $n_1 \gtrsim s_T$ so that $N \gtrsim s_F s_T$,*

$$\sin(\angle \hat{\boldsymbol{W}}, \boldsymbol{W}) \lesssim \sqrt{s_T / n}.$$

**Proof** We let $\boldsymbol{x}_{ij} \sim \mathcal{N}(0, \boldsymbol{\Sigma}_F)$. For the $i$th task, let

$$\hat{\boldsymbol{b}}_i = \frac{1}{n_1} \sum_{j=1}^{n_1} y_{ij} \boldsymbol{x}_{ij}.$$

We fix $\boldsymbol{\beta}_i$ and compute

$$\boldsymbol{E}(y_{ij} \boldsymbol{x}_{ij}) \precsim \boldsymbol{E}(\boldsymbol{x}_{ij} \boldsymbol{x}_{ij}^\top \boldsymbol{\beta}_i) = \boldsymbol{\Sigma}_F \boldsymbol{\beta}_i, \tag{C.12}$$

and

$$\text{Cov}(y_{ij} \boldsymbol{x}_{ij} - \boldsymbol{\Sigma}_F \boldsymbol{\beta}_i) \precsim (\boldsymbol{\beta}_i^\top \boldsymbol{\Sigma}_F \boldsymbol{\beta}_i) \boldsymbol{\Sigma}_F + \sigma^2 \boldsymbol{\Sigma}_F. \tag{C.13}$$

To get the bound above, we can adopt the technique in [37, Lemma 5] such that, write $\boldsymbol{x}_{ij} = \boldsymbol{\Sigma}_F^{1/2} \boldsymbol{z}$, and reduce to $\boldsymbol{E}((\boldsymbol{z}^\top \boldsymbol{\Sigma}_F^{1/2} \boldsymbol{\beta}_i)^2 \boldsymbol{\Sigma}_F^{1/2} \boldsymbol{z} \boldsymbol{z}^\top \boldsymbol{\Sigma}_F^{1/2})$. The proof of [37, Lemma 5] gives the explicit bound of $\|\boldsymbol{E}((\boldsymbol{z}^\top \boldsymbol{\alpha})^2 \boldsymbol{z} \boldsymbol{z}^\top)\|$ for any $\boldsymbol{\alpha}$ that equals above. The vector $\hat{\boldsymbol{b}}_i$ is the average of $y_{ij} \boldsymbol{x}_{ij}$ over all $j = 1, ..., n_1$. With concentration we know that

$$\text{Cov}(\hat{\boldsymbol{b}}_i - \boldsymbol{\Sigma}_F \boldsymbol{\beta}_i) \precsim \frac{\boldsymbol{\beta}_i^\top \boldsymbol{\Sigma}_F \boldsymbol{\beta}_i + \sigma^2}{n_1} \boldsymbol{\Sigma}_F. \tag{C.14}$$

Suppose $\boldsymbol{B} = \boldsymbol{\Sigma}_F[\boldsymbol{\beta}_1, ..., \boldsymbol{\beta}_T]$, and $\hat{\boldsymbol{B}} = [\boldsymbol{b}_1, ..., \boldsymbol{b}_T]$. $\hat{\boldsymbol{B}} - \boldsymbol{B}$ is a matrix with independent columns. Suppose $\boldsymbol{X}$ is approximately rank $s_F$, Let $\boldsymbol{V}_{s_F} \in \mathbb{R}^{d \times d}$ be the projection onto the top-$R$ singular vector space of $\boldsymbol{\Sigma}_F$ and $\boldsymbol{V}_{s_F^\perp} \in \mathbb{R}^{d \times d}$ be the projection onto the $s_F + 1$ to $d$th singular vector space of $\boldsymbol{\Sigma}_F$. With $T$ columns and $T \geq s_F$, we know that

$$\sigma_{\max}(\boldsymbol{V}_{s_F}(\hat{\boldsymbol{B}} - \boldsymbol{B})) \lesssim \frac{T(\max_i \boldsymbol{\beta}_i^\top \boldsymbol{\Sigma}_F \boldsymbol{\beta}_i + \sigma^2) \|\boldsymbol{\Sigma}_F\|}{n_1}$$

$$\sigma_{\max}(\boldsymbol{V}_{s_F^\perp}(\hat{\boldsymbol{B}} - \boldsymbol{B})) \lesssim \frac{\max\{T, d\}(\max_i \boldsymbol{\beta}_i^\top \boldsymbol{\Sigma}_F \boldsymbol{\beta}_i + \sigma^2) \lambda_{s_T+1}(\boldsymbol{\Sigma}_F)}{n_1}$$

With similar argument as before, with probability $1 - \exp(-cT^2)$ for constant $c$,

$$\sigma_{\max}^2(\hat{\boldsymbol{B}} - \boldsymbol{B}) \lesssim \frac{\max\{T\|\boldsymbol{\Sigma}_F\|, d\lambda_{s_F+1}(\boldsymbol{\Sigma}_F)\}(\max_i \boldsymbol{\beta}_i^\top \boldsymbol{\Sigma}_F \boldsymbol{\beta}_i + \sigma^2)\|\boldsymbol{\Sigma}_F\|}{n_1}. \quad \text{(C.15)}$$

We know in (C.9) that $\|\boldsymbol{\Sigma}_F^{1/2}\boldsymbol{\beta}_i\|^2 \leq \mathcal{O}(\log(T)\mathbf{tr}(\boldsymbol{\Sigma}_T\boldsymbol{\Sigma}_F))$ with probability $1 - T^{-c}$ for constant $c$. So that

$$\sigma_{\max}(\hat{\boldsymbol{B}} - \boldsymbol{B}) \lesssim \sqrt{\frac{\max\{T\|\boldsymbol{\Sigma}_F\|, d\lambda_{s_F+1}(\boldsymbol{\Sigma}_F)\}(\log(T)\mathbf{tr}(\boldsymbol{\Sigma}_T\boldsymbol{\Sigma}_F) + \sigma^2)\|\boldsymbol{\Sigma}_F\|}{n_1}}. \quad \text{(C.16)}$$

Now we study $\boldsymbol{B}$. $\boldsymbol{E}(\boldsymbol{BB}^\top) = \boldsymbol{E}(\boldsymbol{\Sigma}_F(\sum_{i=1}^T \boldsymbol{\beta}_i\boldsymbol{\beta}_i^\top)\boldsymbol{\Sigma}_F) = T\boldsymbol{\Sigma}_F\boldsymbol{\Sigma}_T\boldsymbol{\Sigma}_F$.

Thus let

$$n_1 > C_1(\log(T)\mathbf{tr}(\boldsymbol{\Sigma}_T\boldsymbol{\Sigma}_F) + \sigma^2)\|\boldsymbol{\Sigma}_F\|.$$

Now apply the concentration of Gaussian matrix with independent columns [38]. With probability $1 - \mathcal{O}(T^{-C_1} + (T\mathbf{tr}(\boldsymbol{\Sigma}_F\boldsymbol{\Sigma}_T\boldsymbol{\Sigma}_F))^{-C_2} + \exp(-C_3T^2))$, where $C_i$ are constants (the probability can be simplified as $1 - \mathcal{O}(T^{-C})$),

$$\sigma_{s_T}(\boldsymbol{B}) \geq \sqrt{T(\lambda_{s_T}(\boldsymbol{\Sigma}_F\boldsymbol{\Sigma}_T\boldsymbol{\Sigma}_F) - \lambda_{s_T+1}(\boldsymbol{\Sigma}_F\boldsymbol{\Sigma}_T\boldsymbol{\Sigma}_F)) - \mathcal{O}(\sqrt{T\|\boldsymbol{\Sigma}_F\boldsymbol{\Sigma}_T\boldsymbol{\Sigma}_F\|\mathbf{tr}(\boldsymbol{\Sigma}_F\boldsymbol{\Sigma}_T\boldsymbol{\Sigma}_F)})}.$$

Denote the span of top $s_T$ singular vectors of $\hat{\boldsymbol{B}}$ and $\boldsymbol{\Sigma}_F\boldsymbol{\Sigma}_T\boldsymbol{\Sigma}_F$ as $\hat{\boldsymbol{W}}, \boldsymbol{W}$, let

$$T \gtrsim \max\{s_F, \frac{d\lambda_{s_F+1}(\boldsymbol{\Sigma}_F)}{\|\boldsymbol{\Sigma}_F\|}, \frac{\|\boldsymbol{\Sigma}_F\boldsymbol{\Sigma}_T\boldsymbol{\Sigma}_F\|\mathbf{tr}(\boldsymbol{\Sigma}_F\boldsymbol{\Sigma}_T\boldsymbol{\Sigma}_F)}{(\lambda_{s_T}(\boldsymbol{\Sigma}_F\boldsymbol{\Sigma}_T\boldsymbol{\Sigma}_F) - \lambda_{s_T+1}(\boldsymbol{\Sigma}_F\boldsymbol{\Sigma}_T\boldsymbol{\Sigma}_F))^2}\} \quad \text{(C.17)}$$

we plug in (C.16) and Lemma 4,

$$\sin(\angle\hat{\boldsymbol{W}}, \boldsymbol{W}) \lesssim \sqrt{(\frac{d\lambda_{s_F+1}(\boldsymbol{\Sigma}_F)}{T\|\boldsymbol{\Sigma}_F\|} + 1) \cdot \frac{(\mathbf{tr}(\boldsymbol{\Sigma}_T\boldsymbol{\Sigma}_F) + \sigma^2)\|\boldsymbol{\Sigma}_F\|}{n_1(\lambda_{s_T}(\boldsymbol{\Sigma}_F\boldsymbol{\Sigma}_T\boldsymbol{\Sigma}_F) - \lambda_{s_T+1}(\boldsymbol{\Sigma}_F\boldsymbol{\Sigma}_T\boldsymbol{\Sigma}_F))}}$$

$$\approx \sqrt{\frac{(\mathbf{tr}(\boldsymbol{\Sigma}_T\boldsymbol{\Sigma}_F) + \sigma^2)\|\boldsymbol{\Sigma}_F\|}{n_1(\lambda_{s_T}(\boldsymbol{\Sigma}_F\boldsymbol{\Sigma}_T\boldsymbol{\Sigma}_F) - \lambda_{s_T+1}(\boldsymbol{\Sigma}_F\boldsymbol{\Sigma}_T\boldsymbol{\Sigma}_F))}}.$$

∎

### C.2.3 Method of moments with arbitrary $n_1$

In this subsection we will analyze $\hat{\boldsymbol{B}}$ with any $n_1$, and propose the error of MoM estimator.

First, suppose there are at least two samples per task, we can separate the samples into two halves, and compute the following estimator.

**Theorem 7** *Let data be generated as in Phase 1, and let $n_1$ be a even number. Define $\hat{\boldsymbol{b}}_{i,1} = 2n_1^{-1}\sum_{j=1}^{n_1/2} y_{ij}\boldsymbol{x}_{ij}$, $\hat{\boldsymbol{b}}_{i,2} = 2n_1^{-1}\sum_{j=n_1/2+1}^{n_1} y_{ij}\boldsymbol{x}_{ij}$. Define*

$$\hat{\boldsymbol{M}} = n_1^{-1}\sum_{i=1}^T (\boldsymbol{b}_{i,1}\boldsymbol{b}_{i,2}^\top + \boldsymbol{b}_{i,2}\boldsymbol{b}_{i,1}^\top),$$

$$\boldsymbol{M} = \boldsymbol{\Sigma}_F\boldsymbol{\Sigma}_T\boldsymbol{\Sigma}_F.$$

*Then there is a constant $c > 10$, with probability $1 - N^{-c}$,*

$$\|\hat{\boldsymbol{M}} - \boldsymbol{M}\| \lesssim (\tilde{r}_T + \sigma^2)\sqrt{\frac{r_F}{N}} + \sqrt{\frac{r_T}{T}}.$$

**Proof** For simplicity of notation, we will define a random vector $\boldsymbol{x}$ with zero mean and covariance $\boldsymbol{\Sigma}_F$, a random vector $\boldsymbol{\beta}$ with zero mean and covariance $\boldsymbol{\Sigma}_T$, a random variable $\varepsilon$ with zero mean and covariance $\sigma$, and they are subGaussian[11]. Let $y = \boldsymbol{x}^\top\boldsymbol{\beta} + \varepsilon$. We first estimate the mean of $\hat{\boldsymbol{M}}$.

---

[11]We remove the subscripts when there is no confusion.

Note that if we fix $\boldsymbol{\beta}$, $\hat{\boldsymbol{b}}_{i,1}, \hat{\boldsymbol{b}}_{i,2}$ are i.i.d., so

$$\boldsymbol{E}_{\boldsymbol{x},\varepsilon}(\hat{\boldsymbol{b}}_{i,1}) = \boldsymbol{E}_{\boldsymbol{x},\varepsilon}(y\boldsymbol{x}) = \boldsymbol{E}_{\boldsymbol{x},\varepsilon}((\boldsymbol{x}^\top\boldsymbol{\beta} + \varepsilon)\boldsymbol{x}) = \boldsymbol{\Sigma}_F\boldsymbol{\beta},$$

$$\boldsymbol{E}_{\boldsymbol{x},\varepsilon}(\hat{\boldsymbol{M}}) = \frac{1}{2}(\boldsymbol{E}_{\boldsymbol{x},\varepsilon}(\hat{\boldsymbol{b}}_{i,1})\boldsymbol{E}_{\boldsymbol{x},\varepsilon}(\hat{\boldsymbol{b}}_{i,2})^\top + \boldsymbol{E}_{\boldsymbol{x},\varepsilon}(\hat{\boldsymbol{b}}_{i,2})\boldsymbol{E}_{\boldsymbol{x},\varepsilon}(\hat{\boldsymbol{b}}_{i,1})^\top)$$

$$= \boldsymbol{E}_{\boldsymbol{x},\varepsilon}(\hat{\boldsymbol{b}}_{i,1})\boldsymbol{E}_{\boldsymbol{x},\varepsilon}(\hat{\boldsymbol{b}}_{i,1})^\top = \frac{1}{T}\boldsymbol{\Sigma}_F(\sum_{i=1}^T \boldsymbol{\beta}_i\boldsymbol{\beta}_i^\top)\boldsymbol{\Sigma}_F.$$

We take expectation over $\boldsymbol{\beta}_i$ and get $\boldsymbol{M}$. We define the right hand side as $\bar{\boldsymbol{M}}$ for the proof below.

Next, we will bound $\|\hat{\boldsymbol{M}} - \boldsymbol{M}\|$.

[37, Lemma 3] proposes that, with probability $1 - \delta$,

$$\|\boldsymbol{x}_{ij}\|^2 \lesssim \log(1/\delta)\mathbf{tr}(\boldsymbol{\Sigma}_F),$$
$$(\boldsymbol{x}_{ij}^\top\boldsymbol{\beta}_i)^2 \lesssim \log(1/\delta)\mathbf{tr}(\boldsymbol{\Sigma}_F\boldsymbol{\Sigma}_T),$$
$$\varepsilon_{ij}^2 \lesssim \log(1/\delta)\sigma^2.$$

If we enumerate $i = 1, ..., T$ and $j = 1, ..., n_1$, there are in total $Tn_1 = N$ terms. So we set $\delta = N^{-c+1}$ for a constant $c > 1$, then with probability $1 - N^{-c}$, for all $i, j$ we have

$$\|y_{ij}\boldsymbol{x}_{ij}\| = \|(\boldsymbol{x}_{ij}\boldsymbol{\beta}_i + \varepsilon_{ij})\boldsymbol{x}_{ij}\| \lesssim \log^{3/2}(N)\sqrt{(\mathbf{tr}(\boldsymbol{\Sigma}_F\boldsymbol{\Sigma}_T) + \sigma^2)\mathbf{tr}(\boldsymbol{\Sigma}_F)}.$$

Define $\boldsymbol{\delta}_{i,l} = \hat{\boldsymbol{b}}_{i,l} - \boldsymbol{\Sigma}_F\boldsymbol{\beta}_i$ for $l = 1, 2$ (we will use $l = 1$ below, the result for $l = 2$ is the same). Note that $\boldsymbol{\delta}_i$ is zero mean. With [23, Prop. 5.1] we have with probability $1 - N^{-c}$,

$$\|\boldsymbol{\delta}_{i,1}\| \lesssim n_1^{-1/2}\log^{5/2}(N)\sqrt{(\mathbf{tr}(\boldsymbol{\Sigma}_F\boldsymbol{\Sigma}_T) + \sigma^2)\mathbf{tr}(\boldsymbol{\Sigma}_F)} \tag{C.18}$$

Define

$$\boldsymbol{Z}_i = \hat{\boldsymbol{b}}_{i,1}\hat{\boldsymbol{b}}_{i,2}^\top - \boldsymbol{E}_{\boldsymbol{x},\varepsilon}(\hat{\boldsymbol{b}}_{i,1}\hat{\boldsymbol{b}}_{i,2}^\top)$$

$$= (\boldsymbol{\Sigma}_F\boldsymbol{\beta}_i + \boldsymbol{\delta}_{i,1})(\boldsymbol{\Sigma}_F\boldsymbol{\beta}_i + \boldsymbol{\delta}_{i,2})^\top - \boldsymbol{E}_{\boldsymbol{x},\varepsilon}(\hat{\boldsymbol{b}}_{i,1}\hat{\boldsymbol{b}}_{i,2}^\top)$$

$$= \boldsymbol{\delta}_{i,1}(\boldsymbol{\Sigma}_F\boldsymbol{\beta}_i)^\top + \boldsymbol{\Sigma}_F\boldsymbol{\beta}_i\boldsymbol{\delta}_{i,2}^\top + \boldsymbol{\delta}_{i,1}\boldsymbol{\delta}_{i,2}^\top - \boldsymbol{E}_{\boldsymbol{x},\varepsilon}(\boldsymbol{\delta}_{i,1}\boldsymbol{\delta}_{i,2}^\top).$$

Then

$$\|\boldsymbol{E}\boldsymbol{Z}_i\boldsymbol{Z}_i^\top\| \leq \|\boldsymbol{E}(\boldsymbol{\Sigma}_F\boldsymbol{\beta}_i\boldsymbol{\delta}_{i,2}^\top + \boldsymbol{\delta}_{i,1}(\boldsymbol{\Sigma}_F\boldsymbol{\beta}_i)^\top)(\boldsymbol{\Sigma}_F\boldsymbol{\beta}_i\boldsymbol{\delta}_{i,2}^\top + \boldsymbol{\delta}_{i,1}(\boldsymbol{\Sigma}_F\boldsymbol{\beta}_i)^\top)^\top\|$$
$$+ \|\boldsymbol{E}\boldsymbol{\delta}_{i,1}\boldsymbol{\delta}_{i,2}^\top\boldsymbol{\delta}_{i,2}\boldsymbol{\delta}_{i,1}^\top\|. \tag{C.19}$$

Then we can use (C.18) and (C.8) to bound the first term by

$$n_1^{-1}\log^6(N)(\mathbf{tr}(\boldsymbol{\Sigma}_F\boldsymbol{\Sigma}_T) + \sigma)\mathbf{tr}(\boldsymbol{\Sigma}_F)\mathbf{tr}(\boldsymbol{\Sigma}_F^2\boldsymbol{\Sigma}_T)\|\boldsymbol{\Sigma}_F\|^2.$$

And

$$\boldsymbol{E}_{\boldsymbol{x},\varepsilon}\boldsymbol{\delta}_{i,1}\boldsymbol{\delta}_{i,2}^\top\boldsymbol{\delta}_{i,2}\boldsymbol{\delta}_{i,1}^\top = (\boldsymbol{E}_{\boldsymbol{x}}\boldsymbol{\delta}_{i,2}^\top\boldsymbol{\delta}_{i,2})\|\boldsymbol{E}_{\boldsymbol{x}}\boldsymbol{\delta}_{i,1}\boldsymbol{\delta}_{i,1}^\top\|$$
$$\lesssim n_1^{-2}(\boldsymbol{E}_{\boldsymbol{x},\varepsilon}(\boldsymbol{x}^\top\boldsymbol{\beta} + \varepsilon)^2\boldsymbol{x}^\top\boldsymbol{x})\|\boldsymbol{E}_{\boldsymbol{x},\varepsilon}(\boldsymbol{x}^\top\boldsymbol{\beta} + \varepsilon)^2\boldsymbol{x}\boldsymbol{x}^\top\|$$
$$\lesssim n_1^{-2}(\mathbf{tr}^2(\boldsymbol{\Sigma}_F\boldsymbol{\Sigma}_T) + \sigma^4)\mathbf{tr}(\boldsymbol{\Sigma}_F)\|\boldsymbol{\Sigma}_F\|.$$

The second line is due to the fact that $\boldsymbol{\delta}_{i,l}$ is the difference of $(\boldsymbol{x}^\top\boldsymbol{\beta} + \varepsilon)\boldsymbol{x}$ and its mean, and covariance is upper bounded by variance (not subtracting the mean). The $n_1^{-2}$ factor comes from the average over $n_1$ terms. The reasoning of the last line is same as (C.13). Now we can go back to (C.19) and get

$$\|\boldsymbol{E}\boldsymbol{Z}_i\boldsymbol{Z}_i^\top\| \lesssim n_1^{-1}\log^6(N)(\mathbf{tr}(\boldsymbol{\Sigma}_F^2\boldsymbol{\Sigma}_T) + \mathbf{tr}(\boldsymbol{\Sigma}_F\boldsymbol{\Sigma}_T) + \sigma^2)^2\mathbf{tr}(\boldsymbol{\Sigma}_F)\|\boldsymbol{\Sigma}_F\|^2.$$

Next we need to bound the norm of $\boldsymbol{Z}_i$. We use (C.18) and (C.8), with probability $1 - N^{-c}$,

$$\|\boldsymbol{Z}_i\| \leq n_1^{-1/2}\log^3(N)(\mathbf{tr}(\boldsymbol{\Sigma}_F^2\boldsymbol{\Sigma}_T) + \mathbf{tr}(\boldsymbol{\Sigma}_F\boldsymbol{\Sigma}_T) + \sigma^2)\sqrt{\mathbf{tr}(\boldsymbol{\Sigma}_F)}\|\boldsymbol{\Sigma}_F\|$$
$$+ n_1^{-1}\log^5(N)(\mathbf{tr}(\boldsymbol{\Sigma}_F\boldsymbol{\Sigma}_T) + \sigma^2)\mathbf{tr}(\boldsymbol{\Sigma}_F).$$

Define the upper bound for $\|\boldsymbol{E}\boldsymbol{Z}_i\boldsymbol{Z}_i^\top\|$, $\|\boldsymbol{Z}_i\|$ as $Z_1$, $Z_2$ (the right hand side of two above inequalities). Now we apply Bernstein type inequality (Lemma 2), with probability $1 - N^{-c}$,

$$
\|\hat{\boldsymbol{M}} - \bar{\boldsymbol{M}}\|
$$

$$
= \|T^{-1}\sum_{i=1}^{T}\boldsymbol{Z}_i - \boldsymbol{E}_{\boldsymbol{x}}\boldsymbol{Z}_i\|
$$

$$
\lesssim \log(TZ_2)\left(T^{-1/2}\log(N)Z_1^{1/2} + T^{-1}Z_2\log(TZ_2)\right)
$$

$$
\lesssim \log(TZ_2)\Big(\sqrt{\frac{\log^6(N)(\mathbf{tr}(\boldsymbol{\Sigma}_F^2\boldsymbol{\Sigma}_T) + \mathbf{tr}(\boldsymbol{\Sigma}_F\boldsymbol{\Sigma}_T) + \sigma^2)^2\mathbf{tr}(\boldsymbol{\Sigma}_F)\|\boldsymbol{\Sigma}_F\|^2}{n_1T}}
$$

$$
+ \frac{\log^3(N)(\mathbf{tr}(\boldsymbol{\Sigma}_F^2\boldsymbol{\Sigma}_T) + \mathbf{tr}(\boldsymbol{\Sigma}_F\boldsymbol{\Sigma}_T) + \sigma^2)\sqrt{\mathbf{tr}(\boldsymbol{\Sigma}_F)}\|\boldsymbol{\Sigma}_F\|}{n_1^{1/2}T}
$$

$$
+ \frac{\log^5(N)(\mathbf{tr}(\boldsymbol{\Sigma}_F\boldsymbol{\Sigma}_T) + \sigma^2)\mathbf{tr}(\boldsymbol{\Sigma}_F)}{T}\Big)
$$

$$
= \log(TZ_2)\cdot\Big(\log^3(N)\|\boldsymbol{\Sigma}_F\|(\mathbf{tr}(\boldsymbol{\Sigma}_F^2\boldsymbol{\Sigma}_T) + \mathbf{tr}(\boldsymbol{\Sigma}_F\boldsymbol{\Sigma}_T) + \sigma^2)\sqrt{\frac{\mathbf{tr}(\boldsymbol{\Sigma}_F)}{N}}
$$

$$
+ \frac{\log^5(N)(\mathbf{tr}(\boldsymbol{\Sigma}_F^2\boldsymbol{\Sigma}_T) + \mathbf{tr}(\boldsymbol{\Sigma}_F\boldsymbol{\Sigma}_T) + \sigma^2)\sqrt{\mathbf{tr}(\boldsymbol{\Sigma}_F)}\|\boldsymbol{\Sigma}_F\|}{N^{1/2}T^{1/2}}\Big).
$$

The term

$$
\|\boldsymbol{\Sigma}_F\|(\mathbf{tr}(\boldsymbol{\Sigma}_F^2\boldsymbol{\Sigma}_T) + \mathbf{tr}(\boldsymbol{\Sigma}_F\boldsymbol{\Sigma}_T) + \sigma^2)\sqrt{\frac{\mathbf{tr}(\boldsymbol{\Sigma}_F)}{N}}
$$

is the dominant term as shown in the theorem. ∎

The following method of moment estimator is used in [37], where $n_1 \geq 1$. In other words, if there is one sample per task, one can use the following estimator.

**Theorem 8** *Let data be generated as in Phase 1. Define* $\hat{\boldsymbol{b}}_i = n_1^{-1}\sum_{j=1}^{n_1} y_{ij}\boldsymbol{x}_{ij}$, $\boldsymbol{B} = \boldsymbol{\Sigma}_F[\boldsymbol{\beta}_1, ..., \boldsymbol{\beta}_T]$, *and* $\hat{\boldsymbol{B}} = [\hat{\boldsymbol{b}}_1, ..., \hat{\boldsymbol{b}}_T]$. *Define*

$$
\hat{\mathbf{G}} = \hat{\boldsymbol{B}}\hat{\boldsymbol{B}}^\top = T^{-1}\sum_{i=1}^{T}\hat{\boldsymbol{b}}_i\hat{\boldsymbol{b}}_i^\top,
$$

$$
\mathbf{G} = \boldsymbol{E}(\hat{\boldsymbol{B}}\hat{\boldsymbol{B}}^\top) = \boldsymbol{\Sigma}_F\boldsymbol{\Sigma}_T\boldsymbol{\Sigma}_F + n_1^{-1}(\boldsymbol{\Sigma}_F\boldsymbol{\Sigma}_T\boldsymbol{\Sigma}_F + \mathbf{tr}(\boldsymbol{\Sigma}_T\boldsymbol{\Sigma}_F)\boldsymbol{\Sigma}_F + \sigma^2\boldsymbol{\Sigma}_F),
$$

$$
\bar{\boldsymbol{\Sigma}}_T = \sum_{i=1}^{T}\boldsymbol{\beta}_i\boldsymbol{\beta}_i^\top,
$$

$$
\bar{\mathbf{G}} = \boldsymbol{\Sigma}_F\bar{\boldsymbol{\Sigma}}_T\boldsymbol{\Sigma}_F + n_1^{-1}(\boldsymbol{\Sigma}_F\bar{\boldsymbol{\Sigma}}_T\boldsymbol{\Sigma}_F + \mathbf{tr}(\bar{\boldsymbol{\Sigma}}_T\boldsymbol{\Sigma}_F)\boldsymbol{\Sigma}_F + \sigma^2\boldsymbol{\Sigma}_F)
$$

*With probability* $1 - N^c$,

$$
\|\hat{\mathbf{G}} - \bar{\mathbf{G}}\| \lesssim \|\boldsymbol{\Sigma}_F\|(\mathbf{tr}(\boldsymbol{\Sigma}_F^2\boldsymbol{\Sigma}_T) + \mathbf{tr}(\boldsymbol{\Sigma}_F\boldsymbol{\Sigma}_T) + \sigma^2)\sqrt{\frac{\mathbf{tr}(\boldsymbol{\Sigma}_F)}{N}}.
$$

**Proof** First, we compute the expectation of $\hat{\mathbf{G}}$.

$$
\boldsymbol{E}_{\boldsymbol{x},y,\varepsilon}\hat{\mathbf{G}} = \boldsymbol{E}_{\boldsymbol{x},y,\varepsilon}T^{-1}(\sum_{i=1}^{T}\hat{\boldsymbol{b}}_i\hat{\boldsymbol{b}}_i^\top),
$$

$$
\boldsymbol{E}_{\boldsymbol{x},y,\varepsilon}\hat{\boldsymbol{b}}_i\hat{\boldsymbol{b}}_i^\top = \boldsymbol{E}_{\boldsymbol{x},y,\varepsilon}\left(n_1^{-1}\sum_{j=1}^{n_1}(\boldsymbol{\beta}_i^\top\boldsymbol{x}_{ij} + \varepsilon_{ij})\boldsymbol{x}_{ij}\right)\left(n_1^{-1}\sum_{j=1}^{n_1}(\boldsymbol{\beta}_i^\top\boldsymbol{x}_{ij} + \varepsilon_{ij})\boldsymbol{x}_{ij}\right)^\top
$$

$$
= n_1^{-1}\sigma^2\boldsymbol{\Sigma}_F + \boldsymbol{E}_{\boldsymbol{x}}(n_1^{-1}\sum_{j=1}^{n_1}\boldsymbol{x}_{ij}\boldsymbol{x}_{ij}^\top\boldsymbol{\beta}_i)(n_1^{-1}\sum_{j=1}^{n_1}\boldsymbol{x}_{ij}\boldsymbol{x}_{ij}^\top\boldsymbol{\beta}_i)^\top. \tag{C.20}
$$

Now we will study the second term. (C.12) states that $\boldsymbol{E}_{\boldsymbol{x},y,\varepsilon}(\hat{\boldsymbol{b}}_i) = \boldsymbol{\Sigma}_F\boldsymbol{\beta}_i$. And $\hat{\boldsymbol{b}}_i$ is an average of $n_1$ terms, we use the expression of the covariance of sample means to get

$$\mathbf{Cov}(\hat{\boldsymbol{b}}_i) = n_1^{-1}\mathbf{Cov}(\boldsymbol{x}\boldsymbol{x}^\top\boldsymbol{\beta}_i), \tag{C.21}$$

$$\boldsymbol{E}_{\boldsymbol{x},y,\varepsilon}\hat{\boldsymbol{b}}_i\hat{\boldsymbol{b}}_i^\top = \boldsymbol{E}_{\boldsymbol{x}}(n_1^{-1}\sum_{j=1}^{n_1}\boldsymbol{x}_{ij}\boldsymbol{x}_{ij}^\top\boldsymbol{\beta}_i)(n_1^{-1}\sum_{j=1}^{n_1}\boldsymbol{x}_{ij}\boldsymbol{x}_{ij}^\top\boldsymbol{\beta}_i)^\top$$

$$= \boldsymbol{\Sigma}_F\boldsymbol{\beta}_i\boldsymbol{\beta}_i^\top\boldsymbol{\Sigma}_F + n_1^{-1}\mathbf{Cov}(\boldsymbol{x}\boldsymbol{x}^\top\boldsymbol{\beta}_i) \tag{C.22}$$

Now we study $\mathbf{Cov}(\boldsymbol{x}\boldsymbol{x}^\top\boldsymbol{\beta}_i)$.

$$\mathbf{Cov}(\boldsymbol{x}\boldsymbol{x}^\top\boldsymbol{\beta}_i) = \boldsymbol{E}_{\boldsymbol{x}}(\boldsymbol{x}\boldsymbol{x}^\top\boldsymbol{\beta}_i - \boldsymbol{\Sigma}_F\boldsymbol{\beta}_i)(\boldsymbol{x}\boldsymbol{x}^\top\boldsymbol{\beta}_i - \boldsymbol{\Sigma}_F\boldsymbol{\beta}_i)^\top$$

$$= \boldsymbol{E}_{\boldsymbol{x}}(\boldsymbol{x}\boldsymbol{x}^\top\boldsymbol{\beta}_i)(\boldsymbol{x}\boldsymbol{x}^\top\boldsymbol{\beta}_i)^\top - \boldsymbol{\Sigma}_F\boldsymbol{\beta}_i\boldsymbol{\beta}_i^\top\boldsymbol{\Sigma}_F$$

Let $\boldsymbol{x} = \sqrt{\boldsymbol{\Sigma}_F}\boldsymbol{z}$ so that $\boldsymbol{z} \sim \mathcal{N}(0, \boldsymbol{I})$. Let two indices $k, l \in [\mathrm{d}]$. When $k \neq l$,

$$\boldsymbol{E}_{\boldsymbol{x}}[(\boldsymbol{x}\boldsymbol{x}^\top\boldsymbol{\beta}_i)(\boldsymbol{x}\boldsymbol{x}^\top\boldsymbol{\beta}_i)^\top]_{kl} = \boldsymbol{E}_{\boldsymbol{z}}(\sum_{j=1}^d\boldsymbol{\beta}_{i,j}\sigma_j\boldsymbol{z}_j)^2\sigma_k\boldsymbol{z}_k\sigma_l\boldsymbol{z}_l$$

$$= 2\sigma_k^2\sigma_l^2\boldsymbol{\beta}_{i,k}\boldsymbol{\beta}_{i,l}$$

And

$$\boldsymbol{E}_{\boldsymbol{x}}[(\boldsymbol{x}\boldsymbol{x}^\top\boldsymbol{\beta}_i)(\boldsymbol{x}\boldsymbol{x}^\top\boldsymbol{\beta}_i)^\top]_{kk} = \boldsymbol{E}_{\boldsymbol{z}}(\sum_{j=1}^d\boldsymbol{\beta}_{i,j}\sigma_j\boldsymbol{z}_j)^2\sigma_k^2\boldsymbol{z}_k^2$$

$$= \mathbf{tr}(\boldsymbol{\beta}_i^\top\boldsymbol{\Sigma}_F\boldsymbol{\beta}_i)\sigma_k^2 + 2\sigma_k^4\boldsymbol{\beta}_{i,k}^2.$$

So that

$$\boldsymbol{E}_{\boldsymbol{x}}(\boldsymbol{x}\boldsymbol{x}^\top\boldsymbol{\beta}_i)(\boldsymbol{x}\boldsymbol{x}^\top\boldsymbol{\beta}_i)^\top = 2\boldsymbol{\Sigma}_F\boldsymbol{\beta}_i\boldsymbol{\beta}_i^\top\boldsymbol{\Sigma}_F + \mathbf{tr}(\boldsymbol{\beta}_i^\top\boldsymbol{\Sigma}_F\boldsymbol{\beta}_i),$$

$$\mathbf{Cov}(\boldsymbol{x}\boldsymbol{x}^\top\boldsymbol{\beta}_i) = \boldsymbol{E}_{\boldsymbol{x}}(\boldsymbol{x}\boldsymbol{x}^\top\boldsymbol{\beta}_i)(\boldsymbol{x}\boldsymbol{x}^\top\boldsymbol{\beta}_i)^\top - \boldsymbol{\Sigma}_F\boldsymbol{\beta}_i\boldsymbol{\beta}_i^\top\boldsymbol{\Sigma}_F$$

$$= \boldsymbol{\Sigma}_F\boldsymbol{\beta}_i\boldsymbol{\beta}_i^\top\boldsymbol{\Sigma}_F + \mathbf{tr}(\boldsymbol{\beta}_i^\top\boldsymbol{\Sigma}_F\boldsymbol{\beta}_i)\boldsymbol{\Sigma}_F.$$

We plug it back into (C.22) and (C.20) and get

$$\boldsymbol{E}_{\boldsymbol{x},y,\varepsilon}\hat{\boldsymbol{b}}_i\hat{\boldsymbol{b}}_i^\top = \boldsymbol{\Sigma}_F\boldsymbol{\beta}_i\boldsymbol{\beta}_i^\top\boldsymbol{\Sigma}_F + n_1^{-1}(\boldsymbol{\Sigma}_F\boldsymbol{\beta}_i\boldsymbol{\beta}_i^\top\boldsymbol{\Sigma}_F + \mathbf{tr}(\boldsymbol{\beta}_i^\top\boldsymbol{\Sigma}_F\boldsymbol{\beta}_i)\boldsymbol{\Sigma}_F + \sigma^2\boldsymbol{\Sigma}_F).$$

Define $\bar{\boldsymbol{\Sigma}}_T = \frac{1}{T}\sum_{j=1}^T\boldsymbol{\beta}_j\boldsymbol{\beta}_j^\top$. So that

$$\boldsymbol{E}_{\boldsymbol{x},y,\varepsilon}\hat{\mathbf{G}} = \boldsymbol{E}_{\boldsymbol{x},y,\varepsilon}T^{-1}(\sum_{i=1}^T\hat{\boldsymbol{b}}_i\hat{\boldsymbol{b}}_i^\top)$$

$$= \boldsymbol{\Sigma}_F\bar{\boldsymbol{\Sigma}}_T\boldsymbol{\Sigma}_F + n_1^{-1}(\boldsymbol{\Sigma}_F\bar{\boldsymbol{\Sigma}}_T\boldsymbol{\Sigma}_F + \mathbf{tr}(\bar{\boldsymbol{\Sigma}}_T\boldsymbol{\Sigma}_F)\boldsymbol{\Sigma}_F + \sigma^2\boldsymbol{\Sigma}_F) := \bar{\mathbf{G}}.$$

$$\boldsymbol{E}_{\boldsymbol{\beta}}\hat{\mathbf{G}} = \mathbf{G}.$$

We fix all $\boldsymbol{\beta}_i$ and study $\boldsymbol{E}_{\boldsymbol{x},y,\varepsilon}\hat{\mathbf{G}}$. Now we need to show how fast $\hat{\mathbf{G}}$ converges to $\bar{\mathbf{G}}$.

Define

$$\boldsymbol{Z}_i = \hat{\boldsymbol{b}}_i\hat{\boldsymbol{b}}_i^\top - \boldsymbol{E}_{\boldsymbol{x}}(\hat{\boldsymbol{b}}_i\hat{\boldsymbol{b}}_i^\top)$$

$$= (\boldsymbol{\Sigma}_F\boldsymbol{\beta}_i + \boldsymbol{\delta}_i)(\boldsymbol{\Sigma}_F\boldsymbol{\beta}_i + \boldsymbol{\delta}_i)^\top - \boldsymbol{E}_{\boldsymbol{x}}(\boldsymbol{\Sigma}_F\boldsymbol{\beta}_i + \boldsymbol{\delta}_i)(\boldsymbol{\Sigma}_F\boldsymbol{\beta}_i + \boldsymbol{\delta}_i)^\top$$

$$= \boldsymbol{\Sigma}_F\boldsymbol{\beta}_i\boldsymbol{\delta}_i^\top + \boldsymbol{\delta}_i(\boldsymbol{\Sigma}_F\boldsymbol{\beta}_i)^\top + \boldsymbol{\delta}_i\boldsymbol{\delta}_i^\top - \boldsymbol{E}_{\boldsymbol{x}}(\boldsymbol{\Sigma}_F\boldsymbol{\beta}_i\boldsymbol{\delta}_i^\top + \boldsymbol{\delta}_i(\boldsymbol{\Sigma}_F\boldsymbol{\beta}_i)^\top + \boldsymbol{\delta}_i\boldsymbol{\delta}_i^\top).$$

Then

$$\|\boldsymbol{E}\boldsymbol{Z}_i^2\| \leq \|\boldsymbol{E}(\boldsymbol{\Sigma}_F\boldsymbol{\beta}_i\boldsymbol{\delta}_i^\top + \boldsymbol{\delta}_i(\boldsymbol{\Sigma}_F\boldsymbol{\beta}_i)^\top)^2\| + \|\boldsymbol{E}\boldsymbol{\delta}_i\boldsymbol{\delta}_i^\top\boldsymbol{\delta}_i\boldsymbol{\delta}_i^\top\|.$$

Then we can use (C.18) and (C.8) to bound the first term

$$\|\boldsymbol{E}\boldsymbol{Z}_i^2\| \lesssim n_1^{-1}\log^6(N)(\mathbf{tr}(\boldsymbol{\Sigma}_F\boldsymbol{\Sigma}_T) + \sigma)\mathbf{tr}(\boldsymbol{\Sigma}_F)\mathbf{tr}(\boldsymbol{\Sigma}_F^2\boldsymbol{\Sigma}_T)\|\boldsymbol{\Sigma}_F\|^2 + \|\boldsymbol{E}\boldsymbol{\delta}_i\boldsymbol{\delta}_i^\top\boldsymbol{\delta}_i\boldsymbol{\delta}_i^\top\| \tag{C.23}$$

So we need to bound $\|\boldsymbol{E}\boldsymbol{\delta}_i\boldsymbol{\delta}_i^\top\boldsymbol{\delta}_i\boldsymbol{\delta}_i^\top\|$. Note that $\boldsymbol{\delta}_i$ is the average of $\boldsymbol{x}_{ij}(\boldsymbol{x}_{ij}^\top\boldsymbol{\beta}_i + \varepsilon_{ij})$ with respect to index $j = 1, ..., n_1$. So we just let $\boldsymbol{x} \sim \mathcal{N}(0, \boldsymbol{\Sigma}_F)$ and study $\boldsymbol{x}(\boldsymbol{x}^\top\boldsymbol{\beta}_i + \varepsilon_{ij})$. Denote it by $\boldsymbol{u}_i$.

$$\|\boldsymbol{E}_{\boldsymbol{x}}\boldsymbol{u}_i\boldsymbol{u}_i^\top\boldsymbol{u}_i\boldsymbol{u}_i^\top\| = \|\boldsymbol{E}_{\boldsymbol{x}}(\boldsymbol{x}^\top\boldsymbol{\beta}_i + \varepsilon_{ij})^4\boldsymbol{x}\boldsymbol{x}^\top\boldsymbol{x}\boldsymbol{x}^\top\|$$
$$\lesssim \|\boldsymbol{E}_{\boldsymbol{x}}((\boldsymbol{x}^\top\boldsymbol{\beta}_i)^4 + \sigma^4)\boldsymbol{x}\boldsymbol{x}^\top\boldsymbol{x}\boldsymbol{x}^\top\|$$
$$\lesssim (\mathbf{tr}^2(\boldsymbol{\Sigma}_F\boldsymbol{\Sigma}_T) + \sigma^4)\mathbf{tr}(\boldsymbol{\Sigma}_F)\|\boldsymbol{\Sigma}_F\|.$$

So that

$$\|\boldsymbol{E}\boldsymbol{\delta}_i\boldsymbol{\delta}_i^\top\boldsymbol{\delta}_i\boldsymbol{\delta}_i^\top\| \lesssim n_1^{-2}(\mathbf{tr}^2(\boldsymbol{\Sigma}_F\boldsymbol{\Sigma}_T) + \sigma^4)\mathbf{tr}(\boldsymbol{\Sigma}_F)\|\boldsymbol{\Sigma}_F\|.$$

Now we can go back to (C.23) and get

$$\|\boldsymbol{E}\boldsymbol{Z}_i^2\| \lesssim n_1^{-1}\log^6(N)(\mathbf{tr}(\boldsymbol{\Sigma}_F^2\boldsymbol{\Sigma}_T) + \mathbf{tr}(\boldsymbol{\Sigma}_F\boldsymbol{\Sigma}_T) + \sigma^2)^2\mathbf{tr}(\boldsymbol{\Sigma}_F)\|\boldsymbol{\Sigma}_F\|^2.$$

Next we need to bound the norm of $\boldsymbol{Z}_i$. We use (C.18) and (C.8), with probability $1 - N^{-c}$,

$$\|\boldsymbol{Z}_i\| \leq n_1^{-1/2}\log^3(N)(\mathbf{tr}(\boldsymbol{\Sigma}_F^2\boldsymbol{\Sigma}_T) + \mathbf{tr}(\boldsymbol{\Sigma}_F\boldsymbol{\Sigma}_T) + \sigma^2)\sqrt{\mathbf{tr}(\boldsymbol{\Sigma}_F)}\|\boldsymbol{\Sigma}_F\| + n_1^{-1}\log^5(N)(\mathbf{tr}(\boldsymbol{\Sigma}_F\boldsymbol{\Sigma}_T) + \sigma^2)\mathbf{tr}(\boldsymbol{\Sigma}_F).$$

Define the upper bound for $\|\boldsymbol{E}\boldsymbol{Z}_i^2\|, \|\boldsymbol{Z}_i\|$ as $Z_1, Z_2$ (the right hand side of two above inequalities). With Bernstein type inequality (Lemma 2), with probability $1 - N^{-c}$,

$$\|\hat{\mathbf{G}} - \bar{\mathbf{G}}\|$$
$$= \|T^{-1}\sum_{i=1}^{T}\boldsymbol{Z}_i - \boldsymbol{E}_{\boldsymbol{x}}\boldsymbol{Z}_i\|$$
$$\lesssim \log(TZ_2)\left(T^{-1/2}\log(N)Z_1^{1/2} + T^{-1}Z_2\log(TZ_2)\right)$$
$$\lesssim \log(TZ_2)\Bigg(\sqrt{\frac{\log^6(N)(\mathbf{tr}(\boldsymbol{\Sigma}_F^2\boldsymbol{\Sigma}_T) + \mathbf{tr}(\boldsymbol{\Sigma}_F\boldsymbol{\Sigma}_T) + \sigma^2)^2\mathbf{tr}(\boldsymbol{\Sigma}_F)\|\boldsymbol{\Sigma}_F\|^2}{n_1 T}}$$
$$+ \frac{\log^3(N)(\mathbf{tr}(\boldsymbol{\Sigma}_F^2\boldsymbol{\Sigma}_T) + \mathbf{tr}(\boldsymbol{\Sigma}_F\boldsymbol{\Sigma}_T) + \sigma^2)\sqrt{\mathbf{tr}(\boldsymbol{\Sigma}_F)}\|\boldsymbol{\Sigma}_F\|}{n_1^{1/2}T}$$
$$+ \frac{\log^5(N)(\mathbf{tr}(\boldsymbol{\Sigma}_F\boldsymbol{\Sigma}_T) + \sigma^2)\mathbf{tr}(\boldsymbol{\Sigma}_F)}{T}\Bigg)$$
$$= \log(TZ_2) \cdot \Bigg(\log^3(N)\|\boldsymbol{\Sigma}_F\|(\mathbf{tr}(\boldsymbol{\Sigma}_F^2\boldsymbol{\Sigma}_T) + \mathbf{tr}(\boldsymbol{\Sigma}_F\boldsymbol{\Sigma}_T) + \sigma^2)\sqrt{\frac{\mathbf{tr}(\boldsymbol{\Sigma}_F)}{N}}$$
$$+ \frac{\log^5(N)(\mathbf{tr}(\boldsymbol{\Sigma}_F^2\boldsymbol{\Sigma}_T) + \mathbf{tr}(\boldsymbol{\Sigma}_F\boldsymbol{\Sigma}_T) + \sigma^2)\sqrt{\mathbf{tr}(\boldsymbol{\Sigma}_F)}\|\boldsymbol{\Sigma}_F\|}{N^{1/2}T^{1/2}}\Bigg).$$

∎

# D  Proof of Robustness of Optimal Representation

**Theorem 3** *Suppose the data is generated as Phase 2, $\boldsymbol{\Lambda}$ and $\underline{\theta}$ are defined in Def. 1 and the estimated task is obtained as (3). Let the upper bound of $\|\hat{\boldsymbol{M}} - \boldsymbol{M}\|$ be $\mathcal{E}$. The risk of meta-learning algorithm satisfies*

$$risk(\boldsymbol{\Lambda}_{\underline{\theta}}(R), \boldsymbol{\Sigma}_T, \boldsymbol{\Sigma}_F) - risk(\boldsymbol{\Lambda}_{\underline{\theta}}^*(R), \boldsymbol{\Sigma}_T, \boldsymbol{\Sigma}_F) \lesssim \frac{n_2^2 \cdot \mathcal{E}}{d(R - n_2)(2n_2 - R\underline{\theta})\underline{\theta}}.$$

**Proof** In the proof below, we use $\boldsymbol{\Lambda}$ and $\boldsymbol{\Lambda}^*$ to replace $\boldsymbol{\Lambda}_{\underline{\theta}}(R), \boldsymbol{\Lambda}_{\underline{\theta}}^*(R)$ for simplicity. We first decompose the risk as

$$\text{risk}(\boldsymbol{\Lambda}, \boldsymbol{\Sigma}_T, \boldsymbol{\Sigma}_F) - \text{risk}(\boldsymbol{\Lambda}^*, \boldsymbol{\Sigma}_T, \boldsymbol{\Sigma}_F)$$
$$= \underbrace{\text{risk}(\boldsymbol{\Lambda}, \hat{\boldsymbol{\Sigma}}_T, \boldsymbol{\Sigma}_F) - \text{risk}(\boldsymbol{\Lambda}^*, \hat{\boldsymbol{\Sigma}}_T, \boldsymbol{\Sigma}_F)}_{\leq 0}$$
$$+ [\text{risk}(\boldsymbol{\Lambda}, \boldsymbol{\Sigma}_T, \boldsymbol{\Sigma}_F) - \text{risk}(\boldsymbol{\Lambda}, \hat{\boldsymbol{\Sigma}}_T, \boldsymbol{\Sigma}_F)] + [\text{risk}(\boldsymbol{\Lambda}^*, \hat{\boldsymbol{\Sigma}}_T, \boldsymbol{\Sigma}_F) - \text{risk}(\boldsymbol{\Lambda}^*, \boldsymbol{\Sigma}_T, \boldsymbol{\Sigma}_F)].$$

We know $\text{risk}(\mathbf{\Lambda}, \hat{\mathbf{\Sigma}}_T, \mathbf{\Sigma}_F) - \text{risk}(\mathbf{\Lambda}^*, \hat{\mathbf{\Sigma}}_T, \mathbf{\Sigma}_F) \leq 0$ due to the optimality of $\mathbf{\Lambda}$ with task covariance $\hat{\mathbf{\Sigma}}_T$. Now we will bound $\text{risk}(\mathbf{\Lambda}, \mathbf{\Sigma}_T, \mathbf{\Sigma}_F) - \text{risk}(\mathbf{\Lambda}, \hat{\mathbf{\Sigma}}_T, \mathbf{\Sigma}_F)$ for arbitrary $\mathbf{\Lambda}$, and it automatically works for $\text{risk}(\mathbf{\Lambda}^*, \hat{\mathbf{\Sigma}}_T, \mathbf{\Sigma}_F) - \text{risk}(\mathbf{\Lambda}^*, \mathbf{\Sigma}_T, \mathbf{\Sigma}_F)$. Note that in (3.3) we know that

$$\text{risk}(\mathbf{\Lambda}', \mathbf{\Sigma}_T') = f(\boldsymbol{\theta}; \mathbf{\Sigma}_T, \mathbf{\Sigma}_F) := \sum_{i=1}^{R} \frac{n_2(1 - \boldsymbol{\theta}_i)^2}{R(n_2 - \|\boldsymbol{\theta}\|^2)} \tilde{\mathbf{\Sigma}}_{T,i}^R + \frac{n_2}{n_2 - \|\boldsymbol{\theta}\|^2} \sigma^2. \tag{D.1}$$

This function is linear in $\mathbf{\Sigma}_T$ thus we know that

$$|\text{risk}(\mathbf{\Lambda}^*, \hat{\mathbf{\Sigma}}_T, \mathbf{\Sigma}_F) - \text{risk}(\mathbf{\Lambda}^*, \mathbf{\Sigma}_T, \mathbf{\Sigma}_F)| \leq \frac{n_2}{d(n_2 - \|\boldsymbol{\theta}\|^2)} \mathcal{E}. \tag{D.2}$$

Now we need to bound $\|\boldsymbol{\theta}\|^2$. With the constraint $\underline{\theta} \leq \boldsymbol{\theta} < 1 - \frac{R-n_2}{n_2}\underline{\theta}$ and $\sum \boldsymbol{\theta}_i = n_2$, we know that the maximum of $\|\boldsymbol{\theta}\|^2$ happens when $(R - n_2)$ among $\boldsymbol{\theta}_i$ are $\underline{\theta}$ and the others are $1 - \frac{R-n_2}{n_2}\underline{\theta}$. With this we have

$$\|\boldsymbol{\theta}\|^2 \leq (R - n_2)\underline{\theta}^2 + n_2(1 - \frac{R - n_2}{n_2}\underline{\theta})^2$$

$$= (R - n_2)\underline{\theta}^2 + n_2 - 2(R - n_2)\underline{\theta} + \frac{(R - n_2)^2}{n_2}\underline{\theta}^2$$

$$= n_2 - 2(R - n_2)\underline{\theta} + \frac{(R - n_2)R}{n_2}\underline{\theta}^2$$

Thus

$$n_2 - \|\boldsymbol{\theta}\|^2 \geq (R - n_2)\underline{\theta}(2n_2 - R\underline{\theta}).$$

Plugging it into (D.2) and (D.1) leads to the theorem. ∎