# OpenReview forum: "Towards Sample-efficient Overparameterized Meta-learning"
_NeurIPS.cc/2021/Conference — NeurIPS 2021 Poster_

### Official Review · Reviewer_kYDu · 2021-07-13

**Rating:** 6
**Confidence:** 3

**Summary:**

This paper considers the representation learning for multi-task linear regression problem in the setting with arbitrary feature covariance, arbitrary task covariance, and arbitrary representation dimension. In this setting, it aims to address two questions: (1) given a set of $T$ linear regression tasks with $N$ total samples, what is the optimal representation $\Lambda \in \mathbb{R}^{d\times R}$ for a downstream few-shot linear regression task? and (2) what is the sample complexity for finding this representation? To address these questions, the authors find a closed-form solution for the optimal representation when the fine-tuning phase involves an eigen-weighting procedure, which they argue introduces an inductive bias that enables few-shot success. They show that the key to estimating this optimal representation is estimating the "canonical task covariance", which they show necessitates a number of samples growing with the effective ranks (traces) of the task and feature covariances, rather than the ambient dimension $d$. The authors also provide an overall risk bound that shows it is preferable to have smaller $R$ for smaller $N$.


**Limitations And Societal Impact:**

Yes.

**Main Review:**

Strengths:

1. The introduction is mostly clear and does a great job of motivating the linear setting.

2. To the best of my knowledge, arbitrary task covariance had not yet been considered for linear multi-task representation learning, so the dependence of the excess risk on the canonical task covariance is a new result.

3. Showing that the sample complexity grows with the effective ranks of the covariance matrices rather than the ambient dimension is a nice result.

4. The analysis is rigorous and certainly non-trivial. The proofs are correct as far as I could tell.

5. Understanding over-parameterization in representation learning is a highly relevant problem.

Weaknesses/Questions:

1. After the introduction I found this paper very difficult to understand, which leaves me with numerous questions about its contributions (see below). The writing can be significantly improved, both in terms of fixing typos and more importantly, giving more simple and clear explanations of the results. I found the interpretations of Theorems 1 and 3 especially lacking, since they involve highly non-intuitive expressions yet very little explanation was given of them.

2. Regarding Theorem 1 and eigen-weighting: what is the point of the eigen-weighting? How do the results compare to the case in which standard linear regression is executed in the few-shot phase instead? Do any insights from eigen-weighting generalize beyond linear models? I get that eigen-weighting is itself an inductive bias, but I am not sure how this inductive bias helps... right now it seems to only complicate the problem, making the optimal representation (in Theorem 1) non-intuitive and only computable in closed form under that restrictive assumptions that $n_2 \rightarrow \infty$ and the feature and task covariances are simultaneously diagonalizable.

3. Regarding Theorem 2 and surrounding text: $s_F$ (and $s_T$) should be defined outside of the Theorem 2 so that it is clear what they are when you refer to them multiple times outside of the theorem. There should also be some accompanying intuition for this definition. It is also not clear why the relationships between $r_F$ and $s_F$, and $r_T$ and $s_T$ hold in Table 2.  Further, you have introduced $\iota$ in Example 1, but it does not appear in the bound which does not make sense.

4. Regarding Theorem 3: how does $\theta_{-}$ scale, and how can $\Lambda_{\theta_{-}}(R)$ be computed (do you compute the representation using the constraint that its minimum singular value is lower bounded by $\theta_{-}$, or is $\theta_{-}$ evaluated after the fact)? It is not clear that "representation learning error increases with $R$" from the result. Also, the risk becomes negative when $R > n_2$, is there a missing assumption?

5. I am doubtful of the novelty of many of the contributions of this paper compared to [1], which also provides bounds for the linear representation learning + few-shot learning problem in the over-parameterized setting with arbitrary feature covariance, and obtain a bound that depends on effective rank, not dimension (see Theorem 6). The authors do not mention this result in the current work.

6. It is repeatedly stated that [2] gives a sample complexity of $dr^2$, but this is only true under the assumption that the minimum singular value of the task covariance matrix is $O(1/r)$. If it is $O(1)$, then [2] achieves $dr$. The authors need to mention this in the update and do a fair comparison with clear assumptions on the task covariance matrix.

Overall: This paper is promising, but at this point I have too many questions and concerns to recommend acceptance.

[1] Du et al. Few-Shot Learning via Learning the Representation, Provably. 2020.
[2] Tripuraneni et al. Provable Meta-Learning of Linear Representations. 2020.

----------------------

Post-rebuttal: I thank the authors for their thorough response. They have helped me to better understand the results of the paper, and also resolved my concerns about comparisons with [1] and [2]. I believe this is an important result, but the clarity of the paper can be significantly improved in order for the result to be fully appreciated. I have raised my score to a 6.

**Time Spent Reviewing:**

8

---

> ### Author Response · Authors · 2021-08-10
> **Response to Reviewer kYDu**
>
> We appreciate your thorough review and feedback. Below, we sequentially address each of your concerns.
>
> **Point 1: Theorem 1**
>
> (Covered by the response to Reviewer gmou)
>
> First, we discuss the role of Theorem 1 in the paper.  To achieve the overarching goal of characterizing few-shot risk, the simpler case of Section 3 serves as a crucial stepping stone. In order to solve the optimization (3.1), the strategy we adopt in Section 3 is to first show it is equivalent to a different optimization, whose solution can then be obtained by simply analyzing KKT optimality conditions. Theorem 1 characterizes this equivalence.
>
> While there is no closed-form expression for $\Lambda^\ast$, we can prove that it is optimal in both over-parameterized and under-parameterized regimes. In the overparameterized regime, if the optimal representation matrix is applied, the error decreases with increasing representation dimension. Reviewer PuM5 agrees with that "The paper characterizes the optimal representation for every dimensionality R (in an asymptotic regime), observes a double descent phenomenon w.r.t. R in the performance of the representation on test tasks, also concludes that the optimal dimensionality can be overparametrized in many cases." We can compute and plot the risk in Fig. 1b, 3 that shows the double descent phenomena.
>
> Thm. 1 studies the asymptotic case where the dimensions tend to infinity. The dots in Fig. 1b are experimental values for finite dimensional case and the curves are the theoretical value for infinite dimensional case, one can see they match perfectly.
>
> **Point 1 and 4: Theorem 3**
>
> Theorem 3 does not require an explicit computation of the optimal representation by enforcing $\underline{\theta}$. Instead, we use the robustness of such a representation (due to its well-conditioned nature) to deduce its stability. That said, for practical computation of optimal representation, we simply use Algorithm 1. We can then evaluate $\underline{\theta}$ after-the-fact as the minimum singular value of this representation to apply Theorem 3 without assuming an explicit $\underline{\theta}$. The risk (3.3) only works for $R>n_2$, the expression for $R<n_2$ is different, covered by cited work [38] eq (4.4) the second case. It is simple, so we do not discuss it but plot it in Fig. 1b on the left side of the peak as a comparison. We will add a remark to the paper to make this clear.
>
> **Point 2: The importance of eigenweighting**
>
> Since we allow $\Lambda$ to be an arbitrary $d \times R$ matrix, decomposing it using the SVD, we can interpret the transformation $x \to \Lambda x$ as  (1) projecting $x$ onto a subspace of size $R$ and (2) reweighting the contributions of each of the basis elements of the subspace to the projection. We name the latter as eigenweighting. This weighting procedure is extremely crucial to avail the benefit of overparameterization. As you mention, it captures an inductive bias that promotes certain features and demotes others; possessing the right inductive bias has been shown to be tantamount to "benign overfitting" - for instance, see the description of Occam's razor in the seminal work [7] or the analysis in [27, 33] that rests heavily on the bilevel weighting scheme that they define. Figure 1b and Figure 3 show that the risk corresponding to optimal representation/eigen-weighting is smaller than no-weighting case in overparameterized regime. Reviewer PuM5 expresses the importance of benign overfitting "This paper meaningfully extends the linear representation learning by studying the role and need for high dimensionality of representations ...".
> Consider the following example of when min-L2 interpolation would fail but eigenweighted min-L2 interpolation could succeed. Let the true coefficients be 1-sparse and let $d \gg n$; without loss of generality, the labels equal the first feature exactly and all other $d-1$ features are spurious. If we run naive min-L2 interpolation, then the learned model would spread weight across all features and place a nontrivial weight on the spurious features; this would lead to poor generalization performance. However, if there was an eigenweighting that strongly promoted the first feature, this pitfall could be avoided. In the more general nonlinear case, the weighting procedure could be applied to the lifted-features. Eigenweighting is one way to encode an inductive bias, and elucidates the importance of the right inductive bias.
>
> **Point 3: Theorem 2**
>
> Thm. 2 is the result for arbitrary covariances, and Example 1 gives a special example with a bi-level covariance that uses the notation $\iota$ only for this example. The numbers in Tab. 2 are obtained by directly inserting the specific covariance instances into the bound of Thm. 2.
>
> **Key distinctions from paper [1]**
>
> Thanks for helping us take a closer look to [1]. Indeed, [1] does discuss the overparameterized representation and give a bound depending on effective rank. We will discuss this in the revision. In essence, [1] requires at least $O(d)$ samples per source task which is unrealistic compared to our $O(1)$ or $O(r)$ requirements. With $O(d)$ samples, note that, one can solve each task individually without representation. [1] also never shows that overparameterization helps. Instead, we prove that, with enough source samples, you can provably benefit from overparameterized representations. More precisely, the differences are as follows:
>
> (1) There is a term $\zeta$ in the error called "lower-order", but it is not always small, for example, when $T$ is large and $n_1\ll d$, this low-order term is large. [1] needs $n_1>d$ essentially, a much stronger assumption than our $n_1=O(1)$ or $O(r)$.
>
> (2) [1] uses norm-based concentration to bound the error but we don't know if it is an optimal bound. We propose a way of computing the exact asymptotic risk and Fig. 1b shows that it is very accurate in finite dimensional case (The dots in Fig. 1b are experimental values for finite dimensional case and the curves are the theoretical value for infinite dimensional case). It is known that exact result is usually far better than norm-based bound.
>
> (3) We show that overparameterization beats the underparameterized representation by that we compute the exact optimal representation and that the optimal representation is usually overparameterized. This is illustrated in Fig. 1b, 3. [1] does not show the benefit of overparameterization.
>
> (4) [1] uses regularization to avoid overfit, whereas we show the overparameterized interpolation does not overfit.
>
> **Optimality discussion and comparison to [2]**
>
> Short response: Their $\nu$ is at most $O(1/r)$. Both published (ICML) version and the latest arXiv version of [2] clearly state that their sample complexity bound is $O(dr^2)$. We also noticed that [Du et al. ICLR'21] also proves $O(dr)$ result however their result makes the unrealistic assumption of each task having $O(d)$ samples. This greatly simplifies the problem as each task can now be solved individually.
>
> Detailed response: We will use the notations in [2] to be consistent with the context. We compute $\nu$ which is the $r$th singular value of task average matrix $A^TA/t$. Note that [2] assume all tasks are "unit length", so one can prove that $\nu$ cannot be as large as 1 (cannot be larger than $1/r$), otherwise the trace of $A^TA/t$ is at least sum of all singular values which is lager than $r\nu=r$. But the average of rank one matrices $\alpha\alpha^T$ has trace $1$ due to "unit length". If one does not assume all tasks are "unit length", then the bounds in [2] will scale. A simple scaling does not change the spirit of the result of [2].
>
> With our model, $t$ is larger than $r$, we can check the similar setting where task covariance is $I_r/r$ so each task is approximately unit length. And, $A^TA/t$ will converge to the covariance so $\nu=1/r$.

---

### Official Review · Reviewer_gmou · 2021-07-15

**Rating:** 5
**Confidence:** 2

**Summary:**

This paper studies the role of overparameterization in meta representation learning.

Meta-learning studies how to learn from previous tasks to adapt quickly to new ones, while overparameterization studies the settings where the number of training points is smaller than the number of features. Recent works studied the generalization properties of meta-learning a linear representation but did not focus on the overparameterized setting, where the number of training samples per task is lower than the dimension of the meta-learned representation.

This work presents asymptotic results on the expected risk in the overparameterized regime. Furthermore they propose an algorithm to learn a linear representation together with sample complexity bounds. They also provide some empirical results validating their analysis.


**Limitations And Societal Impact:**

The authors discuss limitations in the conclusion section. I can not tell if the list is exaustive.

The work does not have potential negative societal impact since it is mostly theoretical.


**Main Review:**

I anticipate that I am not very familiar with theoretical results in meta representation learning and overparameterization. I apologize if I misunderstood crucial parts of the work and in that case I am willing to increase the score after the discussion.

1. I found the paper hard to read. Some crucial symbols are introduced without definition or are inconsistent. Theorems and algorithms are poorly explained. Contributions and most of the experiments are also unclear.

2. I do not think Theorem 1 (crucial part of the work) is meaningful. It is also not explained well. Usually an upper bound on the risk is provided and then limits are taken. Theorem 1 seems to skip the first part. Furthermore, with fixed $\kappa = \frac{R}{n_{2}}$ . if you take the limit for $n_2 \to \infty$ in 3.4  also $R \to \infty$  and the left hand side of (3.4) is 0 if the risk is bounded. Also it seems that the matrices will have infinite rows and columns in the limit.

3. The authors do not mention [A, B] which also provide learning guarantees with meta-learning of a linear representation.

#### **Final Score**
4. I am giving this paper a 4. I think the problem that the authors are tackling is potentially interesting. Overparameterization probably plays a big role also in meta-learning. However I think the quality of the work is far from being sufficient for acceptance.

#### **Additional comments**

5. Close to line 37: how is training on cifar 10 in Figure 1(a) done? only the last layer is trained or all the weights are fine-tuned? I would write this here.

6. line 62: I think saying that the linear representation mapping throws away features is wrong. It just weighs them.

7. I found the contributions section quite heavy in terms of loosely defined notation and references to future sections.

8. Line 244-247 are not clear

9. Line 52: I think the term “underparametrized” is used in the wrong way here. The model the related work studies has input dimension higher than feature dimension $d < R$ while $R$ could still be (and often is) higher than the number of samples per task $n_2$, which would make it overparameterized in the sense meant by the authors.

10. Observation 2 is not proved anywhere (not even in the supplementary).

#### **Update**
After the discussing with authors and reviewers I better understand the purpose of Theorem 1. However I am still unsure on the significance of the presented results and I still think the paper lacks clarity. I increased my score from 4 to 5.

#### **References**

[A] G. Denevi, C. Ciliberto, D. Stamos, and M. Pontil. Incremental learning-to-learn with statistical guarantees. In Proc. 34th Conference on Uncertainty in Artificial Intelligence (UAI), 2018.

[B] G. Denevi, D. Stamos, C. Ciliberto, and M. Pontil. Online-within-online meta-learning. In Advances in Neural Information Processing Systems, pages 13089–13099, 2019.


**Time Spent Reviewing:**

8

---

> ### Author Response · Authors · 2021-08-10
> **Response to Reviewer gmou**
>
> Thank you for taking the time to review our paper and for the constructive feedback. In our response below, we attempt to clarify aspects of the writing were raised in your review as confusing. We especially hope to illuminate the message of Theorem 1 and it's role in the paper.
>
> ## Response to point 1: Outline of the results
>
> We discuss here the organization of the paper and the significance of each of the theoretical results to alleviate confusions and misunderstandings that were brought up, and to set the context for responses to other concerns. Recall that we apply a linear representation $\Lambda$ to a feature-vector by transforming $x \to \Lambda x$. Using a linear-representation will influence which vector of coefficients is chosen by the min-L2 interpolator, and its subsequent test risk. The overarching goal of the paper is to propose a choice of $\Lambda$ that achieves low risk on a downstream (few-shot) task for linear representation-learning after having already observed $T$ related tasks. This goal is important because it allows us to engage with important questions regarding overparameterization:
> (1) what should the size $R$ of the resultant representation be? This answers the question whether overparameterization is beneficial in few-shot learning.
> (2) What subspace should $\Lambda$ project the features onto?
> (3) What weight should it place on each of these basis directions?
>
> First, section 3 tackles a simpler version of this primary goal: to determine the choice $\Lambda^\ast$ that would lead to the min-L2 interpolator with minimum risk when the task covariance $\Sigma_T$ and feature covariance $\Sigma_F$ are exactly known. Hence, we frame problem (3.1).
>
> In practice, task and feature covariances are rarely known apriori. However, in a meta-learning setup, these can be estimated from previously seen tasks as $\hat{\Sigma}_F$ and $\hat{\Sigma}_T$. In Section 4, we propose and analyze such an estimation scheme. In Section 5, we consider the generalization of Section 3, where we have only estimates of the covariances instead of perfect knowledge; thus, we are able to characterize few-shot risk and tackle our overarching goal.
>
> Please let us know if this helps to clarify the significance of each of the results. Or if there are specific aspects of the paper not addressed below that you are confused by that we could discuss further.
>
> ## Response to Point 2: Clarification of Theorem 1
>
> First, we discuss the role of Theorem 1 in the paper.  To achieve the overarching goal of characterizing few-shot risk, the simpler case of Section 3 serves as a crucial stepping stone. In order to solve the optimization (3.1), the strategy we adopt in Section 3 is to first show it is equivalent to a different optimization, whose solution can then be obtained by simply analyzing KKT optimality conditions. Theorem~1 characterizes this equivalence.
>
> While there is no closed-form expression for $\Lambda^\ast$, we can prove that it is optimal in both over-parameterized and under-parameterized regimes. In the overparameterized regime, if the optimal representation matrix is applied, the error decreases with increasing representation dimension. Reviewer PuM5 agrees with that "The paper characterizes the optimal representation for every dimensionality R (in an asymptotic regime), observes a double descent phenomenon w.r.t. R in the performance of the representation on test tasks, also concludes that the optimal dimensionality can be overparametrized in many cases." We can compute and plot the risk in Fig. 1b, 3 that shows the double descent phenomena.
>
> Theorem 1 calculates the exact asymptotic risk. That is, risk is both upper and lower bounded by this quantity asymptotically where convergence is in probability. The upper and lower bounds are implicit in the proof. As $R\rightarrow 0$, the $risk/R$ term (right hand-side) converges to a scalar. The reason is that $risk$ grows proportional to $O(R)$. This can be seen from the risk definition at Eq(3.2) and noticing that $E[\|\beta\|_2^2]=O(R)$. The right hand side of (3.4) is not 0 because the risk increases with $R$ (as the risk depends on the inner product of feature and error in tasks, the averaged magnitude of each entry will converge so the inner product increases linearly with the dimension), and they are calculated and plotted in Fig. 1b, 3. Thm. 1 studies the asymptotic case where the dimensions tend to infinity, and we use Fig. 1b, 3 to show that the benefit of overparameterization + optimal representation matrix all behave as expected in finite dimensional scenario. The dots in Fig. 1b are experimental values for finite dimensional case and the curves are the theoretical value for infinite dimensional case, one can see they match perfectly.
>
> ## Response to point 3: Key differences from works [A,B]
>
> Thank you for pointing us to these relevant papers from the literature. [A] is more closely relevant to our result. Below, we summarize the crucial distinctions of our work from [A].
>
> Work [A] uses the weighting matrix as the representation and studies the generalization risk, similar to ours. However,
>
> (1) [A] uses Rademacher complexity to bound the generalization risk. We propose a way of computing the exact asymptotic risk and Fig. 1 shows that it is very accurate in the finite dimensional case. Recent literature on overparameterization highlights that classical uniform convergence techniques from learning theory are unable to explain the double-descent phenomenon.
>
> (2) [A] does not show the benefit with respect to a small effective rank of either feature or task, different from the idea of subspace based meta-learning.
>
> (3) We show that overparameterization outperforms the underparameterized representation because the optimal representation is usually overparameterized, whereas [A] does not show the benefit of overparameterization. The benefit of overparameterization is important, as suggested by Review PuM5 "This paper meaningfully extends the linear representation learning by studying the role and need for high dimensionality of representations."
>
> (4) [A] uses regularization to avoid overfitting and the risk increases when $\lambda$ decreases, thus one cannot choose a small $\lambda$, whereas we show the overparameterized interpolation does not overfit.
>
> ## Responses to Additional Comments
>
> **Point 5**
> We only train the last layer of the network. We treat the output of the prior layers to the last layer as a fixed feature-map. Hence, the algorithm can be interpreted as a generalized linear model, or kernel linear regression.
>
> **Point 6**
> The $\Lambda$ considered in the paper is a general $d \times R$ matrix, and is not restricted to be a $d \times d$ diagonal matrix; if the latter were the case, then it would be a weighting matrix as you mention. If $R<d$, then a subspace of dimension $d-R$ is neglected. The non-zero numbers of $\Lambda$ have the effect of weighting.
>
> **Point 8**
> For the MNIST instance, we compute the covariance of features and tasks, and we reach the conclusion that the two covariances have small angle, which means their principal subspaces align. The benefit caused by this alignment is reflected in Thm. 2 and illustrated in Tab. 2 and Fig. 4. Reviewer PuM5 comments "An important phenomenon of alignment of feature covariance and task covariance is introduced and its effect on sample complexity and optimal dimensionality is studied."
>
> **Point 9**
> We use the term "underparameterized" when $R<n_2$ (left of the peak of Fig. 1b) and "overparameterized" when $R>n_2$ (right of the peak of. Fig. 1b).
>
> **Point 10**
> Observation 2 follows directly from the full expression of risk in (3.2) and the discussion between (3.2) and the observation. We will be sure to emphasize this in the text.

---

> > ### Comment · Reviewer_gmou · 2021-08-17
> > **On Theorem 1**
> >
> > Thanks for the detailed response. I now understand sightly better the intent of Theorem 1. However, after taking another look at it, I still have major concerns.
> >
> > 1. Does the risk in RHS of 3.4 really grow with $R$? This would mean that in the considered limit ($n_2, R \to \infty$) the risk is infinity, which seems odd: why is the limit useful in this case? If the risk instead remains bounded, then the right hand side of 3.4 is always 0, which was my previous concern.
> >
> > 2. I still find the theorem hard to understand. I could not find a clear proof  in the appendix and several typos seem to be present: why the superscript $*$ is not used in the theorem but it is used in algorithm 1? who is $p$? are $\boldsymbol{\theta}$ and $\boldsymbol{\theta}^\star$  different (the first seems to be a function of the second through the definition of $\boldsymbol{\Lambda}$)?
> >
> > 3. The fact that also $R \to \infty$ when $n_2 \to \infty$ ($R/n_2$ is fixed) is quite odd and seems to be hidden and not discussed properly. What happens when $R > d$ to $\boldsymbol{\Sigma}^R_{T}$ and $\boldsymbol{\Sigma}^R_{F}$ in algorithm 1? Have
> > they $R-d$ eigenvalues set to 0? What about $\boldsymbol{\Sigma}_{F}^{\perp}$, $ U_1$ and $ \boldsymbol{\Lambda}^{*}$?

---

> > > ### Author Response · Authors · 2021-08-17
> > > **Response on Theorem 1**
> > >
> > > Thanks for giving us a chance to clarify. We address your questions below. Please let us know if further clarification is needed.
> > >
> > > **Q1. Risk definition:** Short answer:  RHS of (3.4) converges to a constant.
> > >
> > > The risk grows with $R$ and the risk is infinity when $n_2,R$ are infinity, thus the risk itself is not useful. Therefore, the RHS of (3.4) *normalizes* the risk by $R$, and since risk grows linearly with $R$, RHS converges to a constant. The constant is a meaningful quantity.
> > >
> > > In summary, our current setting is $\text{risk}\rightarrow\infty$, $\text{risk}/R\rightarrow C$. Right hand side of (3.4) is equal to the latter.
> > >
> > > At a high-level we appreciate the reviewer's concern that risk is typically normalized and finite. We are happy to revise the paper to normalize the risk definition.
> > >
> > > **Q2. Understanding theorem:** The superscript * denotes the optimal solutions. In the algorithm, we are concerned with solving optimization (3.1) in the variable $\Lambda$, and $\Lambda^\star$ is the *optimal* representation. $\theta$ is a change of variable of $\Lambda$ and $\theta^\star$ is a change of variable of $\Lambda^\star$. However, in Theorem 1, the risk calculation is valid for *arbitrary* (including but not restricted to optimal) representation, thus we omitted *.
> > >
> > > In summary, $[\theta, \Lambda]$ pair and $[\theta^\star, \Lambda^\star]$ pair are linked in the same way, however $\theta^\star$ and $\Lambda^\star$ are the minimizers of $f$ and $risk/R$ respectively. We will better clarify this.
> > >
> > > $p$ should be $R$. This is a typo that we also replied to Reviewer PuM5.
> > >
> > > For part of the asymptotic analysis, we directly quote the results of [Thrampoulidis et al]. Line 553 of our supplementary refers to [35] and we did not repeat the proof.
> > >
> > > **Q3. Infinite dimensions:** First, we clarify the setup that $n_2, R, d$ all tend to infinity, with $n_2/d$ and $R/d$ being fixed (we will edit the theorem which only mentions $n_2$ and $R$). Such settings (sample size and dimension grows proportionally) are common in high-dimensional analysis to accurately capture the role of the finite samples. This includes classical works on random matrix theory (e.g. Bai-Yin theorem) to very recent works that analyze asymptotics of random feature regression to shed light on deep networks (see works by Montanari, Mahoney and others). In summary, asymptotic analysis is simply a proxy for analyzing the finite dimensional behavior and often accurately captures the finite dimensional behavior as soon as dimension is in the order of hundreds. In fact, Reviewer PuM5 asked a related question and we responded "The dots in Fig. 1b are experimental values for finite dimensional data and the curves are the theoretical value for infinite dimensional case, one can see they match perfectly".
> > >
> > > Secondly, as we focus on subspace based meta-learning, we never choose $R>d$, because $R=d$ already captures all features and we don't need overlapping features. When we say arbitrary $R$ we mean $0<R\le d$. The underparameterized case is $R\le n_2<d$ and the overparameterized case is $n_2\le R\le d$, which means we compare number of samples $n_2$ with representation dimension $R$. If $R=d$, $\Sigma_F^{\perp}$ does not exist and can be seen as a $0$-dim space. Our proof does not change but we just set the residual $\boldsymbol{\beta}_R^{\perp}$ to $0$.

---

### Official Review · Reviewer_czkt · 2021-07-17

**Rating:** 5
**Confidence:** 3

**Summary:**

This paper presents an analysis of the role of representation width when learning a shared feature extractor across multiple tasks. The authors argue that wider representations improve transfer, even when only a small number of labelled examples are accessible to fine-tune the last linear layer of a neural network. Their theoretical analysis focuses on linear regression with a linear feature extractor, with most experiments also on this setting.

**Limitations And Societal Impact:**

1 paragraph on limitations, none on societal impact.

**Main Review:**

Strengths:

The message of the paper is interesting and relevant to the few-shot learning community. In particular, a thorough study of width in meta-learning is novel and welcome.

Weaknesses:

- Simplified analytical setting: the authors only consider linear regression with a shared, linear feature extractor. Although linear regression has been used in prior work, some more recent theoretical works have shifted towards the more general convex (and sometimes non-convex) setting. See for example [1, 2]. Since linear regression induces a quadratic optimization problem, I am concerned that the proposed approach won't be easily extended to those settings.
- Lack of real-world validation: On top of the simplified analytical setting, theoretical insights are never validated on real-world datasets. Hence, I wonder if those insights carry over to real-world applications. Taking classification tasks as an example, it is well-known that nearest-neighbor classifiers (like ProtoNet) tend to work better with lower-dimensional representations, while SVMs work best with larger representations.
- As a minor remark, I think this paper would benefit from a reference to R2D2 [3] who obtain excellent classification results with a deep feature extractor and ridge regression classifier (but don't say anything about representation width). Their method could also be used to address the lack of real-world validation.

References:

[1] "A Theoretical Analysis of the Number of Shots in Few-Shot Learning", Cao et al., ICLR 2020.

[2] "On the Convergence Theory of Gradient-Based Model-Agnostic Meta-Learning Algorithms", Fallah et al., AISTATS 2020.

[3] "Meta-learning with differentiable closed-form solvers", Bertinetto et al., ICLR 2019.

**Time Spent Reviewing:**

8

---

> ### Author Response · Authors · 2021-08-08
> **Response to Reviewer czkt**
>
> Thank you for your feedback, it is very helpful.
>
> Re: analytical setting. Our setting doesn't take away from the fundamental impact of this work: First, proposed analysis can be extended to loss functions beyond quadratic. We use the fairly general analysis framework of Thrampoulidis et al. COLT'15. For instance, it is recently adapted (by Montanari et al. as well as others) to study max-margin classifiers (which implicitly connects to logistic regression). Our approach can also be adapted similarly.
>
> Second, it is well-understood that linear models can capture the theoretical phenomena surrounding deep learning. In fact, a large chunk of recent theoretical results on deep nets are using linearization trick based on neural tangent kernel (Jacot et al NeurIPS'19). Closer to our work, benefits of overparameterization (e.g. double descent, implicit bias) can also be rigorously characterized thanks to linear(ized) models (see works by Belkin, Montanari, Mahoney and others). These works also show that linear models with general feature covariance provide a rich playground for understanding deep nets. This is exactly why a core focus of our work is understanding the impact of feature/task covariances on meta-learning sample complexity. More broadly, a primary contribution of our work is extending these recent fundamental results (which are restricted to a single task) to the meta-learning setting.
>
> Third, linear meta-learning is a fundamental problem that received a lot of recent attention. Our fundamentally-different analysis also provides a more refined treatment compared to existing papers on this topic (e.g. papers by Tripuraneni et al. ICML'21, Du et al. ICLR'21, Kong et al. ICML'20).
>
> We will highlight these connections and further clarify the fundamental impact of this work in the revised version.
>
> Also, thank you for the references [1,2]. We will provide a thorough discussion of the literature on the nonlinear setting.
>
> We have experiments on real world data, such as Fig. 1a and Line 234-247. Reviewer PuM5 said "An important phenomenon of alignment of feature covariance and task covariance is introduced and its effect on sample complexity and optimal dimensionality is studied. Experiments on simulated data verify some of findings and the presence of the alignment is observed on some real datasets (MNIST, CIFAR)."
>
> The overall idea of R2D2 is less relevant. It proposes a quasi-Newton type algorithm (7) that in practice accelerates the convergence, usually when the true parameter is sparse. We emphasize on the statistical property of the method of moment estimator and the optimal representation matrix. The idea of "feature extractor" (i.e., taking the last layer of neural network as the feature space) is discussed in previous paragraphs.

---

> ### Author Response · Authors · 2021-08-31
> **Follow-up on response**
>
> Thank you for taking the time to review the paper and provide feedback. Since the discussion period is drawing to a close, we wanted to reach out to see if you had any pending concerns or questions; we would be happy to discuss further. Below, we elaborate aspects of our earlier response to re-address concerns that may not have been clear earlier. Primarily, we hope to A) clarify the motivation for the design of the theoretical setup and B) describe the takeaways from the theory from the perspective of practice by revisiting the experiments from the paper.
>
> ### Rationale for theoretical setup
>
> The simplicity of the proposed theoretical setup is intentional because it allows us to isolate and identify interesting phenomena. The paper demonstrates that the linear meta-learning setting had many unanswered open questions and is already rich enough to reveal intriguing and surprising results:
>
> 1. Double-descent (benefit of overparameterization) in the few-shot phase only takes place when the representation from the meta-learning phase is of high-enough quality.  This can be visualized in Figure~5, where the curve is monotonically decreasing only for high enough N.
> 2. Method-of-moments covariance estimation can work even when number of samples << degrees of freedom.
>
> It is rich findings such as the above in simple linear frameworks that have caused linear meta-learning to become a fundamental theoretical problem that has received a lot of attention. The insights from the current paper pave the way for more general cases to be explored in future work.
>
>
> With regards to the genericity of the results, there are reasons to believe that the intuitions developed from the linear case transfer to more general cases:
>
> 1. The proposed analysis can be extended to loss functions beyond quadratic. We use the fairly general analysis framework of Thrampoulidis et al. COLT'15. In particular, recent theoretical work has shown an equivalence between different loss functions, which indicates that insights from the squared loss would transfer to the max-margin SVM [A] and the logistic loss [B]. For instance, the same techniques were recently adapted (by Montanari et al. as well as others) to study max-margin classifiers (which implicitly connects to logistic regression). Our approach can also be adapted similarly. We will add a remark to the paper to clarify this.
> 2. There has been a recent trend in deep-learning theory to rely on linear models to capture various theoretical phenomena:
>     1. A large chunk of recent theoretical results on deep nets use the linearization trick based on neural tangent kernel (Jacot et al NeurIPS'19).
>     2. Closer to our work, benefits of overparameterization (e.g. double descent, implicit bias) can be rigorously characterized thanks to linear(ized) models (see works by Belkin, Montanari, Mahoney and others). These works also show that linear models with general feature covariance provide a rich playground for understanding deep nets. This is why a core focus of our work is understanding the impact of feature/task covariances on meta-learning sample complexity. A primary contribution of our work is extending these recent fundamental results (which are restricted to a single task) to the meta-learning setting.
> 3. The theory was motivated by an experimental observation in Fig~1(a), and the findings regarding canonical covariance were replicated experimentally as well. We elaborate further on these empirical connections below.
>
> We will highlight these connections and further clarify the impact of this work in the revised version.
>
> ### Remarks on [1,2] and challenges of nonconvexity
>
> Thank you for the references [1,2]. In [1], the authors study meta-learning in a classification setup, motivated by prototypical networks that use a neural-network to compute a representation $\phi$. However, their theory makes the simplifying modeling assumptions that the distribution of $\phi(x)$  given any class assignment is normally distributed i.e
> $
> p (\phi(x) | Y(x) = c) \sim \mathcal{N} (\mu_c, \Sigma_c),
> $
> and that the covariance for all classes in the embedding space is equal $\forall (c, c’), \Sigma_c = \Sigma_{c’}$. Hence, the task that the meta-learner is faced with is covariance estimation, similar to our setup and hence not more general. While their main contribution is to study robustness of learned-representation to the number $n_2$ of few-shot samples, our work studies the role of overparameterization in meta-learning and hence addresses a different important question.
>
> The authors of [2] study the convergence behavior of gradient-based algorithms for meta-learning at training time on nonconvex loss functions. However, their work is mostly optimization-focused whilst ours provides statistical guarantees on the generalization ability of learned meta-learning models. Hence, the challenges introduced by nonconvexity in the two cases are not comparable. The R2D2 paper suggested is falls into a similar category of works as [2], and is hence dissimilar from the present work.
>
> To give an indication of what the study of a nonconvex model would look like, consider the simple case where the features are lifted by some nonconvex function $\phi$ and the eigenweighting is applied on the lifted-features instead. To study the generalization properties in this model, one would need to understand the random-matrix concentration properties of the lifted training matrix $\phi(X)$, and this would depend heavily on the specific choice of $\phi$. There are no general tools from random-matrix-theory that can be applied to this end, which explains why the literature even in the non-meta-learning setups is sparse for such settings. The only instance that we are aware of that considers such a setting is [C] on the Fourier-feature-map that, while interesting, uses a nonstandard grid-sampling technique to enable theory.
>
>
> ### Experimental Corroboration of Theory
>
> The theory was motivated by a simulation in Transfer Learning in Figure1(a). Here, features from a ResNet50 deepnet, pretrained on Imagenet, are used to train a linear classifier on CIFAR-10. It can be observed that as the dimension R is increased, few-shot test loss reduces, i.e., we observe double-descent at test time. One of the fundamental questions that our work addresses is what properties the learned representation has to possess for such a phenomenon to manifest. The experimental configuration is detailed in Lines 34-41.
>
> In the theory, we find that the alignment of feature covariance and task covariance is helpful to meta-learning. Experiments in the supplementary material verify that this alignment manifests even in practice for real-datasets (MNIST, CIFAR) which are known to benefit from representation learning. Reviewer PuM5 agrees: “Experiments on simulated data verify some of findings and the presence of the alignment is observed on some real datasets (MNIST, CIFAR).”
>
>
> ### References:
> [A] Hsu, Daniel, Vidya Muthukumar, and Ji Xu. "On the proliferation of support vectors in high dimensions." International Conference on Artificial Intelligence and Statistics. PMLR, 2021.
>
> [B] Soudry, Daniel, et al. "The implicit bias of gradient descent on separable data." The Journal of Machine Learning Research 19.1 (2018): 2822-2878.
>
> [C] Muthukumar, Vidya, et al. "Harmless interpolation of noisy data in regression." IEEE Journal on Selected Areas in Information Theory 1.1 (2020): 67-83.

---

### Official Review · Reviewer_PuM5 · 2021-07-17

**Rating:** 7
**Confidence:** 3

**Summary:**

This paper studies overparametrized representation learning in meta-learning, i.e. representation dimensionality (R) is larger than number of samples ($n_2$) per task. It does so in the setting of linear representations, but goes beyond existing analyses that typically assume that the covariance of task regressors to be low rank and restrict representation dimensionality to be small. The paper characterizes the optimal representation for every dimensionality R (in an asymptotic regime), observes a double descent phenomenon w.r.t. R in the performance of the representation on test tasks, also concludes that the optimal dimensionality can be overparametrized in many cases. It then analyzes the sample complexity of learning such a representation with finite train tasks samples, and finds that choosing R adaptively with number of samples can strike a balance between bias and variance of the representation. An important phenomenon of alignment of feature covariance and task covariance is introduced and its effect on sample complexity and optimal dimensionality is studied. Experiments on simulated data verify some of findings and the presence of the alignment is observed on some real datasets (MNIST, CIFAR).

**Limitations And Societal Impact:**

Limitations of the work are discussed in Section 6.

**Main Review:**

Originality:

This paper meaningfully extends the linear representation learning by studying the role and need for high dimensionality of representations when the features and tasks are not exactly low rank, unlike recent analyses that make such a low rank assumption. It makes good use of previously employed asymptotic analysis for regularized linear regression. I found the discussion and comparison to previous work adequate. A relevant work is [1] that also analyzes high dimensional representations for (overparametrized) meta-learning, but relies on gradient dynamics to automatically uncover the underlying subspace structure.

Quality:

The results seem believable from a skim over the proofs. I didn't verify all the details, especially for the closed form expression in Theorem 1. Technical claims are supported by proofs in the appendix and with simulation experiments in some places.

Clarity:

I found the paper mostly clearly written and easy to follow. I found the discussion before Theorem 2 a bit clunky (line 260-276), since $s_F$ and $s_T$ are only defined in Table 2. I would encourage the authors to rework this part a bit.

Significance:

The paper analyzes more general and realistic settings than those analyzed in previous work on meta-learning linear representations and explores the benefit of covariance alignment in sample complexity. The main contribution is theoretical and it does not necessarily provide any practical algorithmic suggestions. It needs to make simplifying assumptions in some cases, like asymptotic analysis, eigenspace alignment and linear representations, but these do not significantly dilute the conceptual contribution in my opinion.


Additional comments:

- L84: Is DoF degree of freedom? not defined anywhere

- L194: p=R?

- L242,243: Something seems off in the alignment metrics. canonical-identity alignment is equal to 1/d in the way it is defined currently



---------------------------------------------------------------------------------------------------------------------
**Post rebuttal**: I thank the authors for their many responses to reviewer concerns. After reading everything and the reviewer discussions, I am inclined to retain my recommendation of accept (7). I think providing some more details and highlighting the conclusions from Fig 1(a) and 1(b) even later in the paper will help. One of the major concerns raised by other reviewers (which I agree with) is the clarity of the paper. More specifically, Theorem 1 can afford to be written much more clearly, especially the part about how the limit of $n\_2\rightarrow\infty$ is taken. The author response provides a lot of clarification in this regard, but I think it can be improved even further, e.g. the statement "the joint empirical distribution $\frac{1}{R} \sum_{i=1}^{R} \delta_{\Lambda_{R_i}, \tilde{\Sigma}^R_{T_i}}$ converges to a fixed distribution" seems very important but is inadequately discussed. Many readers of this paper will not be familiar with the kind of asymptotic results and analyses presented in this paper, so it will help to clearly explain how the limits are taken, why an asymptotic analysis is needed, and what the results mean for the practical case of finite $d,n\_2,R$. Also, some of the empirical verifications with CIFAR-10 and MNIST and a good suggestion made by another reviewer in the discussions was to extend this analysis to more few-shot learning based datasets. Personally I think that these changes are not too drastic to change my opinions about the paper.


[1] N. Saunshi, Y. Zhang, M. Khodak, S. Arora. ICML 2020. A Sample Complexity Separation between Non-Convex and Convex
Meta-Learning.

**Time Spent Reviewing:**

6

---

> ### Author Response · Authors · 2021-08-08
> **Response to Reviewer PuM5**
>
> Thank you for the positive feedback. We agree that reference [1] is relevant and we will cite it in the final version.
>
> L84: DoF stands for the "degrees of freedom".
> L194: This is a typo, it should be $p\leftarrow R$.
> L242,243: Thanks for pointing this out. Actually we found this typo immediately after the submission and the correct definition is provided in our supplementary material. See Line 300 of the supplementary for the corrected definition in red color.
>
> Regarding asymptotic risk: We agree that we only calculate the risk for an infinite dimensional problem; however, we used the experiment in Fig. 1b  to demonstrate that the finite dimensional case displays a similar behavior. The dots in Fig. 1b are experimental values for finite dimensional data and the curves are the theoretical value for infinite dimensional case, one can see they match perfectly.

---

### Author Response · Authors · 2021-08-08
**Summary of our contributions**

We thank the reviewers for their feedback. To guide the discussion, here, we clarify recurring questions and re-emphasize our key novelties compared to the prior art.

To the best of our knowledge, this is the first work that explains how and when overparameterized (high-dimensional) representations can be provably favorable to low-dimensional representations. This is in contrast to many related results that are restricted to low-dimensional representations to ensure the sample efficiency of the few-shot phase. While a few works provide statistical guarantees for overparameterized representations (such as [Du et al. ICLR'21]., these employ classical norm-based generalization arguments which only provide rather loose upper bounds and cannot deduce the statistical benefit over low-dimensional representations (e.g., see arxiv.org/abs/1902.04742 by Nagarajan and Kolter).
Our main result (Theorem 3) accomplishes this feat via more refined techniques from asymptotic analysis that allow for sharp characterization of the few-shot learning risk (upper and lower bounds). This also reveals a novel message: As the sample size of the source tasks grow, representation quality improves and it becomes provably better to use an overparameterized representation for the target few-shot learning task.

Our analytical framework is *fundamentally different* than other recent meta-learning works (e.g. Tripuraneni et al. ICML'21, Du et al. ICLR'21, Kong et al. ICML'20) and, for the first time, establishes a bridge between high-dimensional analysis (e.g. works by Montanari, Mahoney, Krzakala) to representation-based meta-learning to establish sharper refined bounds.

Secondly, while the setting is linear, our results apply under arbitrary feature/task covariances. Recent literature on deep learning [see works by Montanari, Belkin, Mahoney and others] makes it clear that, linear models with proper feature covariance opens the door to understanding random features which then connects to deep nets (e.g. via kernelization trick of Jacot et al.). To this end, Section 4 establishes the first covariance-aware sample complexity bounds for representation learning which capture sample size in terms of effective dimensionalities of feature/task covariances.

Finally, even for the special but fundamental setting of "identity feature covariance, rank-r task covariance", we establish the first optimal sample complexity guarantees to achieve $O(dr)$ which improves over the $O(dr^2)$ bound of very recent work by [Tripuraneni, Jin, Jordan ICML'21] (see our Table 2 for details).

---

### Author Response · Authors · 2021-09-02
**Thank you for your time and effort. We would be happy to address further questions.**

Dear Reviewers and Area Chair,

We would like to thank all of you for the time and effort you spent reviewing our paper and following up with the discussion phase. This has been incredibly helpful in improving the clarity and thoroughness of the revised manuscript. If you have further suggestions or concerns we would be grateful to hear them (specifically any remaining concerns of Reviewers czkt and gmou).

We know that the review process has required an extra effort this year and we genuinely appreciate it.

Authors

---

### Decision · Program_Chairs · 2021-09-27

**Decision:**

Accept (Poster)

**Comment:**

This paper studies overparameterization for meta-learning in the setting of linear-regression tasks. It answers the questions about the optimal linear representation of features for a new downstream task and the number of samples needed to build this representation. The reviewers are quite diverse at the beginning and raised a number of questions, with one of the main criticisms being on the clarity of Theorem 1. The authors have done a good job at replying the reviews, and the reviewers also have done a good job at discussing the rebuttal. Finally, the originally negative reviewers are able to converge to the borderline and the positive reviewers maintained their scores. I propose acceptance of this paper due to the important contribution of demonstrating overparameterization to meta learning. However, I highly recommend the authors to take the comments from the reviewers and revise the paper carefully for its final version. It would make the paper even stronger if the revision can somehow address the following question raised in the discussion: it is expected the paper can provide either stronger empirical verifications (MNIST and CIFAR100 in the appendix are non-standard in FSL) or for theory beyond linear regression case.